# Biocompatible aggregation-induced emission active polyphosphate-manganese nanosheets with glutamine synthetase-like activity in excitotoxic nerve cells

Jing Wang[1], Xinyang Zhao [1], Yucheng Tao[2], Xiuxiu Wang[1], Li Yan[3], Kuang Yu[4], Yi Hsu [5], Yuncong Chen [1,3] ✉, Jing Zhao [1,3,6] ✉, Yong Huang [7] ✉ & Wei Wei [1,2,3,6] ✉

Glutamine synthetase (GS) is vital in maintaining ammonia and glutamate (Glu) homeostasis in living organisms. However, the natural enzyme relies on adenosine triphosphate (ATP) to activate Glu, resulting in impaired GS function during ATP-deficient neurotoxic events. To date, no reports demonstrate using artificial nanostructures to mimic GS function. In this study, we synthesize aggregation-induced emission active polyP-Mn nanosheets (STPE-PMNSs) based on end-labeled polyphosphate (polyP), exhibiting remarkable GS-like activity independent of ATP presence. Further investigation reveals polyP in STPE-PMNSs serves as phosphate source to activate Glu at low ATP levels. This self-feeding mechanism offers a significant advantage in regulating Glu homeostasis at reduced ATP levels in nerve cells during excitotoxic conditions. STPE-PMNSs can effectively promote the conversion of Glu to glutamine (Gln) in excitatory neurotoxic human neuroblastoma cells (SH-SY5Y) and alleviate Glu-induced neurotoxicity. Additionally, the fluorescence signal of nanosheets enables precise monitoring of the subcellular distribution of STPE-PMNSs. More importantly, the intracellular fluorescence signal is enhanced in a conversion-responsive manner, allowing real-time tracking of reaction progression. This study presents a self-sustaining strategy to address GS functional impairment caused by ATP deficiency in nerve cells during neurotoxic events. Furthermore, it offers a fresh perspective on the potential biological applications of polyP-based nanostructures.

Glutamine synthetase (GS) is a vital enzyme ubiquitous in all living organisms[1–3]. It catalyzes the conversion of glutamate (Glu) and ammonia to glutamine (Gln), which plays a pivotal role in ammonia detoxification[4], interorgan nitrogen flux[4,5], acid-base homeostasis[6], and cell signaling[7]. In the mammalian brain, GS protects neurons from excitatory toxicity, which is intricately linked to the metabolism of excitatory neurotransmitters, Glu, whose homeostasis is essential for preventing ischemic stroke and developing neurodegenerative diseases[8]. Furthermore, research has demonstrated that excitotoxic nerve cells undergo Ca²⁺ influx, leading to mitochondrial depolarization and a subsequent decrease in adenosine triphosphate (ATP) levels[8–10], which further limits the ATP-dependent GS activity.

Consequently, the restoration of GS activity under such conditions would necessitate an alternative mechanism that can function independently of ATP concentrations. In addition, the urgent need for artificial GS development is underscored by the existing challenges faced with recombinant GS, which has seen significant research in the fields of biomedicine[11,12], and genetic engineering[13,14]. These challenges include limited yields, elevated costs, and suboptimal stability, necessitating the pursuit of novel and effective alternatives to overcome these limitations.

Recent advances in biopolymers have shown considerable promise in catalysis due to their excellent biocompatibility and abundance of editable sites[15–17]. Inspired by these developments, we have focused on developing polyphosphate (polyP) nanotechnology and exploring its potential applications in biomimetic catalysis. PolyP is an inorganic polyanionic biopolymer composed of orthophosphate subunits. Present in all living organisms, polyP plays a crucial role in a diverse range of physiological processes[18–22]. The negatively charged multifunctional polymer contains phosphoanhydride bonds that are isoenergetic to those found in ATP[23]. Previous research has demonstrated that the structure of polyP can modified and programmed to perform specific functions at the nanoscale in bone repair[24–27], clotting[28–32], and immunotherapy[33]. Additionally, certain polyP-based nanostructures exhibit close similarities to natural proteins in both morphology and function. However, despite these exciting developments, the potential of polyP-based nanostructures in biomimetic catalysis remains largely unexplored.

To evaluate subcellular localization and monitor the progress of reactions in biological systems, rapid, synchronous, and non-destructive visualization is crucial[34–37]. However, most inorganic nanomaterials used in these applications lack fluorescence properties[38,39]. The concept of aggregation-induced emission (AIE), introduced by Tang et al. in 2001[40], offers a revolutionary solution. Nonemissive AIEgens can be induced to undergo fluorescent emission through the restriction of intramolecular motion (RIM) effect associated with aggregation inherent to self-assembly processes of nanostructures[41,42]. AIE technology enables the long-term fluorescence tracking of dynamic biological processes with high photostability, making it a powerful tool in nanoscience[43–49]. By utilizing AIE-active nanostructures as catalysts, it becomes possible to monitor the progress of biochemical reactions through the detection of alterations in emission intensity or wavelength.

In this work, we end-label polyP with water-soluble tetraphenylethene (STPE) and introduce $Mn^{2+}$ to obtain AIE-active STPE-PMNSs. The resulting fluorescent nanosheets exhibit GS-like activity, effectively promoting the conversion of Glu to Gln even in the absence of ATP. This self-feeding mechanism compensates for the deficiencies of natural GS function caused by reduced intracellular ATP levels during excitotoxic events, enabling efficient conversion of Glu to Gln by STPE-PMNSs in an ATP-deficient environment. Consequently, STPE-PMNSs can efficiently promote the conversion of Glu in excitatory neurotoxic SH-SY5Y cells and alleviate Glu-induced neurotoxicity. Furthermore, the AIE signals can track subcellular compartment localization. In the absence of ATP, the Gln synthesis promotes nanosheet aggregation, resulting in an enhanced fluorescence signal that enables real-time monitoring of intracellular Gln levels. This study presents the nanostructures with ATP-independent GS-like activity that can efficiently convert Glu to Gln in ATP-deprived environments.

## Results

### Synthesis and characterization of STPE-PMNSs

The synthesis of STPE-PMNSs was achieved by first attaching the water-soluble AIEgens (STPE) to the terminal $-NH_2$ of polyP, following a hierarchical assembly strategy using STPE-polyP and $MnCl_2$ as reaction material (Fig. 1)[50–52]. The synthesis and characterization details are fully described in the Supplementary Information (Supplementary

Figs. 1–13). The final products were observed to be layered structures with a lateral size of 110–150 nm, consistent with the results of dynamic light scattering (DLS) (Fig. 2a and Supplementary Fig. 14b). The powder X-ray diffraction (PXRD) pattern showed an overall amorphous sample, rather than a crystal (Supplementary Fig. 14a). However, high-angle annular dark-field scanning transmission electron microscope (HAADF-STEM) images of STPE-PMNSs in certain regions revealed interatomic distances between the two Mn atoms in the transverse and longitudinal orientations are 3.57 Å and 5.25 Å, respectively (Fig. 2b). These results indicated STPE-PMNSs may exhibit partially crystalline nature. Energy-dispersive X-ray spectroscopy (EDXS) elemental mapping showed an even distribution of P, Mn, O, N, and S on the nanosheets, revealing the successful introduction of STPE (Fig. 2c). The zeta potential value of STPE-PMNSs is 12.62 eV, with no significant difference compared with PMNSs (Fig. 2d and Supplementary Fig. 15). Raman, Fourier-transform infrared (FT-IR), and X-ray photoelectron spectroscopy (XPS) results further demonstrated the structure of STPE-PMNSs. Raman analysis of the STPE-PMNSs showed major bands located at around 717.0, 616.8, 566.0, and 455.0 $cm^{-1}$, corresponding to Mn-O symmetric stretching. The peak at 944.0 $cm^{-1}$ is attributed to intramolecular symmetric vibrations of $PO_4^{3-}$, whereas the weaker peaks at 891.9, 980.1, and 1023.7 $cm^{-1}$ were associated with the bending and asymmetric stretching of $PO_4^{3-}$ (Supplementary Fig. 16b). FT-IR spectra showed distinct differences between PMNSs and STPE-PMNSs. The latter exhibit asymmetric stretching vibrations (1623 $cm^{-1}$, 1509 $cm^{-1}$, 1455 $cm^{-1}$) for the carbon skeleton of the benzene ring as well as $v_{S=O}$ and $v_{N-H}$ at 1398 $cm^{-1}$ and 3413 $cm^{-1}$, respectively (Supplementary Fig. 16a), indicating successful introduction of STPE into the nanosheets. XPS analysis of Mn $2p$ showed the predominant oxidation state of Mn is $Mn^{(II)}$. Significantly, the O $1s$ XPS spectrum revealed an Mn-O to P-O bond ratio of approximately 1:1 (Fig. 2e–g and Supplementary Fig. 17), in agreement with theoretical simulation. Additionally, we investigated the fluorescent properties of STPE-PMNSs and found that the fluorescence emission of STPE-polyP and STPE-PMNSs were at the same position (Supplementary Figs. 18 and 19).

### STPE-PMNSs promote the extracellular conversion of Glu to Gln

To evaluate the potential of STPE-PMNSs in biocatalysis, we examined their superoxide dismutase (SOD)-like, glutathione peroxidase (GPx)-like, catalase (CAT)-like, lactate dehydrogenase (LDH)-like, cytochrome oxidase (COX)-like, and GS-like activities. Supplementary Fig. 20 revealed mild SOD, GPx, and CAT activity, but no LDH and COX activity. Interestingly, the synthesized nanosheets demonstrated high GS-like activity (Fig. 3a). Moreover, control experiments indicated that the introduction of STPE did not significantly affect the activity of the nanosheets (Fig. 3b). Intriguingly, compared to natural GS, the STPE-PMNSs showed comparable activity even in the absence of ATP (Fig. 3c and Supplementary Fig. 21). Gln formation reached 75.6% and 76.3% conversion in the absence and presence of ATP, respectively (Supplementary Figs. 23–43). We further investigated the relationship between GS-like activity and the concentration of STPE-PMNSs. As depicted in Fig. 3d, GS-like activity increased within the concentration range of 0–200 μg $ml^{-1}$, which agrees with the observations in the colorimetric assays (Supplementary Fig. 22).

To gain further insight into the reaction kinetics, we determined the reaction rates under various concentrations of Glu, $NH_4^+$, and ATP (Supplementary Figs. 44–47 and Supplementary Table 1). The kinetics equation of STPE-PMNSs is fitted by the Michaelis–Menten equation as follows:

$$V = V_{max}[S]/(K_m + [S]) \tag{1}$$

where $V$ is the initial reaction rate, $V_{max}$ is the maximal velocity, $[S]$ is the concentration of substrate, and $K_m$ is the Michaelis constant. The reaction rate aligned well with Michaelis–Menten kinetics, exhibiting

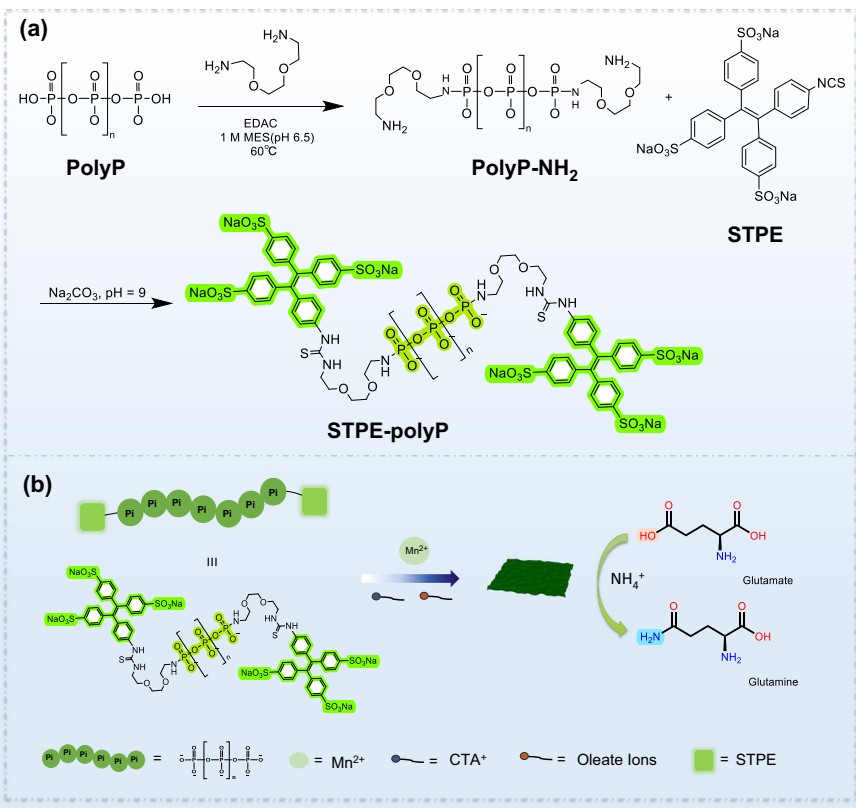

**Fig. 1 | Schematic illustration of the fabrication process of the STPE-polyP and STPE-PMNSs. a** Synthetic route to modification of polyP. **b** Summary diagram of the synthesis of STPE-PMNSs and the promotion of conversion of Glu to Gln.

enzyme-like first-order kinetic behavior. The apparent $K_m$ values for Glu, $NH_4^+$, and ATP were 0.75 mM, 0.099 mM, and 0.52 mM, respectively (Fig. 3e–g). It appears that STPE-PMNSs display a higher affinity for substrates than natural GS (apparent $K_m$ for Glu, 2.04 mM; $NH_4^+$, 0.16 mM; ATP, 1.84 mM) (Supplementary Figs. 48–51 and Supplementary Table 2).

To evaluate the reusability of STPE-PMNSs in comparison to natural GS, we resubjected the recycled nanosheets to the reaction after ultra-filtration. The GS concentration was determined using the BCA method before each cycle, ensuring no GS loss after ultrafiltration (Supplementary Fig. 52). Remarkably, STPE-PMNSs maintained over 69% of their activity after three cycles. The presence of ATP enhanced the stability of STPE-PMNSs (Fig. 3h), as the phosphorous content in polyP remained constant (little leaching), unlike when ATP was absent (Fig. 4b). In contrast, GS experienced a 75.1% activity loss in the second cycle and over lost 90% activity in the third cycle (Supplementary Fig. 53a).

We also investigated the effects of reaction conditions on the GS-like activity of STPE-PMNSs. The results indicated that STPE-PMNSs could sustain more than 72.4% activity under a broad scope of conditions, including temperatures up to 60 °C, pH levels ranging from 6 to 10, and 80% organic solvents (MeOH, DMF, MeCN, DMSO) (Fig. 3i, j and Supplementary Figs. 53 and 54).

Furthermore, we assessed the influence of STPE-PMNSs morphologies on the GS-like activity by comparing nanosheets of varying thicknesses. The results revealed that the thinner nanosheets (2.26 nm) exhibited the highest activity, while the nanoparticles showed the lowest activity (Supplementary Figs. 55 and 56). These findings suggest the activity of STPE-PMNSs is inversely proportional to their thickness.

**Investigation of the mode of action underlying the GS-like activity of STPE-PMNSs**

The role of STPE-PMNSs appears to vary under different reaction conditions. The polyP in nanosheets can function as a phosphate

donor to activate Glu in the absence of ATP. This role is evident from Fig. 4a, b, which shows a gradual decrease in phosphorus content within the nanosheets over time, indicating phosphate leaching from polyP (Supplementary Figs 57–59). Additionally, in the absence of ATP, fluorescence intensity was increased during the reaction process, and significant aggregation of STPE-PMNSs was observed post-reaction, supporting a change of structure (Fig. 4c–e and Supplementary Figs 60 and 61). These findings suggest that the polyP component of STPE-PMNSs serves as the phosphate donor when there are no external phosphates added (i.e., ATP).

STPE-PMNSs may play two roles in the conversion of Glu to Gln. First, STPE-PMNSs serve as a phosphate source through their polyP component. Neither ATP nor $NH_4^+$ alone could drive the reaction (Fig. 4g). Additionally, polyP or $Mn^{2+}$ individually did not promote the conversion (Fig. 4f). This indicates that STPE-PMNSs provide more than just phosphate. Second, the Mn ions within STPE-PMNSs may have catalytic functions similar to those in natural GS. Nanosized polyP-Ca or $MnO_2$ separately lacked the GS-like activity, while the STPE-PMNSs containing both polyP and $Mn^{2+}$ exhibited activity (Fig. 4f). This suggests that $Mn^{2+}$ plays an active role. Specifically, we suspect $Mn^{2+}$ binds the carboxylate of Glu through coordination, lowering the free energy barriers for both phosphate transfer and the subsequent aminolysis.

In contrast, STPE-PMNSs exhibited true catalytic activity when the reaction was conducted in the presence of ATP. The phosphorus content remained stable over time in this scenario, as opposed to when ATP was absent (Fig. 4b). The addition of ATP also enhanced the recyclability of STPE-PMNSs. TEM images revealed that the STPE-PMNSs maintained their structure post-reaction, signifying their catalytic nature when ATP is present (Fig. 4d).

DFT calculations were performed to help understand the ATP-like activity of polyP and the role of Mn ions in this reaction. The following mechanistic scenario is proposed. The initial stage of the reaction

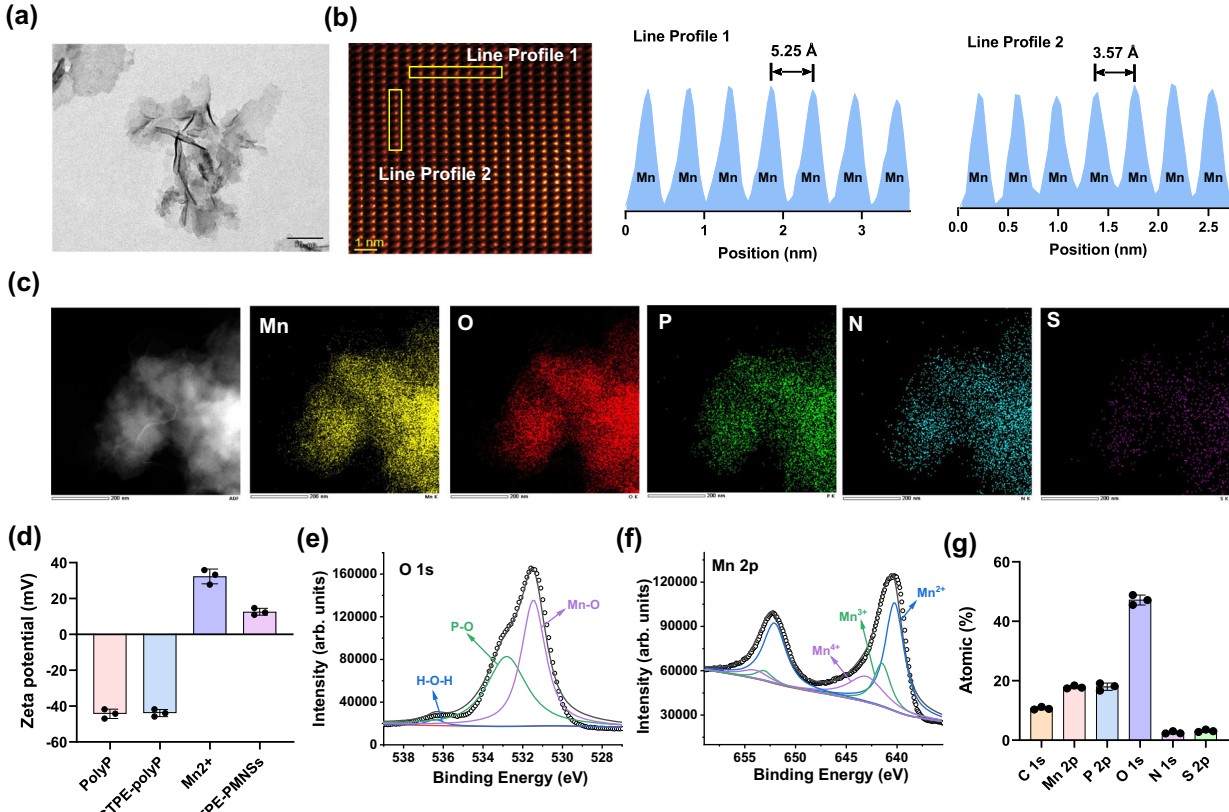

**Fig. 2 | Morphology and structural characterization of the STPE-PMNSs. a** A TEM image of the STPE-PMNSs, Scale bar: 50 nm. **b** A HAADF-STEM image of STPE-PMNSs (left) and intensity surface plot from a yellow line rectangle (middle and right). **c** STEM images and elemental mapping of STPE-PMNSs. Scale bar: 200 nm. **d** Zeta potential of polyP, STPE-polyP, $Mn^{2+}$, and STPE-PMNSs. XPS analysis for O *1s* spectra (**e**) Mn *2p* spectra (**f**), and atomic percentage (**g**) of STPE-PMNSs. Data points presented in (**d**, **g**) represent three (*n* = 3) independent experiments for each experimental group and are displayed as mean ± standard deviation. Experiments of (**a**–**c**) were repeated three times with similar results.

involves the activation of Glu and $NH_4^+$. In the presence of high ATP concentration, both Glu and ATP bind to the $Mn^{2+}$ on the surface of the nanosheets with binding energy of −2.86 eV (denoted as **1** in Fig. 5a). Following this interaction, Glu is phosphorylated by ATP, resulting in the formation of activated γ-glutamyl phosphate and ADP (Fig. 5a). This process is believed to be facilitated by $Mn^{2+}$ in the STPE-PMNS (step 1 in the Fig. 5a). In contrast, without $Mn^{2+}$ (i.e., in the absence of STPE-PMNSs), the phosphorylation of Glu by ATP is significantly less thermodynamically favorable, with energy change of −0.20 eV (Supplementary Fig. 62). This finding suggests that $Mn^{2+}$ ions play a crucial role in the activation of Glu. Specifically, it is reasonable to propose that $Mn^{2+}$ also plays a crucial role in the activation of Glu by coordinating with the carboxylate, thereby promoting phosphorylation and subsequent aminolysis. $NH_4^+$ is presumed to be activated by the basic unlabeled terminal $NH_2$ groups in the nanosheets (Supplementary Fig. 63a, b).

When ATP levels are low or absent, Glu binds to $Mn^{2+}$ on the surface of the nanosheets (denoted as $a_1$, with a binding energy of −0.38 eV), similar to the binding mode in the presence of ATP. Subsequently, Glu is phosphorylated by the nearby polyP, generating a γ-glutamyl phosphate polyoxyethylene bis (amine) species, (the step from $a_1$ to $b_1$ in the revised Fig. 5b). This event also generates new unlabeled terminal phosphates which can activate $NH_4^+$ (Supplementary Fig. 64a, b). The resulting phosphate group exposed nanosheets are still capable of converting Glu to Gln, as shown in Supplementary Fig. 65, DFT calculation verified the reaction could progress continuously. It is important to note that the considerable steric hindrance of the STPE group discourages interactions between Glu and adjacent Mn atoms. Consequently, adsorbed Glu is unable to attack polyP-STPE.

In the second step, Gln is formed through a nucleophilic attack initiated by ammonia on the Cγ of the γ-glutamyl phosphate intermediate. This process involves the nucleophilic attack of ammonia on the Cγ of the γ-glutamyl phosphate intermediate (denoted as 2 in Fig. 5a and denoted $c_1$ in Fig. 5b). Then, after a protonated Gln is formed, it spontaneously transfers a proton to the released phosphate group. The energy profile of the overall reaction is depicted in Supplementary Figs. 63c and 64c, with total reaction energies of −4.60 and −0.77 eV in the presence or absence of ATP, respectively. Notably, our experiments indicate that STPE-PMNSs exhibit partially crystalline nature. The DFT model used in the experiments is based on the crystalline portion of the STPE-PMNSs, taking into account their structural inhomogeneity.

It is important to highlight that ATP and polyP in the nanosheets may compete for Glu activation. Under conditions of a high ATP concentration (above 500 μM), ATP serves as the primary phosphate source, and STPE-PMNSs act as a catalyst for the reaction. Conversely, when ATP levels are low (0–10 μM), the polyP within the nanosheets becomes the essential phosphate donor for Glu activation (Fig. 4h). Additionally, the consumption of phosphate groups in STPE-PMNSs prompted us to explore the duration of conversion activity. Our results demonstrate that the sustainable conversion durations of STPE-PMNSs are 60 h, 48 h, 48 h, and 36 h at Glu concentrations of 2 mM, 10 mM, 20 mM, and 100 mM, respectively (Fig. 4i).

## Subcellular distribution of STPE-PMNSs and their intracellular GS-like activity

Before conducting the reaction in cells, we initially assessed the biocompatibility and intracellular uptake of STPE-PMNSs. The viability of

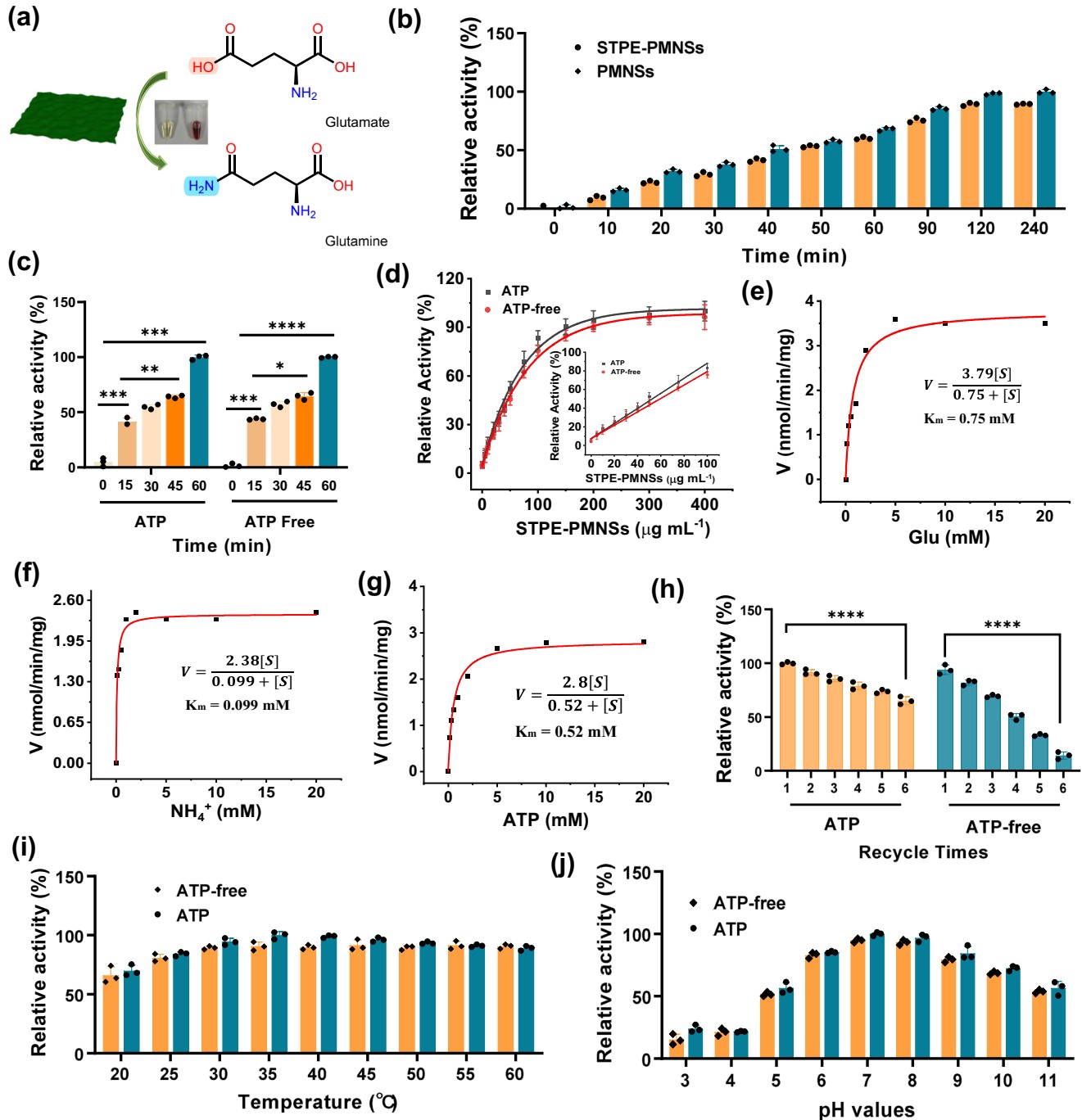

**Fig. 3 | Extracellular activity of STPE-PMNSs in promoting Gln synthesis.**
**a** Schematic representation of GS-like activity of STPE-PMNSs. **b** Comparison of GS-like activity between STPE-PMNSs and PMNSs at various reaction times. **c** Relative activity of STPE-PMNSs with or without ATP. **d** Dose-response of extracellular GS-like activity of STPE-PMNSs in the presence and absence of ATP, showing a linear relation in the low-concentration region (inset). Michaelis-Menten curves of Glu (**e**), $NH_4^+$ (**f**), and ATP (**g**). **h** Reusability of STPE-PMNSs in the presence and absence of ATP (****$p < 0.0001$). Effects of temperature (**i**) and pH (**j**) on the activity of STPE-PMNSs with and without ATP. Data points presented in (**b**–**d**) and (**h**–**j**) represent three ($n = 3$) independent experiments for each experimental group and are displayed as mean ± standard deviation. $p$ values of (**h**) were determined by a two-sided $t$-test.

SH-SY5Y cells remained largely unaffected by a 24-h treatment with STPE-PMNSs and PMNSs (Supplementary Figs. 66 and 67). Furthermore, long-term toxicity test on three types of nerve cells (SH-SY5Y, PC-12, and U87) demonstrated that cell viability persisted at 80% after exposure to STPE-PMNSs (200 μg ml⁻¹) for 7 days indicating excellent biocompatibility (Supplementary Fig. 68a). Besides, we further determined the toxicity of aggregated STPE-PMNSs. The results showed that at a concentration of 200 μg ml⁻¹, the cell survival rate was 78.27%, which was not significantly different from the survival rate when STPE-

PMNSs (74.18%) were administered alone. This suggests that the aggregation of STPE-PMNSs does not increase cytotoxicity (Supplementary Fig. 69). We then examined the intracellular uptake of STPE-PMNSs by SH-SY5Y cells. The intracellular fluorescence signal displayed a gradual increase as the incubation time with STPE-PMNSs lengthened (Supplementary Fig. 70a). ICP−OES results demonstrated that the cellular uptake of the Mn element reached 7.29 ng/10⁴ cells after 24 h (Supplementary Fig. 70b), signifying rapid and robust cellular uptake of STPE-PMNSs, sufficient to exert physiological effects

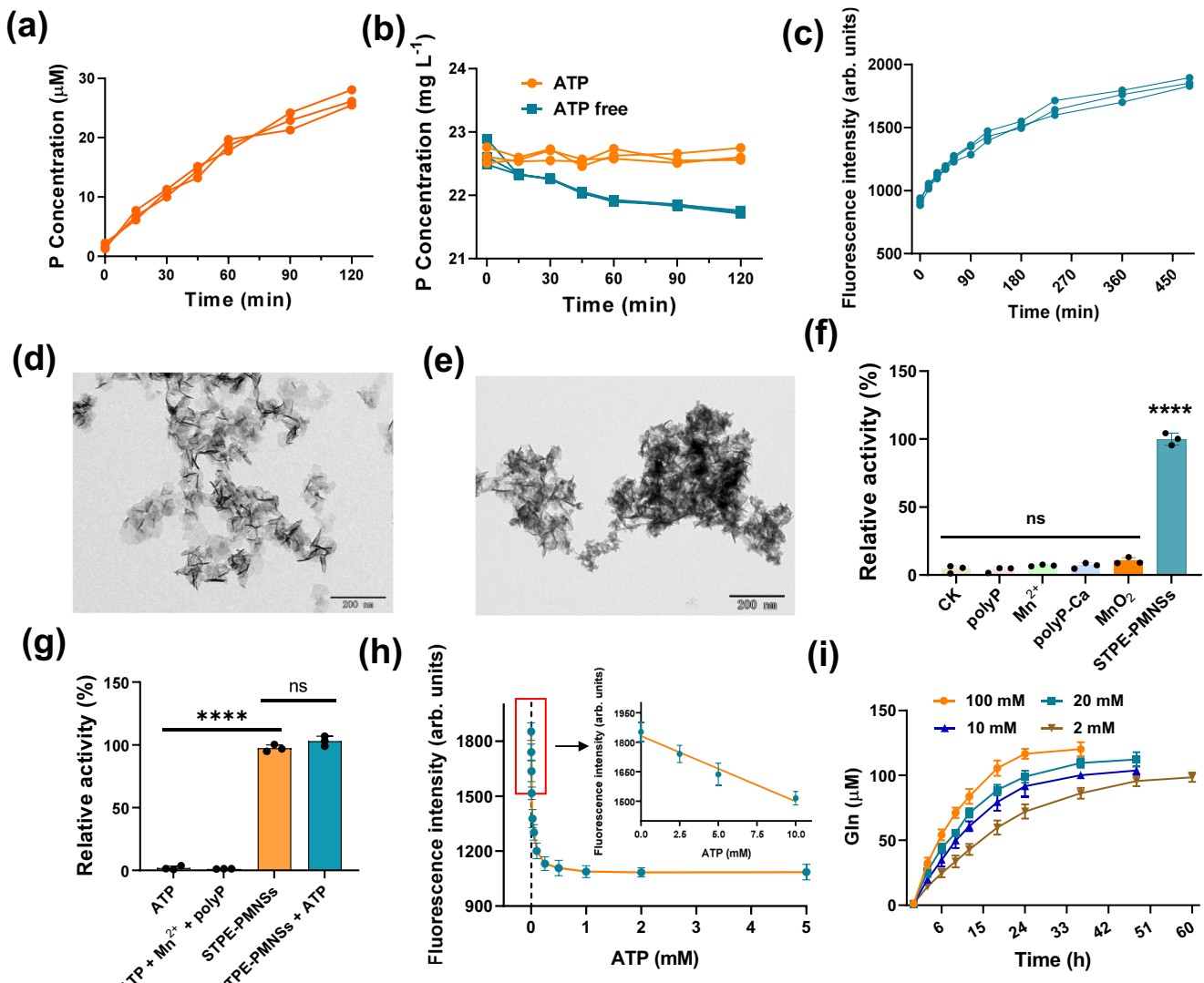

**Fig. 4 | Morphology and structural analysis of post-reaction nanosheets with/ without ATP. a** Free phosphorus (P) level within the reaction solution in the absence of ATP. **b** P content of STPE-PMNSs with (20 mM) or without ATP. **c** Changes in fluorescence intensity during the reaction. TEM image of STPE-PMNSs after 2 h with (**d**) or without (**e**) ATP. **f** Control experiments to confirm whether polyP and $Mn^{2+}$ are required for the GS-like activity (****$p < 0.0001$). **g** Reactions were conducted to determine whether STPE-PMNSs act solely as reaction substrates or possess catalytic activity in the absence of ATP (****$p < 0.0001$). In the absence of STPE-PMNSs, ATP (20 mM) and ammonia alone could not convert Glu to Gln, while the reaction proceeded normally with STPE-PMNSs present. **h** The correlation between fluorescence intensity and ATP concentration in the reaction revealed a linear relationship in the low-concentration region (inset). At a low ATP level, polyP in the nanosheets activates Glu, with the fluorescence intensity at its highest level when there is no ATP. As ATP concentration increases, ATP gradually replaces polyP in activating Glu, causing the fluorescence intensity to decrease. Once ATP fully replaces polyP, the fluorescence intensity stabilizes and no longer declines. **i** Conversion duration of STPE-PMNSs at varying Glu concentrations without ATP. Data points presented in (**f**–**i**) represent three ($n = 3$) independent experiments for each experimental group and are displayed as mean ± standard deviation. **f** was analyzed by one-way ANOVA and **g** was analyzed by a two-sided $t$-test. Experiments of (**d**, **e**) were repeated three times with similar results.

post-ingestion. Additionally, we eliminated the STPE-PMNSs from the culture medium and observed a gradual decrease in intracellular Mn content over time. This suggests that the material can be effectively cleared from the cells (Supplementary Fig. 71). To verify the cellular uptake mechanism of STPE-PMNSs, we utilized bio-transmission electron microscopy (Bio-TEM) to observe cells treated with STPE-PMNSs. The findings indicated that the nanosheets were encased within vesicles and transported intracellularly following treatment with STPE-PMNSs (Supplementary Fig. 72). Moreover, the uptake of STPE-PMNSs by cells under both starvation and normal conditions reveals that the uptake of the STPE-PMNSs uptake necessitates energy expenditure (Supplementary Figs. 73 and 74). Consequently, we deduced that STPE-PMNSs are internalized by cells through endocytosis.

We then explored the subcellular localization of STPE-PMNSs. The results exhibited a significant overlap between the green fluorescence emitted by STPE-PMNSs and the red fluorescence emitted by lysosomes, implying that the primary localization of STPE-PMNSs within cells is within the lysosomes (Supplementary Fig. 75).

Our extracellular experiments have demonstrated a correlation between the aggregation of STPE-PMNSs and ATP concentration. It has been known that nerve cells undergoing excitotoxic events exhibit ATP deficiency[8,9]. To further explore this, we analyzed ATP levels in SH-SY5Y cells stimulated with 4-AP. Our results showed that the ATP content in the 4-AP stimulated group was approximately 6.73 μM, which is half of the level in the control group (Supplementary Fig. 76). This observation implies that nanosheets continue to aggregate

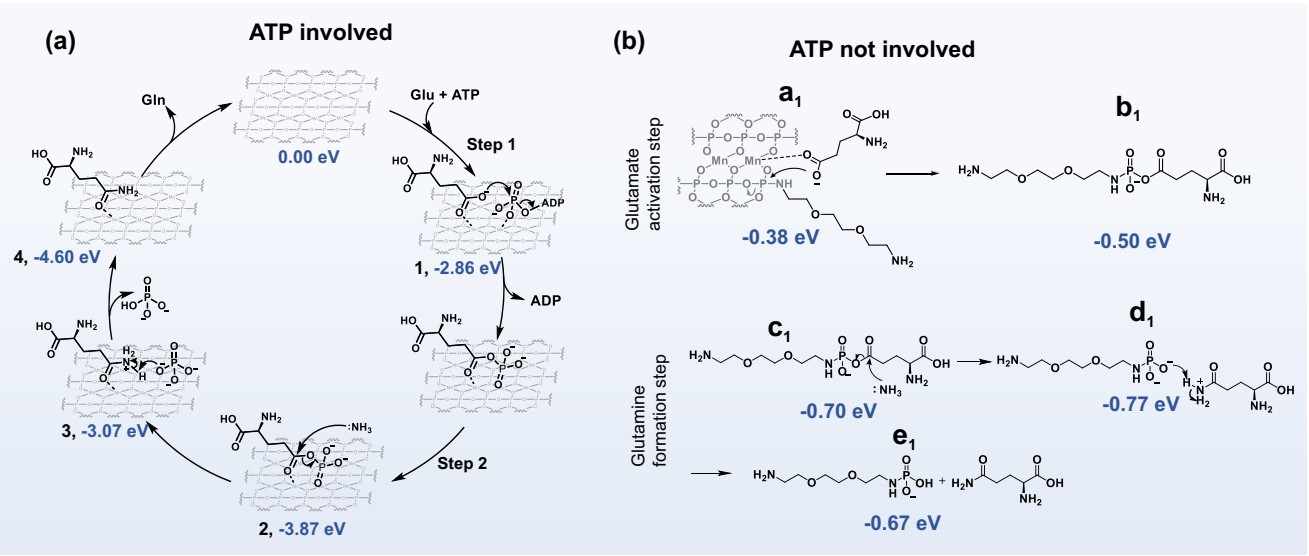

**Fig. 5 | Proposed reaction mechanisms by DFT. a** The proposed catalytic cycle of the GS-like activity of STPE-PMNSs in the presence of ATP. **b** The activation of Glu (up) and the formation of Gln (down) in the absence of ATP.

intracellularly due to phosphate consumption under low ATP conditions.

To evaluate the aggregation of nanosheets intracellularly, we utilized confocal laser scanning microscopy (CLSM) to monitor the real-time distribution of nanosheets within cells and the changes in fluorescence intensity. In the 4-AP and STPE-PMNSs co-treatment group, the fluorescence intensity increased continuously as the reaction progressed, a phenomenon consistent with extracellular observations and potentially attributable to the aggregation of STPE-PMNSs. In contrast, the 4-AP untreated group displayed no significant changes in fluorescence intensity (Fig. 6a). The obtained results prompted our investigation into the feasibility of evaluating the progress of the reaction by measuring changes in intracellular fluorescence intensity. Subsequently, we conducted a relative quantification of fluorescence intensity in Fig. 6a and assessed intracellular Gln levels at the same time intervals (Fig. 6b, c). Our results revealed a linear correlation between alterations in fluorescence intensity and changes in intracellular Gln content (Fig. 6d). Consequently, fluorescence intensity can serve as an indicator of intracellular Gln content during the experiment.

### The protective effect of STPE-PMNSs on nerve cells

Glu serves as the primary excitatory neurotransmitter within the central nervous system and plays a critical role in various cognitive functions, including memory and learning[53]. Numerous studies have highlighted the significant implications of Glu in the development and progression of various neuropathological conditions. For instance, Glu excitotoxicity has been implicated in the pathogenesis of neurological disorders such as cerebral ischemia[54], traumatic brain injury[55], multiple sclerosis[56], epilepsy[57], and Alzheimer's disease[58]. Therefore, targeting excitotoxic events is essential for treating these diseases.

The excellent GS-mimicking activity of STPE-PMNSs prompted us to evaluate their cytoprotection effect on nerve cells. Previous research has demonstrated that Glu-induced excitotoxicity leads to an influx of extracellular $Ca^{2+}$, resulting in nerve apoptosis[6]. Accordingly, we assessed the extracellular $Ca^{2+}$ influx and the effect of apoptosis after STPE-PMNSs treatment. Supplementary Fig. 77 reveals a significant increase in intracellular $Ca^{2+}$ concentration following 4-AP treatment, but a notable decrease after STPE-PMNSs administration. Moreover, the STPE-PMNSs treatment group can rescue Glu-induced

apoptosis, as illustrated in Fig. 6g. These findings align with the observed upregulation of Gln and downregulation of Glu content in the cell lysis of various treatment groups (Fig. 6e, f).

Dose-response experiments indicated that as the concentration of STPE-PMNS increased, intracellular Glu levels decreased correspondingly, while Gln levels increased (Supplementary Fig. 78). Western blot (WB) analysis revealed that the anti-apoptosis protein (Bcl-2) was significantly upregulated after treatment, while the apoptosis protein (Bak) is downregulated (Fig. 6h–k). These results suggest that STPE-PMNSs can promote the conversion of Glu to Gln and rescue Glu-induced nerve apoptosis.

In comparison, $MnO_2$ treatment does not significantly reduce the Glu content in the cells and cannot attenuate Glu-induced neurotoxicity (Supplementary Figs. 79 and 80). Furthermore, we evaluated the Glu scavenging ability of STPE-PMNSs extracellularly. Glu and STPE-PMNSs were co-incubated with the cells, and then the Glu levels in the medium were measured. The findings demonstrate a gradual decrease in Glu levels in the culture medium with increasing co-incubation time with STPE-PMNSs (Supplementary Fig. 81). Additionally, flow cytometry and WB analysis revealed that STPE-PMNSs could alleviate Glu-induced excitotoxicity and rescue nerve cell apoptosis caused by excessive Glu through promoting the extracellular conversion of Glu to Gln (Supplementary Fig. 82). Notably, the activation of Glu by STPE-PMNSs leads to the continuous depletion of the nanosheets. The remaining nanosheets intracellular are eventually transported to the lysosome for digestion and degradation.

## Discussion

We have successfully labeled water-soluble TPE at the end of polyP and synthesized AIE-active STPE-PMNSs by the hierarchical assembly. The prepared STPE-PMNSs exhibit remarkable activity in promoting the conversion of Glu to Gln under physiological conditions, both in the presence and absence of ATP. Further investigation into the mode of action of STPE-PMNSs has revealed that the polyP demonstrates ATP-like functionality, by supplying essential phosphates for Glu activation and accounting for the promoted conversion of Glu to Gln. This ATP-independent activation of Glu by STPE-PMNSs offers potential benefits in regulating Glu homeostasis in ATP-deficient environments. For example, in excitotoxic SH-SY5Y cells, STPE-PMNSs effectively decrease Glu concentration and alleviate Glu-induced neurotoxicity. The AIE fluorescence of the nanosheets enables real-time monitoring

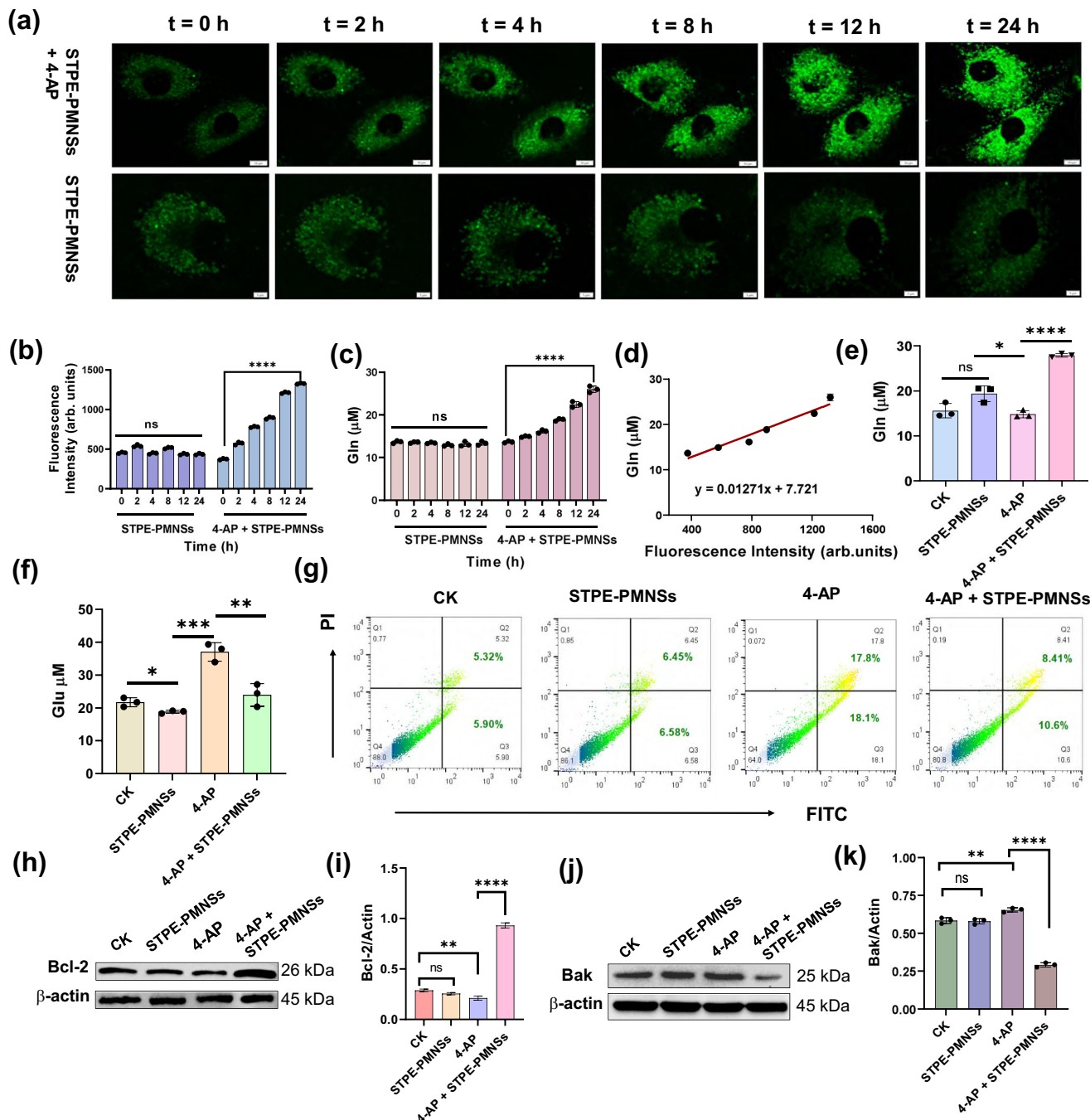

**Fig. 6 | Assessment of the intracellular GS-like activity of STPE-PMNSs.** Intracellular fluorescence signals were monitored at different time points (**a**) (Scale bar: up: 10 μm; down: 5 μm), followed by quantification of the respective fluorescent signals (****$p < 0.0001$) (**b**). Cells underwent treatment with STPE-PMNSs for 6 h, followed by PBS washing. Glu production was induced in SH-SY5Y cells using 4-AP for 4 h, and cells were washed with PBS again. Intracellular fluorescence signals were subsequently observed at different time points, with corresponding signal quantification. Changes in intracellular Gln content were recorded at various time points (**c**) (****$p < 0.0001$), along with the linear relationship between Gln content and fluorescence intensity (**d**). Intracellular Gln (**e**) (*$p = 0.0515$, ****$p = 0.0002$) and Glu (**f**) (*$p = 0.0226$, **$p = 0.3465$, ***$p = 0.0011$) levels in distinct treatment groups were assessed after 24 h. **g** Apoptosis levels in SH-SY5Y cells were evaluated after 24 h across different treatment groups. **h, i** Bcl-2 expression levels in SH-SY5Y cells were measured after 24 h for each treatment group. **$p = 0.0051$, ****$p < 0.0001$. **j, k** Bax expression levels in SH-SY5Y cells were also analyzed after 24 h across treatment groups. **$p = 0.0065$, ****$p < 0.0001$. Data points presented in (**b, c, e, f**) represent three ($n = 3$) biologically independent samples and are displayed as mean ± standard deviation. $p$ values of (**b, c, e, f, i, k**) were determined by a two-sided $t$-test. The samples in (**i, k**) were derived from the same experiment and blots were processed in parallel. Experiments of (**a, g**) were repeated three times with similar results.

of their intracellular distribution. This study represents the application of polyP-based nanostructures in biomimetic reactions, broadening the scope of polyP nanotechnology in biomedical research. Furthermore, by coupling AIEgens to polyP, we showcase a promising strategy for investigating the modes of action in biopolymers.

## Methods
### Cell lines
SH-SY5Y was obtained from the China Type Culture Collection (catalog number: SCSP-5014, CSTR:19375.09.3101HUMSCSP5014). PC-12 cells were obtained from the China Type Culture Collection (catalog

number: SCSP-517, CSTR:19375.09.3101RATSCSP517). U87 cells were obtained from ATCC (HTB-14). The cell lines were listed by the International Cell Line Authentication Committee as cross-contaminated or misidentified. SH-SY5Y cells were cultured in MEM/F12 + 10% FBS, U87 cells were cultured in MEM + 1% Non-Essential Amino Acids (NEAA) + 1 mM Sodium Pyruvate (NaP) + 10% FBS and PC-12 cells were cultured in RPMI-1640 + 10% HS + 5% FBS at 37 °C in a $CO_2$ incubator (95% relative humidity, 5% $CO_2$).

## Preparation of TPE-NH₂

1-Bromo-1,2,2-triphenylethylene (3.35 g, 10.0 mmol 1.0 equiv), (4,4,5,5-Tetramethyl-1,3,2-dioxaborolan-2-yl)aniline (2.30 g, 10.5 mmol 1.05 equiv), potassium carbonate (4.15 g, 30 mmol 3.0 equiv), and tetrakis(triphenylphosphine)palladium (347 mg, 3 mol%) were charged in a 350 ml pressure flask. THF (50 ml) and $H_2O$ (25 ml) were added under a nitrogen atmosphere and the mixture was refluxed overnight. After cooling to room temperature, the mixture was extracted with DCM three times. The combined organic phase was concentrated and 4-(1,2,2-Triphenylethenyl) benzylamine (white solid, 3.0 g, 86%) was afforded by silica gel chromatography (petroleum ether: dichloromethane = 1: 1). $^1H$ NMR (400 MHz, DMSO-$d_6$) δ 7.14 (dd, $J$ = 8.2, 6.4 Hz, 1H), 7.08 (td, $J$ = 6.5, 3.8 Hz, 3H), 7.06–6.98 (m, 3H), 6.98–6.88 (m, 2H), 6.62–6.56 (m, 1H), 6.32–6.26 (m, 1H), 5.05 (s, 1H). $^{13}C$ NMR (101 MHz, DMSO-$d_6$) δ 147.76, 144.67, 144.45, 144.42, 141.67, 138.35, 132.10, 131.36, 131.29, 131.20, 130.79, 128.24, 128.13, 128.03, 126.72, 126.47, 126.45, 113.60, 40.65, 40.44, 40.23, 40.02, 39.81, 39.60, 39.39. HRMS (ESI, $CH_3CN$, m/z, relative intensity): 347.1663 (calcd for $[C_{26}H_{21}N]$ +: 347.1674).

## Preparation of TPE-NCS

We synthesized TPE-NCS according to the existing route with slight changes[59]. 4-(1,2,2-Triphenylethenyl) benzylamine (1.74 g, 5 mmol 1.0 equiv) in 15 ml dry ethanol was added triethylamine (675 µl, 5 mmol 1.0 equiv) and carbon disulfide (3 ml, 50 mmol 10.0 equiv) at room temperature for 30 min. The mixture was cooled in an ice bath and then di-tert-butyl dicarbonate (Boc₂O) (1.08 g, 4.95 mmol 0.99 equiv) dissolved in dry ethanol (5 ml) was added, followed by the immediate addition of DMAP (6 mg, 1 mol %) in dry ethanol (1 ml). The mixture was kept in an ice bath for 10 min and allowed to reach room temperature for another 30 min. The solvent was concentrated and TPE-NCS (yellow oil, 1.3 g, 67%) was afforded by silica gel chromatography (petroleum ether: ethyl acetate = 20: 1). $^1H$ NMR (400 MHz, DMSO-$d_6$) δ 7.21 – 7.07 (m, 11H), 7.04 – 6.89 (m, 8H). $^{13}C$ NMR (101 MHz, DMSO-$d_6$) δ 143.44, 143.26, 143.21, 143.05, 142.10, 139.74, 134.01, 133.82, 133.62, 132.55, 131.13, 131.11, 131.04, 129.44, 129.39, 129.26, 129.19, 128.47, 128.43, 128.29, 127.34, 127.26, 127.20, 125.90, 40.65, 40.44, 40.23, 40.02, 39.81, 39.60, 39.39. HRMS (ESI, $CH_3CN$, m/z, relative intensity): 389.1235 (calcd for $[C_{27}H_{19}NS]$ +: 389.1238).

## Synthesis of water-soluble TPE-NCS (STPE-NCS)

We synthesized STPE-NCS according to the existing route with slight changes[60]. TPE-NCS (500 mg, 1.28 mmol) was added to conc. $H_2SO_4$ (10 mL) in a 75 mL pressure flask and the mixture was stirred and heated at 120 °C for 4 h. The mixture was then slowly added to ethyl acetate at 0 °C and placed in -20 °C refrigerator for 4 h. The precipitate was filtered and dissolved in water, which was neutralized with aqueous NaOH. The solution was then added to acetone, and the precipitate was filtered and washed with acetone three times to afford STPE (light yellow solid, 310 mg, 35%). $^1H$ NMR (400 MHz, $D_2O$) δ 7.55–7.43 (m, 6H), 7.19 (dq, $J$ = 8.5, 2.0 Hz, 4H), 7.16–7.10 (m, 2H), 6.86–6.78 (m, 2H), 6.57–6.49 (m, 2H). $^{13}C$ NMR (101 MHz, $D_2O$) δ 146.64, 146.46, 146.19, 145.65, 142.36, 140.67, 140.47, 140.40, 138.44, 133.21, 132.43, 131.71, 131.69, 131.60, 125.06, 125.02, 124.95, 115.48. HRMS (ESI, $CH_3OH$, m/z, relative intensity): 622.9462 (calcd for $[C_{27}H_{16}NO_9S_4 - 3H]$ -: 622.9468).

## Synthesis of amine-labeled polyP-NH₂

We synthesized polyP-NH₂ according to Wender's route with slight changes[52]. PolyP 45 was dissolved in 1 M MES buffer to a final concentration of 50 mM. 2,2′-(Ethylenedioxy)bis(ethylamine) (20 equiv.) was added to the polyP solutions followed by 1-ethyl-3-(3-dimethylaminopropyl) carbodiimide hydrochloride (EDAC) (20 equiv.) and the pH was adjusted to 6.5 with 1 M HCl. The mixture was heated to 60 °C for 2 h, after which time the pH of the reaction was increased to 10 with 1 M NaOH and the crude product was purified by precipitation into 45 ml ethanol (3x). The precipitate was collected by centrifugation (6 min at 1100 × g) and lyophilized to dryness. $^1H$ NMR (400 MHz, Deuterium Oxide) δ 3.77–3.54 (m, 16H), 3.20–3.10 (m, 4H), 3.09–3.00 (m, 4H). $^{13}C$ NMR (101 MHz, $D_2O$) δ 69.57, 66.50, 39.00. $^{31}P$ NMR (162 MHz, $D_2O$) δ 2.67, −0.69, −0.78, −5.78, −5.90, −20.88, −21.49, −21.51, −21.51, −21.72, −21.98, −22.41, −22.99, −23.73, −23.75, −23.95.

## Preparation of STPE-labeled polyP (STPE-polyP)

PolyP-NH₂ was dissolved in 0.1 M carbonate buffer (pH 9.0) to a final concentration of 230 mM. STPE (4 equiv) was added and the reaction was stirred for 20 h at room temperature. The crude product was purified by precipitation into 45 ml of ethanol (3x), collected by centrifugation (15 min at 1100 × g), dissolved in alkaline water (pH -8.0), and then further purified by preparative size exclusion chromatography (Sephadex 25). The samples were lyophilized for 2 days. $^1H$ NMR (400 MHz, $D_2O$) δ 7.60–7.43 (m, 6H), 7.28–7.12 (m, 6H), 6.85 (d, $J$ = 8.5 Hz, 2H), 6.55 (d, $J$ = 8.5 Hz, 2H), 3.77–3.51 (m, 35H), 3.19–3.00 (m, 16H). $^{13}C$ NMR (101 MHz, $D_2O$) δ 146.55, 146.38, 146.13, 142.25, 140.67, 140.49, 140.42, 138.62, 133.91, 132.43, 131.66, 131.58, 125.05, 125.02, 124.96, 116.00, 69.57, 66.50, 39.00. $^{31}P$ NMR (162 MHz, $D_2O$) δ −0.71, −5.58, −21.50, −21.54, −21.71, −21.94, −22.42, −23.02, −23.75, −23.98.

## Preparation of STPE-PMNSs

The STPE-PMNSs were constructed with a hierarchical assembly strategy with the assistance of CTAB and sodium oleate at room temperature. A solution of STPE-polyP (0.032 mmol, 220 mg), and CTAB (0.75 mmol, 273.33 mg) in $ddH_2O$ (15 ml, pH 7.4) was prepared. Then, the system was ultrasonically dispersed for 30 min. A solution of $MnCl_2$ (0.735 mmol, 91.9 mg) and sodium oleate (0.8 mmol, 243.2 mg) in $ddH_2O$ (15 ml, pH 6.5) was stirred for 30 min at room temperature. Briefly, the STPE-polyP/CATB solution was added to aqueous $Mn^{2+}$-oleate aqueous. The obtained mixed solution was stirred for 2 h, and the resulting brown product was obtained by centrifugation and washed with ethanol and $ddH_2O$. The samples were lyophilized for 2 days.

## Characterization of nanomaterials

TEM and AFM images of STPE-PMNSs were obtained from Hitachi HT7700 and Bruker MultiMode 8, respectively. STEM-HAADF images were obtained from Themis Z (Thermo Scientific). Zeta potential and size of STPE-PMNSs were measured on Nano-ZS instrument (Malvern Instruments Limited) and Brookhaven BI-200SM, respectively. FT-IR was obtained from Nicolet 7000-c. The fluorescence emission spectra were measured on a fluorescence spectrophotometer (Shimadzu, RF-6000). XRD patterns were recorded on an X-ray diffractometer (Bruker D8 Advance) with Cu Kα radiation ($\lambda$ = 1.54060 Å) and XPS was measured on Thermo ESCALAB 250Xi spectroscope. HRTEM and EDXS were performed on a JEOL JEM-F200. $^1H$ NMR, $^{13}C$ NMR, and $^{31}P$ NMR spectra were obtained on Bruker nuclear resonance (400 MHz) spectrometer. HRMS data was collected on Thermo Scientific Xcalibur.

## Fluorescence emission spectrum test

For STPE-polyP, 250 mg of STPE-polyP was dissolved in 5 ml of distilled water and set aside. Prepare 11 Eppendorf tubes and sequentially add 500, 450, 400, 350, 300, 250, 200, 150, 100, 50, and 0 µl of absolute

ethanol. Then, 50, 100, 150, 200, 250, 300, 350, 400, 450, 500, and 550 μl of STPE-polyP aqueous solution was added. After fully shaking the mixtures, the fluorescence emission spectrum (Ex = 330 nm) was measured. All experiments were independently carried out with three replicates.

For STPE-PMNSs, STPE-PMNSs solutions with concentrations of 100, 90, 80, 70, 60, 50, 40, 30, 20, and 10 μg ml$^{-1}$ were configured, and the fluorescence emission spectra (Ex = 330 nm) were tested after mixing. All experiments were independently carried out with three replicates.

### GS-like activity assessment

To assess the activity of the natural GS, we initially prepared a reaction stock solution containing 100 mM imidazole-HCl, 50 mM Glu, 25 mM 2-mercaptoethanol, 20 mM ATP, 20 mM MgCl$_2$, and 125 mM NH$_4$Cl. Subsequently, a quenching agent containing 0.37 M FeCl$_3$, 0.67 M HCl, and 0.2 M trichloroacetic acid was also prepared. The enzymatic reaction was initiated by adding 50 μg of natural GS to 500 μl of the stock solution. The reaction mixture was then incubated at 37 °C for 15 min, followed by the addition of 750 μl quenching agents. The absorbance of the reaction mixture was measured at 540 nm. The GS-like activity of STPE-PMNSs was assessed in a HEPES buffer solution (100 mM, pH 7.3). A reaction mixture was prepared containing 20 mM Glu, 20 mM ATP, and 20 mM NH$_4^+$ ions in the HEPES buffer. The enzymatic reaction was initiated by adding 50 mg of STPE-PMNSs to 500 ml of the reaction mixture. The reaction mixture was then incubated at 37 °C for 15 min, followed by the addition of 750 μl quenching agents. The absorbance of the reaction mixture was measured at 540 nm. Evaluation of Gln content in the reaction system by the standard curve method. All experiments were independently carried out with three replicates.

### Dose-response assessment

The dose-response of Glu to Gln converted by STPE-PMNSs was determined using a colorimetric GS assay kit (Sangon Biotech) according to the manufacturer's instructions. In brief, aliquots of 320 μl reactants and 140 μl substrates were mixed with STPE-PMNSs (0–400 μg ml$^{-1}$). In total, 150 μl was extracted in 96-well plates and incubated for 30 min at 37 °C, and added 50 μl 0.37 M FeCl$_3$, 0.67 M HCl, 0.2 M trichloroacetic acid. The absorbance of the colorimetric substrates was recorded by a microplate reader at 540 nm. Blank controls were also included at each concentration of STPE-PMNSs. The relative enzymatic activities of STPE-PMNSs were calculated as follows:

$$\text{Relative enzymatic activity (\%)} = \frac{OD_x - OD_{BL}}{OD_m - OD_{BL}} \times 100\% \qquad (2)$$

where OD$_x$ is the absorbance of the tested sample; OD$_m$ is the highest absorbance of STPE-PMNS under the optimal conditions; and OD$_{BL}$ is the absorbance of blank controls. All experiments were independently carried out with three replicates.

### SOD-like/GPx-like/CAT-like/LDH-like/COX-like activity assessment

The STPE-PMNSs (100 μg ml$^{-1}$) were in the recommended buffers of each assay kit including SOD, GPx, CAT, LDH, and COX assay kits. Experiments were according to the corresponding manufacturers' protocol. All experiments were independently carried out with three replicates.

### Ability to reuse STPE-PMNSs

At the initial reaction cycle, 100 μg ml$^{-1}$ STPE-PMNSs or GS was added into a 1 ml aqueous solution. After 12 h of reaction at 4 °C, the STPE-PMNSs were separated by an ultrafiltration membrane (MWCO at

3000 Da, UFC500396, Millipore) at 4000 × g and 4 °C for 20 min. The pellets of STPE-PMNSs or GS above the membrane were resuspended in an equal volume of solutions for the second reaction cycle. The supernatants collected in each reaction cycle were added 500 μl of 0.37 M FeCl$_3$, 0.67 M HCl, 0.2 M trichloroacetic acid, and then added to 96-well plates (200 μl/well) to detect the absorbance at 540 nm. The suspensions containing 100 μg ml$^{-1}$ STPE-PMNSs or GS were subjected to the same procedures and included as blanks. Six reaction cycles were performed. The percentages of remaining activities of STPE-PMNSs or GS in each reaction cycle were calculated by the following formula:

$$\text{Activity (\%)} = \frac{OD_{Rn} - OD_{BL}}{OD_{R1} - OD_{BL}} \times 100\% \qquad (3)$$

where OD$_{Rn}$ and OD$_{R1}$ represent the absorbance of substrates in STPE or GS mediated reactions in the $n$th and first cycles, respectively. OD$_{BL}$ is the absorbance of blank solutions at 540 nm. All experiments were independently carried out with three replicates.

### Impacts of temperature, pH, and organic solvent on the activity of STPE-PMNSs

Aliquots of STPE-PMNSs (100 μg ml$^{-1}$) and GS (100 μg ml$^{-1}$) were pre-treated in ddH$_2$O at 20–60 °C, 10 mM phosphate buffers at pH 3–11 or different ratios of DMF, DMSO, MeOH, MeCN solutions (0%, 10%, 20%, 30%, 40%, 50%, 60%, 70%, 80%) for 2 h. Then, the treated GS and STPE-PMNSs samples were collected by ultrafiltration (3000 Da) and resuspended in solutions containing Glu and NH$_4^+$ for 2 h reactions. After that, the absorbance in 540 nm at different conditions was examined. All experiments were independently carried out with three replicates.

### Calculation of the apparent Michaelis-Menten constant of STPE-PMNSs and natural GS

The measurements of the apparent $Km$ of STPE-PMNSs were performed at 37 °C in HEPES buffer (100 mM, pH 7.3). To measure the apparent $Km$ of STPE-PMNSs for Glu, we performed the reactions at growing concentrations of Glu, from 0 to 20 mM, while keeping the STPE-PMNSs, NH$_4^+$ and ATP concentration constant (100 μg ml$^{-1}$, 20 mM and 20 mM, respectively). An equivalent experiment was carried out to determine the apparent $Km$ for NH$_4^+$ and ATP. For that, we varied the NH$_4^+$ or ATP concentration from 0 to 20 mM while holding the STPE-PMNSs, Glu, and ATP or NH$_4^+$ concentrations to 100 μg ml$^{-1}$, 20 mM, and 20 mM, respectively.

For natural GS, the apparent $Km$ was determined in reaction mixtures consisting of 100 mM imidazole-HCl, 40 mM Glu, 25 mM 2-mercaptoethanol, 20 mM ATP, 20 mM MgCl$_2$, and 40 mM NH$_4$Cl. Stock solutions of imidazole, Glu, and ATP were titrated to pH 7.2. All reagents were stored on ice before assaying. When Glu was varied, NH$_4^+$ and GS were held constant at 40 mM and 100 μg ml$^{-1}$, respectively. An equivalent experiment was carried out to determine the apparent $Km$ for NH$_4^+$ and ATP. For that, we varied the NH$_4^+$ or ATP concentration from 0 to 40 mM while holding the STPE-PMNSs, Glu, and ATP or NH$_4^+$ concentrations to 100 μg ml$^{-1}$, 40 mM, and 40 mM, respectively. In both cases, enzyme-like first-order kinetic behavior was observed.

### $^1$H NMR analysis

STPE-PMNSs (1 mg) were added to 1 ml of D$_2$O containing 40 mM NH$_4^+$ and 40 mM Glu. After a certain time of reaction, the STPE-PMNSs were separated by an ultrafiltration membrane (MWCO at 3000 Da, UFC500396, Millipore) at 4000 × g 4 °C for 20 min. Afterward, 500 μl of supernatant was removed to measure $^1$H NMR. Experiments were repeated three times with similar results.

## Computation method

In the calculation model without ATP, DFT calculations were constructed and implemented in the Vienna ab initio simulation package (VASP. 6.3.0). Using the electron exchange and correlation energy treated within the generalized gradient approximation in the Perdew–Burke–Ernzerhof functional (GGA-PBE) the calculations were performed with a plane-wave basis set defined by a kinetic energy cutoff of 450 eV. The long-range dispersion interactions between adsorbates and surface were treated by applying the DFT-D3 method developed by Grimme et al. The k-point sampling was obtained from the Monkhorst–Pack scheme with a $(3 \times 3 \times 1)$ mesh for optimization and electronic structure. The geometry optimization and energy calculation were finished when the electronic self-consistent iteration and force reached $10-5$ eV and 0.02 eV Å$^{-1}$, respectively.

For the calculation model in the presence of ATP, the geometry optimizations of the cluster models were performed at r2SCAN-3c level using ORCA 5.0.3 considering the balance between the computational load and the accuracy. The performance of the currently chosen method is verified to be suitable for the adsorption on polar salt and non-polar coinage-metal surfaces with considerably lower cost. The Conductor-like Polarizable Continuum Model (CPCM) was used to describe the solvent effect with taking water as solvent. The edged atoms were freeze during the geometric relaxation. Single point energy calculations were done at PBE/def2-SVP + CPCM (H$_2$O) level with Gaussian 16 The energetic difference ($\Delta E$) is calculated by: $\Delta E = E^{1+ads} - E^1 - E^{ads}$, where $E^{1+ads}$ refer to the total energy after adsorbs and $E^1$ and $E^{ads}$ refers to the energy of $\mathbf{1}$ and adsorbents, respectively.

## Competitive experiments of Glu activation by ATP and polyP

STPE-PMNSs (100 μg ml$^{-1}$) were added to in HEPES buffer (100 mM, pH 7.3). We performed the reactions at growing concentrations of ATP, from 0 to 20 mM, while keeping the STPE-PMNSs, NH$_4^+$ and Glu concentration constant (100 μg ml$^{-1}$ 20 mM and 20 mM, respectively). After reaction 2 h, we measured changes in fluorescence emission at 467 nm. All experiments were independently carried out with three replicates.

## Cell viability assay

SH-SY5Y cells were seeded in 96-well plates at a density of $1 \times 10^4$ cells/well. After 24 h of incubation, 100 μl culture medium containing STPE-PMNSs were added to each well. Following another 24-h incubation, 10 μl of CCK8 solution was added to each well. After incubating for 4 h, the absorbance at 450 nm was measured. All experiments were independently carried out with three replicates.

## In vitro cellular uptake of STPE-PMNSs

We conducted a quantitative analysis of the internalization of STPE-PMNSs by cells over a defined time interval. SH-SY5Y cells ($1.2 \times 10^6$ cells ml$^{-1}$) were seeded in 6-well tissue culture plates and grown to ~90% confluence. Growth media in plates was replaced with media containing PMNSs. Cells were incubated with nanomaterials for 0, 3 h, 6 h, 12 h, 18 h, and 24 h at 37 °C. Finally, the medium was removed from the wells, and the cells were washed three times with PBS to clear away nanomaterials outside of the cells. Next, 200 μl of 1% Triton X-100 in 0.1 M NaOH solution was added to lyse the cells. After treatment, Mn content in the lysates was determined using ICP–OES. All experiments were independently carried out with three replicates.

## ICP-OES/MS measurement

Transfer 50 μl of prepped cell lysate into a 1.5 ml Eppendorf tube, followed by the addition of 100 μl of concentrated nitric acid for digestion at 95 °C for 1 h. After cooling, introduce 50 μl of H$_2$O$_2$ and maintain at 95 °C for 30 min. Subsequently, adjust the volume to 1 ml with ddH$_2$O and measure of Mn content on an inductively coupled plasma optical emission spectrometry/mass spectrometer.

## Intracellular imaging of STPE-PMNSs

For CLSM, SH-SY5Y cells were first seeded onto 2-cm culture dishes at a density of $8.0 \times 10^4$ cells ml$^{-1}$ for 24 h. Then the culture medium was changed with as-prepared DMEM containing STPE-PMNSs (25 μg ml$^{-1}$). The cells were incubated with nanomaterials for 6 h at 37 °C, and then the medium was removed from the dish. After rinsing with PBS three times to remove residual nanomaterials, the cells were incubated with 1 mM 4-AP (induces cells to produce Glu) for 4 h at 37 °C. Subsequently, the culture medium was discarded, and after washing 3 times with PBS, the cells were monitored using a confocal laser scanning microscope with a living cell CO$_2$ culture system. (CLSM, OLYMPUS SpinSR). All experiments were independently carried out with three replicates.

## Apoptosis analysis

SH-SY5Y cells were seeded in 6-well tissue culture plates at a density of $1.5 \times 10^5$ cells ml$^{-1}$ and set up in CK, 4-AP, STPE-PMNSs, and STPE-PMNSs + 4-AP groups, respectively, and after 24 h of incubation, the cells were treated in the STPE-PMNSs group and STPE-PMNSs + 4-AP group with 25 μg ml$^{-1}$ of STPE-PMNSs for 24 h, after which the cells were washed three times with PBS, and 4-AP was added to the STPE-PMNSs + 4-AP group for 4 h. The cells were collected, washed, and incubated with *Annexin V-FITC* and *PI* according to the instructions of the *Annexin V-FITC* Apoptosis Detection Kit (KeyGEN). Then, the cells were further washed and analyzed by flow cytometry (BD FACSCalibur). All experiments were independently carried out with three replicates.

## Glu and Gln levels

SH-SY5Y cells were seeded in 6-well tissue culture plates at a density of $1.5 \times 10^5$ cells ml$^{-1}$ and set up in CK, 4-AP, STPE-PMNSs, and STPE-PMNSs + 4-AP groups, respectively, and after 24 h of incubation, the cells were treated in the STPE-PMNSs group and STPE-PMNSs + 4-AP group with 25 μg ml$^{-1}$ of STPE-PMNSs for 24 h, after which the cells were washed three times with PBS, and 4-AP was added to the STPE-PMNSs + 4-AP group for 4 h. The cells were collected, and then the cell lysates were obtained with a Whole Cell Lysis Assay (KeyGEN BioTECH Co., Ltd. Nanjing, China). The Glu and Gln contents were determined using the corresponding content assay kit. Both of their contents were determined by comparing the absorbance value with the calibration plot for standard solutions. The absorbance values were measured at 340 nm and 450 nm. All experiments were independently carried out with three replicates.

## Western blot analysis

SH-SY5Y cells were seeded in 6-well tissue culture plates at a density of $1.5 \times 10^5$ cells ml$^{-1}$ and set up in CK, 4-AP, STPE-PMNSs, and STPE-PMNSs + 4-AP groups, respectively, and after 24 h of incubation, the cells were treated in the STPE-PMNSs group and STPE-PMNSs + 4-AP group with 25 μg ml$^{-1}$ of STPE-PMNSs for 24 h, after which the cells were washed three times with PBS, and 4-AP was added to the STPE-PMNSs + 4-AP group for 4 h. The cells were collected, and then the cell lysates were obtained with a Whole Cell Lysis Assay. The protein concentration was determined by the Super-Bradford Protein Assay Kit (CWBiotech, Inc., Beijing, China). Briefly, the extracts were first separated by 10% SDS–PAGE and transferred to a polyvinylidene difluoride membrane (Bio-Rad, CA, USA). The membrane was blocked with 5% BSA in TBST at 25 °C for 1 h and then incubated with antibodies at 4 °C overnight. The expression of α-actin was used as the internal standard. Primary antibodies against the following proteins were used: α-actin (1:1000 for WB), Bcl-2 (1:1000 for WB), and Bak (1:1000 for WB). The appropriate secondary antibodies (1:1000 for WB) were purchased from CST. Details are shown in the Source Data. All experiments were independently carried out with three replicates.

## Ca²⁺ content measurement

SH-SY5Y cells were seeded in 6-well tissue culture plates at a density of $1.5 \times 10^5$ cells ml$^{-1}$ and set up in CK, 4-AP, STPE-PMNSs, and STPE-PMNSs + 4-AP groups, respectively, and after 24 h of incubation, the cells were treated in the STPE-PMNSs group and STPE-PMNSs + 4-AP group with 25 µg ml$^{-1}$ of STPE-PMNSs for 24 h, after which the cells were washed three times with PBS, and 4-AP was added to the STPE-PMNSs + 4-AP group for 4 h. The cytosolic Ca$^{2+}$ concentration was determined with Fluo-4 AM (Beyotime Biotechnology Co., Ltd, Shanghai, China) according to the manufacturer's procedure. The Ca$^{2+}$ signals were analyzed by flow cytometry. All experiments were independently carried out with three replicates.

## Statistics and reproducibility

The Investigators were not blinded to allocation during experiments and outcome assessment. All multiple comparisons have been corrected before adjusted $p$ values are presented. The tests used are indicated in figure legends.

## Reporting summary

Further information on research design is available in the Nature Portfolio Reporting Summary linked to this article.

## Data availability

The data that support the findings of this study are available from the corresponding author upon request. The data generated in this study are provided in the Supplementary Information/Source Data file. Source data are provided with this paper.

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

## Acknowledgements

Financial support was provided by the National Nature Science Foundation of China (22025701 to J.Z.; 22177048 to W.W.; 22207053 to X.X.W. and 22293052 to J.Z.), Natural Science Foundation of Jiangsu Province (BK20232020 to J.Z., BK20220764 to X.X.W.), the Fundamental Research Funds for the Central Universities, and Shenzhen Bay Laboratory Open Fund Project (SZBL2021080601013 to J.Z., W.W., and K.Y.).

## Author contributions

Jing Wang, Xinyang Zhao, and Yucheng Tao conceived and designed the experiments, interpreted the data, and wrote the manuscript. Xiuxiu Wang, Li Yan and Yi Hsu supported the experiments for measuring the GS-like activity. Kuang Yu interpreted the DFT data and discussed the research procedures with Jing Zhao and Yong Huang. Yuncong Chen guided the synthesis of AIE groups. Wei Wei provided experimental ideas and revised the manuscript.

## Competing interests

The authors declare no competing interests.

## Additional information

[1]State Key Laboratory of Coordination Chemistry, Chemistry and Biomedicine Innovation Center (ChemBIC), School of Chemistry and Chemical Engineering, Nanjing University, Nanjing 210093, PR China. [2]School of Life Sciences, Nanjing University, Nanjing 210093, PR China. [3]Nanchuang (Jiangsu) Institute of Chemistry and Health, Sino-Danish Ecolife Science Industrial Incubator, Jiangbei New Area, Nanjing 210000, PR China. [4]Tsinghua-Berkeley Shenzhen Institute and Institute of Materials Research (iMR), Tsinghua Shenzhen International Graduate School, Tsinghua University, Shenzhen, Guangdong, PR China. [5]Taipei Wego Private Senior High School, Taipei, TWN, PR China. [6]Shenzhen Research Institute, Nanjing University, Shenzhen, PR China. [7]Department of Chemistry, The Hong Kong University of Science and Technology, Clear Water Bay, Kowloon, Hong Kong SAR, PR China. ✉e-mail: chenyc@nju.edu.cn; Jingzhao@nju.edu.cn; yonghuang@ust.hk; weiwei@nju.edu.cn

