## [Peer Review File · Nature Communications]

Reviewers' Comments:

Reviewer #1:

Remarks to the Author:

In this manuscript, the authors synthesized the first artificial GS, AIE-activated STPE-PMNSs nanosheets, and further applied them into biomedicine field to relieve Glu-induced neurotoxicity. DFT calculations were also performed to present the energy change during the reaction for the conversion of Glu to Gln. To some extent, this work has some implications for the development of more artificial GS. However, compared with the catalytic mechanism of natural GS, the catalytic mechanism proposed in this work seems not very reasonable, and more in-depth research is needed on the intrinsic catalytic mechanism. Substantial revision of this manuscript is recommended.

- (1) The results presented in supplementary Fig. 13a and 13b show that the STPE-PMNSs synthesized in this work are amorphous. What is the basis for the chosen of the crystal structure model in the computational part in this work? At the same time, please provide clearer crystal structure along the energy profile (Fig. 4b), crystal configurations in cif format for all intermediates.
- (2) The STPE-PMNSs nanosheets agglomerated after the reaction. However, no experimental results can clearly support that the physicochemical properties of the materials did not change after the reaction.
- (3) The conversion of glutamic acid to glutamine via the catalytic activity of natural GS is dependent on ATP, which is used to activate Glu. Surprising, the STPE-PMNSs synthesized in this work catalyzes similar reactions is ATP-free. However, poly P in STPE-PMNSs contains phosphate similar to ATP. Is it possible that the N-P bond or P-O bond in STPE-PMNSs was acted as the same function as natural GS to activate Glu, which is led to the aggregation of STPE-PMNSs after reaction? The above possibilities can be ruled out by experimental analysis, such as whether the material structure changes after the catalytic reaction with and without ATP.
- (4) From the perspective of reaction thermodynamics and kinetic, we generally believe that one reaction is not thermodynamically feasible when the reaction energy barrier is higher than 1.5 eV. However, the energy curve presented in Fig.4b shows that the reaction energy of rate-determining step reaches up to 2.01 eV. Based on the results, it can be deduced that the reaction energy barrier should be higher than the reaction energy, and much higher than the natural GS's 0.98 eV.
- (5) Energy changes of a few speculated intermediates along the reaction profile are not enough to reveal the catalytic reaction mechanism of the conversion of Glu to Gln. In addition, the total catalytic reaction energy of 1.99 eV illustrates that the reaction is thermodynamically infeasible. Some of the more critical intermediate structures should be provided, such as how NH_4^+ is activated and how the OH adsorbate on the material surface resulting from the decomposition of Glu is subsequently consumed in order to complete the reaction cycle.

Reviewer #2:

Remarks to the Author:

This paper summarizes the research related to the development of manganese nanosheets enabled with polyP tetraphenylethylene analogs with size-tunable, uniform, and stable fluorescent properties, through a hierarchical assembly strategy. The synthesized nanosheets served as artificial glutamine synthetase with an ability to convert glutamate to glutamine for biomedical applications. Their catalytic properties are tested in human neuroblastoma cells with a high efficacy in conversion of glutamate to glutamine alleviating neurotoxicity. This is an interesting and important area of research. The paper is generally well written; however, the quality of the paper can be enhanced if the following points can be addressed.

1. There are a few grammatical errors in the manuscript which need to be rectified. For example; Abstract: "Glutamine synthase (GS) plays crucial roles". "Extensively explored for chemical engineering and biomedical".

2. In Line 202, the authors have mentioned that "Figure 4i revealed that polyP-Ca and MnO₂ could not catalyze the reaction of Glu to Gln, indicating the combination of polyP and Mn²⁺ is necessary for the catalytic effects". The authors need to discuss why this combination can have the catalytic properties and why polyP-Ca and MnO₂ do not possess any catalytic properties as individual entities.

3. MnO₂ is a commonly used catalyst. Among all the transition metal oxides, it is known to exhibit stable performance in applications such as supercritical water oxidation, is known to have a relatively high catalytic activity in the catalytic decomposition of organic compounds by oxidation and also inside the cells and animals. Why did the authors not measure or compare the intracellular catalytic activity of MnO₂ alone with the nanosheets that the authors synthesized in Figure 5 experiment?

4. In Figure 5 (a) The authors mentioned that the nanosheets have a high biocompatibility 24h post exposure but in the figure only 12h data is presented. Please include the 24h data in the main file instead of supporting information.

5. Did the authors evaluate the dose response for the synthesized nanosheets for Glu to Gln conversion?

6. In the discussion, the authors need to discuss more about the impact of Glu cytotoxicity and its impact on nerve cells and the prospects in which the proposed nanomaterials can be applied in the real-world application.

Reviewer #3:

Remarks to the Author:

In the manuscript entitled AIE-Active Polyphosphate-Manganese Nanosheets as Artificial Glutamine Synthase in Living Cells the authors obtain Mn²⁺ AIE-active STPE-PMNSs that seem to catalyze the conversion of Glu to Gln and exhibited GS-like activity also in living cells.

The finding is surprising and very interesting. However several critical points require further work to confirm the authors' claim and more experimental details need to be included.

Here are some comments:

STPE-PMNSs possess comparable enzymatic efficiency and similar catalytic reaction kinetics to that of natural GS. How the enzymatic-like activity is evaluated cannot be inferred from the Methods and should be described in details. For the cellular assay, it looks like the particles cross freely the cellular membrane. However, an evaluation of the amount intracellularly delivered should be reported.

-line 159: "V is the reaction rate", change with V is the initial reaction rate

-line 160-162: The apparent Michaelis-Menten constants of Glu and NH₄⁺ substrates are 130.67 μM and 156.65 μM, respectively, revealing enzyme-like first-order kinetic behavior (Figs 3c-d and Supplementary Fig. 30)_

It is not clear from the Methods the GS they are referring to for the K_m values. For human GS K_m should be around 2-3mM and for other species that value should not differ much from that value. So it is not clear where the reported values for K_m come from.

-The sentence "revealing enzyme-like first-order kinetic behavior" is conceptually wrong. You cannot infer from K_m that the kinetic is first-order. You evaluate if the kinetic is first order, then K_m are calculated.

-The protective effect of STPE-PMNSs was tested intracellularly after using a molecule that raises intracellular Glu levels. It would be interesting to test Glu excitotoxicity in neuronal cells, by evaluating the Glu scavenging ability of STPE-PMNSs extracellularly.

-Supplementary Fig. 22 . The authors report the % Glu and Gln with time. It looks like a time course, but I doubt that the trend is linear in such a large time range (100 min).

-Supplementary Fig. 22. Again, the activity of "natural GS" is reported. Are they referring to the human one? The activity is evaluated for 60 min, which is a very large time range. The activity is usually evaluated from 2 to 15 minutes (doi: 10.1042/bj3280159).

-Supplementary table 1 and 2 report the reaction rate. The unit of measure are not correct in my opinion. The enzymatic activity indicates the amount of substrate converted in the time unit in the reaction volume (for instance: micromol/min/mL). If they know the used amount of these particles, it should be even better to normalize for the amount/volume of particles to obtain micromole/min/mg of particles.

-Supplementary Fig. 30. The unit is not correct, it should be micromol/min

Reviewer #4:

Remarks to the Author:

The manuscript describes polyphosphate-Mn nanosheets (PMNSs) as biomimetic catalyst with glutamine synthetase activity, a novel property for this nanomaterial and a promising alternative to natural glutamine synthetase enzyme. Further, nanosheets are functionalized with AIE-active groups (STPE), which enable visualization of nanosheets and fluorescence monitoring of aggregation status. The authors provided rigorous characterization of nanosheets and glutamine synthetase-like activity in vitro, performed computational analysis to elucidate the mechanism of the catalytic activity, and demonstrated biocompatibility and functionality of STPE-PMNSs in live cells. The manuscript presents the work of potential broad interest spanning fields of synthetic and computational chemistry, bio-nanotechnology, and molecular and cell biology. However, the manuscript lacks clarity on novelty and impact of the reported developments / findings, which obscures the key scientific contributions of the manuscript and leads to the following requests:

1) There is no support for the claim that this work "provides opportunities for applying multifunctional polymers to biomedicine" and other similarly broad statements. The manuscript would be strengthened by, instead, clearly describing the novelty and impact of the specific developments / findings presented. In general, it would help for the claims to be strictly limited to the actual material properties, phenomena observed, and applications demonstrated.

2) The AIE element does not appear to be essential for nanosheet assembly or glutamine synthetase-like activity; hence, addition of STPE to nanosheets is puzzling. The rationale for including STPE needs to be discussed, particularly in reference to increased material complexity and potential alterations to material properties. The role of AIEgen as an aggregation sensor might need to be more clearly outlined and linked to its uses within the STPE-PMNS structure.

3) Description of the computational work and the proposed mechanism of catalytic activity lacks details. The calculation results and interpretation need to be presented, and the logic used to determine steps and sequence of events needs to be outlined. It is unclear what drives the endothermic reaction (and whether there is any experimental evidence for that), where the energy for the reaction comes from, and why the system would not remain in the lowest energy state. Further, the model does not explain nanosheet aggregation during catalysis (observed experimentally) or the role of the unique combination of polyphosphate with Mn²⁺ in nanosheets. Thus, computational component needs a more thorough analysis and a better integration into the overall flow of the manuscript.

4) Aggregation of nanosheets during catalytic activity, accumulation in cells, and long-term toxicity need to be further assessed, particularly since live cell application is presented as the main objective of the manuscript.

Minor concerns and comments:

- 1) Term "glutamine synthetase" (not "synthase") should be consistently used throughout the manuscript.
- 2) Abbreviation "AIE" needs to be spelled-out in the Abstract
- 3) The relevance of emphasizing ref. 14 and 15 in the Introduction (lines 44-48) is unclear – this discussion should be directly linked to the topic of the manuscript (or removed).
- 4) Abbreviation "RIM" (line 68) should stand for "restriction of intramolecular motion"
- 5) Discussion in lines 59-71 of the Introduction should describe how AIE would achieve "visualization of biomimetic catalysis processes" (a stated objective for including AIEgens), building on the current understanding that stable fluorescent self-assembled nanomaterials can be prepared using AIEgen components.
- 6) Glutamine synthetase reuse experiment (Fig. 3e) needs a confirmation of the enzyme retention after ultrafiltration, as enzyme loss would also manifest in lower activity.
- 7) High-resolution microscopy images with cell cross-sections are needed to prove the intracellular localization of STPE-PMNSs. Images in Fig. 5a show localization patterns that could result from extracellular binding of nanosheets to cell membrane.
- 8) Further, it would be helpful to discuss the mechanism of PMNS intracellular uptake and distribution (e.g., lack of endosomal localization).

Dear Editor Sunjie Ye:

Thank you very much for your consideration of our manuscript entitled " **AIE-Active Polyphosphate-Manganese Nanosheets as Artificial Glutamine Synthase in Living Cells**" (Research Article, No. NCOMMS-23-19898) to *Nature Communications*. We appreciate your kind help with our submission.

These comments from the referees are all quite constructive and very helpful to improve our paper. We are deeply grateful for the insightful comments from the referees. After careful consideration and studies, we provide a point-by-point response letter to those reviewer comments, including a list of changes made to the manuscript, hoping to meet with their approval. Revised portions are marked in yellow highlights in the manuscript.

In addition to addressing the requested changes, we have corrected several errors that we had noticed. We hope that the revisions are satisfactory and the manuscript may be accepted for publication in *Nature Communications*.

If you have any questions about our revised manuscript, please don't hesitate to let me know.

Yours sincerely,

Dr. Jing Zhao

State Key Laboratory of Coordination Chemistry
Nanjing University, China

Reviewer 1

In this manuscript, the authors synthesized the first artificial GS, AIE-activated STPE-PMNSs nanosheets, and further applied them to the biomedicine field to relieve Glu-induced neurotoxicity. DFT calculations were also performed to present the energy change during the reaction for the conversion of Glu to Gln. To some extent, this work has some implications for the development of more artificial GS. However, compared with the catalytic mechanism of natural GS, the catalytic mechanism proposed in this work seems not very reasonable, and more in-depth research is needed on the intrinsic catalytic mechanism. Substantial revision of this manuscript is recommended.

Query 1

The results presented in supplementary Fig. 13a and 13b show that the STPE-PMNSs synthesized in this work are amorphous. What is the basis for the choice of the crystal structure model in the computational part of this work? At the same time, please provide clearer crystal structure along the energy profile (Fig. 4b), and crystal configurations in cif format for all intermediates.

Response

We thank the reviewer for this comment.

There are no diffraction peaks in the XRD pattern, which may be attributed to layers are not stacked into a periodic structure. The configuration in our computation is an idealized representation constructed based on experiment results. XPS analysis demonstrated a phosphorus-oxygen (P-O) to manganese-oxygen (Mn-O) bond ratio of approximately 1:1, and the predominant oxidation state of manganese (Mn) is Mn (II). The atomic ratio of phosphorus (P) to manganese (Mn) is also approximately 1:1 (Fig. 2g-i). Additionally, the presence of phosphate groups (PO_4^{3-}) was confirmed through FT-IR and Raman spectroscopy (Fig. 2c-d). TEM images revealed a layered structure in PMNS. Based on these findings, we construct a model for PMNSs. To evaluate the thermodynamic stability of the constructed PMNSs structure, ab-initio molecular dynamics (AIMD) simulations have been performed. The AIMD results indicate that the PMNSs catalyst can maintain its structural stability under 800K conditions (Supplementary Fig. 52). To obtain the actual configuration of PMNSs, we are currently

focused on cultivating PMNSs crystals. Nevertheless, the growth of polymer crystals is challenging and time-consuming, and we plan to further advance this aspect of our research in the future. Additionally, we have conducted a more comprehensive investigation into the catalytic mechanism of PMNSs and have included the CIF files of the intermediates in the supplementary materials.

Revisions Made

(Please refer to the manuscript on page 6, lines 10-20)

Distinct differences are distinguished in the FT-IR spectra between PMNSs and STPE-PMNSs. The peaks centered at 1088, 867, and 988 cm^{-1} belonged to asymmetric stretching of ν_{as} (P-O) (Fig. 2c). Raman analysis of the STPE-PMNSs exhibits three major bands located at around 717.0, 616.8, 566.2, and 455 cm^{-1} (Fig. 2d), which are related to Mn-O symmetric stretching. The peak centered at 944.0 cm^{-1} is attributed to intramolecular symmetric vibrations of PO_4^{3-} , while the weaker 891.9, 980.1, and 1023.7 cm^{-1} peaks relate to the bending and asymmetric stretching of PO_4^{3-} . The binding energy features located at 652.3 eV and 640.3 eV are attributed to the Mn $2p_{1/2}$ and Mn $2p_{3/2}$, respectively, which demonstrate that Mn predominantly exists in the Mn(II) state. Significantly, the O 1s XPS spectrum examination indicates an Mn-O to P-O bond ratio of approximately 1:1 (Figs. 2g-i).

Fig. 2 XPS analysis for O 1s spectra (g), Mn 2p spectra (h), and atomic percentage (i) of STPE-PMNSs.

Fig. 2 (c) Comparison of Fourier transforms infrared (FT-IR) spectroscopy assay between PMNSs and STPE-PMNSs. (d) Raman analysis of STPE-PMNSs.

(Please refer to the manuscript on page 14, line 12)

The AIMD results indicate that the PMNSs catalyst can maintain its structural stability under 800 K conditions (Supplementary Fig. 52).

Supplementary Fig. 52 The AIMD diagram of PMNSs catalyst.

Query 2

The STPE-PMNSs nanosheets agglomerated after the reaction. However, no experimental results can support that the physicochemical properties of the materials did not change after the reaction.

Response

We thank the reviewer for this comment.

We collected the nanosheets after 2 h catalysis and further assessed their

physicochemical properties. Firstly, TEM images revealed that despite the occurrence of agglomeration, the fundamental sheet structure remained intact (Supplementary Fig. 32). The EDX results showed that P, Mn, O, S, N, and C are distributed along the overall outline of the nanosheets (Supplementary Fig. 47c). XPS analysis indicated that the valence of various elements did not change significantly (Supplementary Fig. 33). Furthermore, the fluorescence emission peaks of the recycled nanosheets remained unchanged (Supplementary Fig. 30b), and the reusability of nanosheets indicated the aggregated nanosheets retained their GS-like activity (Supplementary Fig. 34). These results indicate that the physicochemical properties of the nanosheets remained unaffected by the catalytic process.

Revisions Made

(Please refer to the manuscript on page 9, lines 1-3)

Moreover, the physicochemical properties of the recovered nanosheets remained consistent before and after catalysis (Supplementary Fig. 32-34).

Supplementary Fig. 32 The TEM images of STPE-PMNSs before (a) and after (b) 2 h catalysis.

Supplementary Fig. 47 (c-d) STEM images and elemental mapping for STPE-PMNSs after catalysis. Scale bar: 30 nm.

Supplementary Fig. 33 XPS analysis of STPE-PMNSs after catalysis.

Supplementary Fig. 30 (b) The fluorescence emission spectrum of STPE-PMNSs before and after the catalytic process.

Supplementary Fig. 34 A comparative analysis of catalytic activities among varied concentrations of STPE-PMNSs and catalyzed STPE-PMNSs.

Query 3

The conversion of glutamic acid to glutamine via the catalytic activity of natural GS is dependent on ATP, which is used to activate Glu. Surprisingly, the STPE-PMNSs synthesized in this work catalyze similar reactions and are ATP-free. However, poly P in STPE-PMNSs contains phosphate similar to ATP. Is it possible that the N-P bond or P-O bond in STPE-PMNSs acted as the same function as natural GS to activate Glu, which led to the aggregation of STPE-PMNSs after the reaction? The above possibilities can be ruled out by experimental analysis, such as whether the material structure changes after the catalytic reaction with and without ATP.

Response

We thank the reviewer for this comment.

We performed catalytic experiments with and without ATP and collected nanosheets after catalysis for further analysis. TEM images revealed that in the absence of ATP, the

nanosheets exhibited significant aggregation after catalysis, whereas the nanosheets did not aggregate in the presence of ATP (Supplementary Fig. 46). EDX and XPS results indicated the structure of aggregated nanosheets retained unchanged, but ICP-OES analyses demonstrated a decrease in phosphorus (P) content within the catalyzed nanosheets, accompanied by an increase in free P level in the solution (Supplementary Figs. 47-51). The analysis of the catalytic mechanism (refer to Query 4 for specific details) indicated this decrease was attributed to the activation of Glu by the terminal phosphate group of the polyP within STPE-PMNSs. This activation is analogous to the role of ATP in the catalysis of natural GS, resulting in the detachment of the terminal phosphate group from the polyP, which may further lead to the aggregation of STPE-PMNSs after the reaction.

Revisions Made

(Please refer to the manuscript on page 13, lines 9-19)

Significantly, the enzymatic catalysis of natural GS necessitates the involvement of ATP while STPE-PMNSs exhibit ATP-independent catalytic activity (Fig. 4g). To investigate the underlying cause, we conducted a catalytic reaction in the presence or absence of ATP and subsequently collected the catalyzed nanosheets for further analysis. TEM images reveal that in the absence of ATP, the nanosheets exhibited significant aggregation after catalysis, while in the presence of ATP, the nanosheets did not aggregate (Supplementary Fig.46-47). ICP-OES and XPS analyses demonstrated a decrease in phosphorus (P) content within the catalyzed nanosheets, accompanied by an increase in free P level in the solution (Supplementary Fig.48-51). Coincidentally, the polyP presence within the nanosheets possesses high-energy phosphate bonds like ATP, leading us to hypothesize that they play the same role.

Supplementary Fig. 46 The TEM image of STPE-PMNPs in the presence (a) or absence (b) of ATP.

Supplementary Fig. 47 STEM images and elemental mapping for STPE-PMNPs in the presence (a) or absence (b) of ATP. Scale bar: 30 nm.

Supplementary Fig. 48 XPS analysis of STPE-PMNS in the presence (a) or absence (b) of ATP.

Supplementary Fig. 49 The atomic percentage from XPS of STPE-PMNS in the presence (a) or absence (b) of ATP.

Supplementary Fig. 50 The phosphorus (P) content of STPE-PMNS in the presence (a) or absence (b) of ATP.

Supplementary Fig. 51 The free phosphorus (P) level within the catalytic system solution in the absence of ATP.

Query 4

From the perspective of reaction thermodynamics and kinetics, we generally believe that one reaction is not thermodynamically feasible when the reaction energy barrier is higher than 1.5 eV. However, the energy curve presented in Fig.4b shows that the reaction energy of the rate-determining step reaches up to 2.01 eV. Based on the results, it can be deduced that the reaction energy barrier should be higher than the reaction energy, and much higher than the natural GS's 0.98 eV.

Response

We thank the reviewer for this comment.

We re-proposed a potential catalytic mechanism of STPE-PMNS in combination with DFT calculations. The catalytic reaction proceeds in two steps.

The first step is the activation of the reactants. Firstly, Glu is adsorbed on the surface of the nanosheet, which binds with the metal atom Mn and then attacks the terminal phosphate group of polyP to originate γ -glutamyl phosphate (Fig. 5a). This step is the activation process of Glu, which is an exergonic reaction with a reaction-free energy of -0.89 eV. The active site of NH_4^+ is the terminal phosphate of polyP within STPE-PMNS, a negatively charged region that rapidly binds NH_4^+ , where NH_4^+ loses its proton and becomes a basic residue ready for the nucleophilic attack on the C γ of γ -glutamyl phosphate intermediate (Fig. 5a). Activation of NH_4^+ can happen either before or after the formation of γ -glutamyl phosphate intermediate.

The second step corresponds to Gln formation, via a nucleophilic attack of ammonia

to the C γ of the γ -glutamyl phosphate intermediate. At this phase, ammonia starts the nucleophilic attack on the C γ of γ -glutamyl phosphate intermediate, with the concomitant bond breaking of the C γ -phosphate and bond formation of the C γ -ammonia bond (Fig. 5a). Subsequently, upon the formation of protonated glutamine, a rapid proton transfer occurs to the released phosphate group, resulting in the production of Gln. The energy profile is shown in Fig. 5b, and the total catalytic reaction energy is -1.53 eV. The results indicated that it is thermodynamically feasible. The catalytic mechanism is consistent with a mechanism catalyzed by natural enzymes.

Revisions Made

(Please refer to the manuscript on page 14 line 6 to page 15 line 14)

To confirm this, we proposed a potential catalytic mechanism of STPE-PMNS in combination with DFT calculations. The configuration in our computation is an idealized representation constructed based on experiment results. The AIMD results indicate that the PMNSs catalyst can maintain its structural stability under 800 K conditions (Supplementary Fig. 52). The catalytic reaction proceeds in two steps. The first step is the activation of the reactants. At the beginning, Glu is adsorbed on the surface of the nanosheet, which binds with the metal atom Mn and then attacks the terminal phosphate group of polyP to originate γ -glutamyl phosphate (Fig. 5a). This step is the activation process of Glu, which is an exergonic reaction with a reaction-free energy of -0.89 eV. The active site of NH $_4^+$ is the terminal phosphate of polyP within STPE-PMNS, a negatively charged region that rapidly binds NH $_4^+$, where NH $_4^+$ loses its proton and becomes a basic residue ready for the nucleophilic attack on the C γ of γ -glutamyl phosphate intermediate (Fig. 5a). Activation of NH $_4^+$ can happen either before or after the formation of γ -glutamyl phosphate intermediate.

The second step corresponds to Gln formation, via a nucleophilic attack of ammonia to the C γ of the γ -glutamyl phosphate intermediate. At this step, ammonia starts the nucleophilic attack on the C γ of γ -glutamyl phosphate intermediate, with the concomitant bond breaking of the C γ -phosphate and bond formation of the C γ -ammonia bond (Fig. 5a). Subsequently, upon the formation of protonated glutamine, a rapid proton transfer occurs to the released phosphate group, resulting in the production

of Glu. The energy profile is shown in Fig. 5b, and the total catalytic reaction energy is -1.53 eV. The catalytic mechanism is consistent with a mechanism catalyzed by natural enzymes⁶³⁻⁶⁷.

Fig. 5 Reaction mechanism and energies of STPE-PMNSs catalyzing the conversion of Glu to Gln. (a) Glu (left) and NH_4^+ (right) activation and Gln formation step (down). (b) Energy profile for STPE-PMNSs catalyzed conversion of Glu to Gln.

Query 5

Energy changes of a few speculated intermediates along the reaction profile are not enough to reveal the catalytic reaction mechanism of the conversion of Glu to Gln. In addition, the total catalytic reaction energy of 1.99 eV illustrates that the reaction is thermodynamically infeasible. Some of the more critical intermediate structures should be provided, such as how NH_4^+ is activated and how the OH adsorbate on the material surface resulting from the decomposition of Glu is subsequently consumed to complete the reaction cycle.

Response

We thank the reviewer for this comment.

We propose a new reaction mechanism as described in **query 4**. The total catalytic reaction energy is -1.53 eV, indicating that it is thermodynamically feasible. Furthermore, the active site of NH_4^+ is the terminal phosphate of polyP within STPE-PMNSs, a negatively charged region that rapidly binds to NH_4^+ , where NH_4^+ loses its proton and becomes a basic residue ready for the nucleophilic attack on the C_γ of γ -glutamyl phosphate intermediate (Fig. 5a).

Revisions Made

(Please refer to the manuscript on page 14 line 6 to page 15 line 14)

To confirm this, we proposed a potential catalytic mechanism of STPE-PMNS in combination with DFT calculations. The configuration in our computation is an idealized representation constructed based on experiment results. The AIMD results indicate that the PMNSs catalyst can maintain its structural stability under 800 K conditions (Supplementary Fig. 52). The catalytic reaction proceeds in two steps. The first step is the activation of the reactants. At the beginning, Glu is adsorbed on the surface of the nanosheet, which binds with the metal atom Mn and then attacks the terminal phosphate group of polyP to originate γ -glutamyl phosphate (Fig. 5a). This step is the activation process of Glu, which is an exergonic reaction with a reaction-free energy of -0.89 eV. The active site of NH_4^+ is the terminal phosphate of polyP within STPE-PMNS, a negatively charged region that rapidly binds NH_4^+ , where NH_4^+ loses its proton and becomes a basic residue ready for the nucleophilic attack on the C_γ of γ -glutamyl phosphate intermediate (Fig. 5a). Activation of NH_4^+ can happen either before or after the formation of γ -glutamyl phosphate intermediate.

The second step corresponds to Gln formation, via a nucleophilic attack of ammonia to the C_γ of the γ -glutamyl phosphate intermediate. At this step, ammonia starts the

nucleophilic attack on the C_γ of γ -glutamyl phosphate intermediate, with the concomitant bond breaking of the C_γ -phosphate and bond formation of the C_γ -ammonia bond (Fig. 5a). Subsequently, upon the formation of protonated glutamine, a rapid proton transfer occurs to the released phosphate group, resulting in the production of Gln. The energy profile is shown in Fig. 5b, and the total catalytic reaction energy is -1.53 eV. The catalytic mechanism is consistent with a mechanism catalyzed by natural enzymes⁶³⁻⁶⁷.

Fig. 5 Reaction mechanism and energies of STPE-PMNSs catalyzing the conversion of Glu to Gln. (a) Glu (left) and NH_4^+ (right) activation and Gln formation step (down). (b) Energy profile for STPE-PMNSs catalyzed conversion of Glu to Gln.

Reviewer 2

This paper summarizes the research related to the development of manganese nanosheets enabled with polyP tetraphenylethylene analogs with size-tunable, uniform, and stable fluorescent properties, through a hierarchical assembly strategy. The synthesized nanosheets served as artificial glutamine synthetase with the ability to convert glutamate to glutamine for biomedical applications. Their catalytic properties are tested in human neuroblastoma cells with a high efficacy in the conversion of glutamate to glutamine alleviating neurotoxicity. This is an interesting and important area of research. The paper is generally well written; however, the quality of the paper can be enhanced if the following points can be addressed.

Query 1

There are a few grammatical errors in the manuscript which need to be rectified. For example; Abstract: “Glutamine synthase (GS) plays crucial roles”. “Extensively explored for chemical engineering and biomedical”.

Response

We thank the reviewer for this comment.

We have corrected the grammatical errors in the revised manuscript.

Revisions Made

(Please refer to the manuscript on page 1, line 16)

Glutamine synthetase (GS) plays a crucial role in maintaining ammonia and glutamate (Glu) homeostasis in living organisms, which is extensively explored for chemical engineering and biomedical.

(Please refer to the manuscript on page 4, line 7)

Additionally, we found that STPE-PMNSs can serve as an alternative to GS in SH-SY5Y cells to efficiently catalyze the conversion of Glu to Gln, relieving Glu-induced neurotoxicity.

Query 2

In Line 202, the authors mentioned that “Figure 4i revealed that polyP-Ca and MnO₂

could not catalyze the reaction of Glu to Gln, indicating the combination of polyP and Mn^{2+} is necessary for the catalytic effects". The authors need to discuss why this combination can have catalytic properties and why polyP-Ca and MnO_2 do not possess any catalytic properties as individual entities.

Response

We thank the reviewer for this comment.

For natural GS, Mn^{2+} or Mg^{2+} has a structural as well as a catalytic role, which is involved in stabilizing the structure, binding to Glu, and promoting phosphoryl transfer of ATP to Glu¹⁻⁴. For STPE-PMNSs, in addition to their role as stabilizers of nanosheets, Mn^{2+} ions also function as metal centers akin to natural enzymes. They are involved in both the facilitation of Glu binding and the phosphoryl transfer of polyP to Glu. Furthermore, the terminal phosphate of polyP in the nanosheets acts as an ATP-like molecule to activate Glu. Consequently, the presence of the same metal center in STPE-PMNSs and natural GS, as well as the comparable Glu-activating functions exhibited by polyP and ATP, determines the catalytic activity of STPE-PMNSs (Fig. 5a).

Besides, we compared the surface properties of STPE-PMNSs, polyP-Ca, and MnO_2 . Our findings revealed that the surface charge of STPE-PMNSs was opposite to that of Glu (Supplementary Fig. 44). This result suggests that the surface properties of STPE-PMNSs may enhance the adsorption of Glu. The adsorption energy of STPE-PMNSs for Glu was determined to be -0.89 eV (Fig. 5). It is an exothermic process, indicating its thermodynamic feasibility. On the other hand, the surface charges of polyP-Ca and MnO_2 were found to reject Glu, resulting in Glu being difficult to adsorb on the surface of the nanosheets. Consequently, polyP-Ca and MnO_2 do not possess the corresponding catalytic properties.

Revisions Made

(Please refer to the manuscript on page 12 line 15 to page 13 line 2)

Fig. 4f revealed that polyP-Ca and MnO_2 could not catalyze the reaction of Glu to Gln, indicating the combination of polyP and Mn^{2+} is necessary for the catalytic effects. To validate this finding, we compared the surface properties of STPE-PMNSs, polyP-Ca, and MnO_2 . Zeta potential results indicate that the surface charge of STPE-PMNSs is

opposite to that of Glu (Supplementary Fig.44), suggesting the surface characteristics of STPE-PMNSs might enhance Glu adsorption. The adsorption energy of Glu on STPE-PMNSs is -0.89 eV (Fig. 5b), supporting its thermodynamic feasibility. Conversely, the surface charges of polyP-Ca and MnO₂ were found to reject Glu (Supplementary Fig.44), resulting in the lack of corresponding catalytic properties of polyP-Ca and MnO₂.

Supplementary Fig. 44 Zeta potential of polyP-Ca, MnO₂, and STPE-PMNSs.

Fig. 5 Reaction mechanism and energies of STPE-PMNSs catalyzing the

conversion of Glu to Gln. (a) Glu (left) and NH_4^+ (right) activation and Gln formation step (down). (b) Energy profile for STPE-PMNSs catalyzed conversion of Glu to Gln.

Query 3

MnO_2 is a commonly used catalyst. Among all the transition metal oxides, it is known to exhibit stable performance in applications such as supercritical water oxidation and is known to have relatively high catalytic activity in the catalytic decomposition of organic compounds by oxidation and also inside the cells and animals. Why did the authors not measure or compare the intracellular catalytic activity of MnO_2 alone with the nanosheets that the authors synthesized in the Figure 5 experiment?

Response

We thank the reviewer for this comment.

As suggested, we evaluated the intracellular GS-like activity of MnO_2 . As shown in Supplementary Fig. 63, the treatment with MnO_2 did not result in a significant down-regulation in Glu content within the cells. Furthermore, the results obtained from flow cytometry and WB analysis indicated that MnO_2 was unable to attenuate Glu-induced neurotoxicity.

Revisions Made

(Please refer to the manuscript on page 19, lines 20-22)

In the intracellular activity assessment, MnO_2 treatment does not significantly reduce the Glu content in the cells, and could not attenuate Glu-induced neurotoxicity (Supplementary Fig. 63).

Supplementary Fig. 63 The extracellular catalytic activity study of MnO₂. (a) Intracellular GS-like activity of MnO₂. (b) Cell viability of SH-SY5Y cells treated with different concentrations of MnO₂ for 24 h. (c-d) The content of intracellular Glu and Gln in different treatment groups after 24 h. (e) Flow cytometry analysis of the apoptosis level of SH-SY5Y cells after 24 h in different treatment groups. (f-g) Bcl-2 expression levels in SH-SY5Y cells after 24 hours in different treatment groups. (h-i) Bax expression levels in SH-SY5Y cells after 24 hours in different treatment groups. All data are representative of at least three (n = 3) independent experiments for each experimental group and are displayed as mean ± standard deviation. **P* < 0.05, indicates a statistically significant difference (Student's t-test).

Query 4

In Figure 5 (a) The authors mentioned that the nanosheets have a high biocompatibility 24h post exposure but in the figure only 12h data is presented. Please include the 24 h data in the main file instead of supporting information.

Response

We thank the reviewer for this comment.

We employed confocal microscopy to re-monitor the intracellular fluorescence signal of STPE-PMNS continuously for 24 h. The corresponding data was included in Fig. 6a of the main file.

Revisions Made

(Please refer to the manuscript on page 18, lines 14-20)

We utilized confocal laser scanning microscopy (CLSM) to monitor the real-time distribution of nanosheets within cells as well as changes in fluorescence intensity. As the reaction progressed, the fluorescence intensity continuously increased. This phenomenon is consistent with what was observed extracellular and may be attributed to the further aggregation of STPE-PMNSs. In contrast, the 4-AP untreated group does not exhibit significant changes in fluorescence intensity (Fig.6a).

Fig. 6 (a) Intracellular fluorescence signals were observed at different time points.

Query 5

Did the authors evaluate the dose-response for the synthesized nanosheets for Glu to Gln conversion?

Response

We thank the reviewer for this comment.

As suggested, we conducted dose-response experiments to investigate the catalytic activity of STPE-PMNSs. The extracellular dose response of STPE-PMNSs was initially verified within a concentration range of 0-400 $\mu\text{g mL}^{-1}$. In the range of 0-100 $\mu\text{g mL}^{-1}$, the activity increased linearly with the concentration (Fig. 3c). Subsequently, the intracellular dose response of STPE-PMNSs was examined. Following a 4 h

induction of 4-AP, cells were exposed to varying concentrations of STPE-PMNSs for 24 h. Supplementary Fig. 61 illustrates that as the concentration of STPE-PMNSs increased, the intracellular Glu content decreased while the Gln increased, providing further evidence for the dose-dependence of STPE-PMNSs catalysis. The above results collectively indicate that the catalytic effect of STPE-PMNSs is dose-dependent.

Revisions Made

(Please refer to the manuscript on page 8, lines 11-12)

As shown in Fig. 3c in the range of 0-200 $\mu\text{g mL}^{-1}$, the GS-like activity was enhanced with increasing concentration.

Fig. 3c Extracellular GS-like activity of STPE-PMNSs and the inset is linear relation in the low-concentration region.

(Please refer to the manuscript on page 16 line 19 to page 17 line 3)

Dose-response experiments demonstrate that with an increase in the concentration of STPE-PMNSs, there was a corresponding decrease in intracellular Glu levels, while Gln levels increased (Supplementary Fig. 61). These findings provide support for the dose-dependent STPE-PMNSs catalysis.

Supplementary Fig. 61 The intracellular dose-response of GS-like activity of STPE-

PMNSs.

Query 6

In the discussion, the authors need to discuss more about the impact of Glu cytotoxicity and its impact on nerve cells and the prospects in which the proposed nanomaterials can be applied in real-world applications.

Response

We thank the reviewer for this comment.

We have added the discussion about the impact of Glu cytotoxicity and its impact on nerve cells and prospects for practical applications of STPE-PMNSs in the revised manuscript.

Revisions Made

(Please refer to the manuscript on page 15, lines 15-22)

Glu serves as the primary excitatory neurotransmitter within the central nervous system and plays a critical role in various cognitive functions, including memory and learning⁶⁸. Disruptions in Glu homeostasis can impair synaptic function and result in neuronal death through a process known as excitotoxicity⁶⁹⁻⁷¹. Numerous studies have demonstrated the significant implications of this pathological mechanism in the development and progression of various neuropathological conditions. For instance, glutamate excitotoxicity has been implicated in the pathogenesis of neurological disorders such as cerebral ischemia⁷², traumatic brain injury⁷³, multiple sclerosis⁷⁴, epilepsy⁷⁵, and Alzheimer's disease⁷⁶. Consequently, acting on excitotoxic events is of utmost importance for treating these diseases.

(Please refer to the manuscript on page 4, lines 13-15)

Our study shows the first artificial GS, which effectively works in both chemical systems and living cells and has the potential to substitute natural GS in nerve cells to execute corresponding physiological functions.

Reviewer 3

In the manuscript entitled “AIE-Active Polyphosphate-Manganese Nanosheets as Artificial Glutamine Synthase in Living Cells”, the authors obtain Mn^{2+} AIE-active STPE-PMNSs that seem to catalyze the conversion of Glu to Gln and exhibited GS-like activity also in living cells.

The finding is surprising and very interesting. However, several critical points require further work to confirm the authors' claim, and more experimental details need to be included.

Here are some comments:

Query 1

STPE-PMNSs possess comparable enzymatic efficiency and similar catalytic reaction kinetics to that of natural GS. How the enzymatic-like activity is evaluated cannot be inferred from the Methods and should be described in detail. For the cellular assay, it looks like the particles cross freely the cellular membrane. However, an evaluation of the amount intracellularly delivered should be reported.

Response

We thank the reviewer for this comment.

We determined the catalytic activity of STPE-PMNSs and natural GS respectively employing the catalytic system developed by Allewell et al ¹.

The activity was determined in reaction mixtures consisting of 100 mM imidazole-HCl, 50 mM Glu, 25 mM 2-mercaptoethanol, 20 mM ATP, 20 mM $MgCl_2$, and 125 mM NH_4Cl in this assay. Stock solutions of imidazole, Glu, and ATP were titrated to pH 7.2. All reagents were stored on ice before assaying. 500 μL of the reaction mixture was pre-incubated for 2 min at 37 °C and the reaction was initiated by the addition of 250 μL GS (final concentration 100 $\mu g mL^{-1}$). The reaction was quenched after 15 min with 750 μL of 0.37 M $FeCl_3$, 0.67 M HCl, and 0.2 M trichloroacetic acid. The absorption was then read at 540 nm. Controls were carried out in the absence of Glu. The GS-like activity of STPE-PMNSs (final concentration 100 $\mu g mL^{-1}$) was carried out in HEPES buffer (100 mM, pH 7.3) in the absence of ATP. Meanwhile, in the above experimental setup, a standard curve was generated by introducing varying quantities of Gln without

the presence of any reactants. By utilizing the standard curve, the Gln content was determined by substituting the obtained absorbance values.

In addition, we quantified the internalization of STPE-PMNS by cells. Firstly, we assessed the uptake of STPE-PMNSs by monitoring the fluorescence signal changes within the cells at specific time intervals. The intracellular fluorescence signal exhibited an increase as the incubation time with STPE-PMNSs was extended (Supplementary Fig. 56a). Subsequently, we employed ICP-OES to quantify the intracellular Mn content of cells at different time points, thereby assessing the uptake of the nanosheets. ICP-OES results show that the cellular uptake of Mn element is $7.29 \text{ ng}/10^4 \text{ cells}$ after 24 h (Supplementary Fig. 56b). These results confirmed the strong cellular uptake of STPE-PMNSs.

Revisions Made

(Please refer to the manuscript on page 24, lines 5-18)

GS-like activity assessment. The activity was determined in reaction mixtures consisting of 100 mM imidazole-HCl, 50 mM Glu, 25 mM 2-mercaptoethanol, 20 mM ATP, 20 mM MgCl_2 , and 125 mM NH_4Cl in this assay. Stock solutions of imidazole, Glu, and ATP were titrated to pH 7.2. All reagents were stored on ice before assaying. 500 μL of the reaction mixture was pre-incubated for 2 min at 37 °C and the reaction was initiated by the addition of 250 μL GS (final concentration $100 \mu\text{g mL}^{-1}$). The reaction was quenched after 15 min with 750 μL of 0.37 M FeCl_3 , 0.67 M HCl, and 0.2 M trichloroacetic acid. The absorption was then read at 540 nm (Tecan Infinite M1000 PRO). Controls were carried out in the absence of Glu. The GS-like activity of STPE-PMNSs (final concentration $100 \mu\text{g mL}^{-1}$) was carried out in HEPES buffer (100 mM, pH 7.3) in the absence of ATP. Meanwhile, in the above experimental setup, a standard curve was generated by introducing varying quantities of Gln without the presence of any reactants. By utilizing the standard curve, the Gln content was determined by substituting the obtained absorbance values.

(Please refer to the manuscript on page 26, lines 12 to 20)

In Vitro Cellular Uptake of PMNSs. We conducted a quantitative analysis of the

internalization of STPE-PMNSs by cells over a defined time interval. SH-SY5Y cells (1.2×10^6 cells mL^{-1}) were seeded in 6-well tissue culture plates and grown to $\sim 90\%$ confluence. Growth media in plates was replaced with media containing PMNSs. Cells were incubated with nanomaterials for 0, 3 h, 6 h, 12 h, 18 h, and 24 h at 37°C . Finally, the medium was removed from the wells, and the cells were washed three times with PBS to clear away nanomaterials outside of the cells. Next, $200\ \mu\text{L}$ of 1% Triton X-100 in 0.1M NaOH solution was added to lyse the cells. After treatment, Mn content in the lysates was determined using ICP-OES.

(Please refer to the manuscript on page 16, lines 11-17)

Then, we tested the intracellular uptake of STPE-PMNSs by SH-SY5Y cells. The intracellular fluorescence signal exhibits a progressive increase as the incubation time with STPE-PMNSs extends (Supplementary Fig. 56a). ICP-OES results show that the cellular uptake of Mn element is $7.29\ \text{ng}/10^4$ cells after 24 h (Supplementary Fig. 56b), indicating rapid and strong cellular uptake of STPE-PMNSs that ensure they are sufficient to exert physiological effects after ingestion.

Supplementary Fig. 56 Internalization of STPE-PMNSs by cells at specific time intervals. (a) fluorescence signal changes within the cells over various periods. (b) ICP results of the SH-SY5Y cells uptake of STPE-PMNSs.

Query 2

line 159: “V is the reaction rate”, change with V is the initial reaction rate.

Response

We thank the reviewer for pointing out our errors and we have corrected the errors

in the revised manuscript.

Revisions Made

(Please refer to the manuscript on page 9, lines 11-17)

The kinetics equation of STPE-PMNSs was fitted by the Michaelis–Menten equation as follows:

$$V = V_{max}[S]/(K_m + [S]) \quad (1)$$

where V is the initial reaction rate, V_{max} is the maximal velocity, $[S]$ is the concentration of substrate, and K_m is the Michaelis constant.

Query 3

Line 160-162: The apparent Michaelis–Menten constants of Glu and NH_4^+ substrates are 130.67 μM and 156.65 μM , respectively, revealing enzyme-like first-order kinetic behavior (Figs 3c-d and Supplementary Fig. 30). It is not clear from the Methods the GS they are referring to for the K_m values. For humans GS K_m should be around 2-3mM and for other species that value should not differ much from that value. So it is not clear where the reported values for K_m come from.

Response

We thank the reviewer for this comment.

We re-determined the apparent Michaelis-Menten constant of STPE-PMNSs and natural GS respectively according to the following protocol:

The measurements of the apparent K_m of STPE-PMNSs were performed at 37 °C in HEPES buffer (100 mM, pH 7.3). To measure the apparent K_m of STPE-PMNSs for Glu, we performed the catalytic reactions at growing concentrations of Glu, from 0 to 20 mM, while keeping the STPE-PMNSs and the NH_4^+ concentration constant (100 $\mu\text{g mL}^{-1}$ and 20 mM, respectively). In this experimental series, an apparent K_m of 0.75 mM of Glu was obtained. An equivalent experiment was carried out to determine the apparent K_m for NH_4^+ . For that, we varied the NH_4^+ concentration from 0 to 20 mM while holding the Glu and STPE-PMNSs concentrations to 20 mM and 100 $\mu\text{g mL}^{-1}$, respectively. In this case, an apparent K_m of 0.099 mM of NH_4^+ was obtained.

For natural GS, the apparent K_m was determined in reaction mixtures consisting of 100 mM imidazole-HCl, 40 mM Glu, 25 mM 2-mercaptoethanol, 20 mM ATP, 20 mM $MgCl_2$, and 40 mM NH_4Cl . Stock solutions of imidazole, Glu, and ATP were titrated to pH 7.2. All reagents were stored on ice before assaying. When Glu was varied, NH_4^+ and GS were held constant at 40 mM and $100 \mu g mL^{-1}$, respectively. When NH_4^+ was varied, Glu and GS were held constant at 40 mM and $100 \mu g mL^{-1}$. ATP was held constant at 20 mM. The apparent Michaelis–Menten constants of Glu and NH_4^+ substrates are 2.04 mM and 0.16 mM, respectively. In both cases enzyme-like first order kinetic behavior was observed.

Our results indicate that the Michaelis-Menten constant for natural GS is approximately 2.04 mM for Glu, which closely aligns with the range of 2-3 mM proposed by the reviewers (Supplementary Fig. 40). This discrepancy could potentially be attributed to subtle differences in pH, temperature, and reaction environment.

Revisions Made

(Please refer to the manuscript on page 25 line 20 to page 26 line 6)

Calculation of the apparent Michaelis-Menten constant of STPE-PMNSs and natural GS. The measurements of the apparent K_m of STPE-PMNSs were performed at 37 °C in HEPES buffer (100 mM, pH 7.3). To measure the apparent K_m of STPE-PMNSs for Glu, we performed the catalytic reactions at growing concentrations of Glu, from 0 to 20 mM, while keeping the STPE-PMNSs and the NH_4^+ concentration constant ($100 \mu g mL^{-1}$ and 20 mM, respectively). An equivalent experiment was carried out to determine the apparent K_m for NH_4^+ . For that, we varied the NH_4^+ concentration from 0 to 20 mM while holding the Glu and STPE-PMNSs concentrations to 20 mM and $100 \mu g mL^{-1}$, respectively.

For natural GS, the apparent K_m was determined in reaction mixtures consisting of 100 mM imidazole-HCl, 40 mM Glu, 25 mM 2-mercaptoethanol, 20 mM ATP, 20 mM $MgCl_2$, and 40 mM NH_4Cl . Stock solutions of imidazole, Glu, and ATP were titrated to pH 7.2. All reagents were stored on ice before assaying. When Glu was varied, NH_4^+ and GS were held constant at 40 mM and $100 \mu g mL^{-1}$, respectively. When NH_4^+ was varied, Glu and GS were held constant at 40 mM and $100 \mu g mL^{-1}$. ATP was held

constant at 20 mM. In both cases enzyme-like first order kinetic behavior was observed.

Supplementary Fig. 40 Michaelis–Menten curves of natural GS with Glu and NH_4^+ as substrate.

Query 4

The sentence “revealing enzyme-like first-order kinetic behavior” is conceptually wrong. You cannot infer from K_m that the kinetic is first-order. You evaluate if the kinetic is first order, then K_m is calculated.

Response

We thank the reviewer for pointing out shortcomings in the manuscript.

We revised the expression of this sentence into “We observed an enzyme-like first-order kinetic behavior from the Michaelis–Menten curves. The app K_m values for Glu and NH_4^+ substrates were determined to be 0.75 mM and 0.099 mM, respectively.”

Revisions Made

(Please refer to the manuscript on page 9, lines 15-17)

We observed an enzyme-like first-order kinetic behavior from the Michaelis–Menten curves. The apparent K_m values for Glu and NH_4^+ substrates were determined to be 0.75 mM and 0.099 mM, respectively (Figs. 3d-e).

Query 5

The protective effect of STPE-PMNSs was tested intracellularly after using a molecule that raises intracellular Glu levels. It would be interesting to test Glu excitotoxicity in neuronal cells, by evaluating the Glu scavenging ability of STPE-PMNSs

extracellularly.

Response

We thank the reviewer for this comment.

We evaluated the Glu scavenging ability of STPE-PMNSs extracellular. Glu and STPE-PMNSs were co-incubated with the cells and then the levels of Glu in the medium were measured. The findings demonstrated a gradual decrease in Glu levels in the culture medium with increasing co-incubation time with STPE-PMNSs (Supplementary Fig. 64). Additionally, flow cytometry and WB analysis revealed that STPE-PMNSs can alleviate Glu-induced excitotoxicity and rescue neuronal cell apoptosis caused by excessive Glu through the extracellular conversion of Glu to Gln (Supplementary Fig. 65).

Revisions Made

(Please refer to the manuscript on page 20, lines 1-8)

Furthermore, we evaluated the Glu scavenging ability of STPE-PMNSs extracellular. Glu and STPE-PMNSs were co-incubated with the cells and then the levels of Glu in the medium were measured. The findings demonstrate a gradual decrease in Glu levels in the culture medium with increasing co-incubation time with STPE-PMNSs (Supplementary Fig. 64). Additionally, flow cytometry and WB analysis revealed that STPE-PMNSs could alleviate Glu-induced excitotoxicity and rescue neuronal cell apoptosis caused by excessive Glu through catalyzing the extracellular conversion of Glu to Gln (Supplementary Fig. 65).

Supplementary Fig. 64 (a) Cell viability of SH-SY5Y cells treated with various concentrations of Glu for 24 h. (b) Relationship between Glu levels in the medium and

incubation time of STPE-PMNSs.

Supplementary Fig. 65 Extracellular Glu scavenging ability of STPE-PMNSs. (a) The apoptosis level of SH-SY5Y cells after 24 h in different treatment groups. (b-c) Bcl-2 expression levels in SH-SY5Y cells after 24 hours in different treatment groups. (d-e) Bax expression levels in SH-SY5Y cells after 24 hours in different treatment groups. All data are representative of at least three ($n = 3$) independent experiments for each experimental group and are displayed as mean \pm standard deviation. $*P < 0.05$, indicates a statistically significant difference (Student's t-test).

Query 6

Supplementary Fig. 22. The authors report the % Glu and Gln with time. It looks like a time course, but I doubt that the trend is linear in such a large time range (100 min).

Response

We thank the reviewer for this comment.

The relationship between Glu and Gln levels depicted in Supplementary Fig. 29 (modified versions) is established based on the data acquired from NMR integrations presented in Supplementary Figs. 20-27. We narrowed the time interval and re-examined the relationship between Glu and Gln levels within the 0-100 min timeframe.

Our results indicate a linear decrease in Glu levels and a corresponding increase in Gln levels during the initial 60 min of the reaction system (Supplementary Fig. 29). However, between 60-240 min, the conversion rates of Glu to Gln gradually decelerates, resulting in a non-linear change in the levels of Glu and Gln during this period. The independent NMR data obtained at each time point has been included in the Supplementary Information (Supplementary Figs. 20-27).

Revisions Made

(Please refer to the manuscript on page 8, lines 14-15)

The ^1H NMR spectra revealed Gln as the conversion product of the reaction with a conversion rate of 75.57% in 4 h (Supplementary Figs. 20-29).

Supplementary Fig. 28 Merged ^1H NMR spectra from Supplementary Fig. 20 to Supplementary Fig. 27.

Supplementary Fig. 29 The results of NMR integrations from Supplementary Fig. 20 - Supplementary Fig. 27. The conversion rates of Glu and Gln in the catalytic system (a) and a linear relation in the 0-60 min (b).

Query 7

Supplementary Fig. 22. Again, the activity of “natural GS” is reported. Are they referring to the human one? The activity is evaluated for 60 min, which is a very large time range. The activity is usually evaluated from 2 to 15 minutes (doi: 10.1042/bj3280159).

Response

We thank the reviewer for this comment.

The "natural GS" here refers to human glutamine synthetase (EC 6.3.1.2). As shown in Supplementary Figs. 38-40, we conducted a reassessment of the catalytic kinetics of natural GS over 2-14 min, employing the catalytic system developed by Allewell et al¹.

Revisions Made

(Please refer to the manuscript on page 9, lines 9-10)

We also calculated the catalytic reaction rates of STPE-PMNSs and GS in these dose ranges (Supplementary Figs. 36-39 and Supplementary Tables 1, 2).

Supplementary Fig. 38 The reaction rate of natural GS in different Glu concentrations.

Supplementary Fig. 39 The reaction rate of natural GS in different NH_4^+ concentrations.

Supplementary Fig. 40 Michaelis–Menten curves of natural GS with Glu and NH_4^+ as substrate.

Query 8

Supplementary table 1 and 2 report the reaction rate. The unit of measure is not correct in my opinion. The enzymatic activity indicates the amount of substrate converted in the time unit in the reaction volume (for instance: micromol/min/mL). If they know the used amount of these particles, it should be even better to normalize the amount/volume of particles to obtain micromole/min/mg of particles.

Response

We thank the reviewer for pointing out our errors and we have corrected the errors in the revised manuscript.

Revisions Made

(Please refer to the Supplementary Information on pages 31 and 33)

Supplementary Table 1. Summary of reaction rate of STPE-PMNSs in different Glu and NH₄⁺ concentrations.

Glu (mM)	Reaction rate (nmol/min/mg)	NH ₄ ⁺ (mM)	Reaction rate (nmol/min/mg)
0	2.5 x 10 ⁻³	0	6.3 x 10 ⁻³
0.1	0.8	0.1	1.4
2.5 x 10 ⁻¹	1.2	0.25	1.5
0.5	1.4	0.5	1.8
1.0	1.7	1.0	2.3
2.0	2.9	2.0	2.4
5.0	3.6	5.0	2.3
10.0	3.5	10.0	2.3
20.0	3.5	20.0	2.4

The unit of reaction rate is nmol/min per mg of STPE-PMNSs.

Supplementary Table 2. Summary of reaction rate of natural GS in different Glu and NH₄⁺ concentrations.

Glu (mM)	Reaction rate (nmol/min/mg)	NH ₄ ⁺ (mM)	Reaction rate (nmol/min/mg)
0	0.5 x 10 ⁻²	0	0.6 x 10 ⁻²
0.1	2.7	0.5 x 10 ⁻¹	3.9
0.5	3.9	0.1	7.4
1.0	5.7	2.5 x 10 ⁻¹	10.6
2.0	7.9	0.5	16.4
5.0	9.0	1.0	16.9
10.0	11.3	2.0	17.1
20.0	13.3	5.0	17.2
40.0	15.4	10.0	19.4

The unit of reaction rate is nmol/min per mg of natural GS.

Query 9

Supplementary Fig. 30. The unit is not correct, it should be micromol/min.

Response

We thank the reviewer for pointing out our shortcomings and we have corrected this

in the manuscript.

Revisions Made

(Please refer to the Supplementary Information on page 34)

Supplementary Fig. 40 Michaelis–Menten curves of natural GS with Glu and NH_4^+ as substrate.

Reviewer 4

The manuscript describes polyphosphate-Mn nanosheets (PMNSs) as biomimetic catalysts with glutamine synthetase activity, a novel property for this nanomaterial and a promising alternative to natural glutamine synthetase enzyme. Further, nanosheets are functionalized with AIE-active groups (STPE), which enable visualization of nanosheets and fluorescence monitoring of aggregation status. The authors provided rigorous characterization of nanosheets and glutamine synthetase-like activity in vitro, performed computational analysis to elucidate the mechanism of the catalytic activity, and demonstrated biocompatibility and functionality of STPE-PMNSs in live cells. The manuscript presents the work of potential broad interest spanning fields of synthetic and computational chemistry, bio-nanotechnology, and molecular and cell biology. However, the manuscript lacks clarity on the novelty and impact of the reported developments/findings, which obscures the key scientific contributions of the manuscript and leads to the following requests:

Query 1

There is no support for the claim that this work “provides opportunities for applying multifunctional polymers to biomedicine” and other similarly broad statements. The manuscript would be strengthened by, instead, clearly describing the novelty and impact of the specific developments/findings presented. In general, it would help for the claims to be strictly limited to the actual material properties, phenomena observed, and applications demonstrated.

Response

We thank the reviewer for pointing out shortcomings in the manuscript

We have made revisions to general statements such as "provides opportunities for applying multifunctional polymers to biomedicine" by the characteristics and applications of STPE-PMNSs. Firstly, we revised the expression in the Abstract “Our study presents the first artificial GS and the combination of AIE properties based on long-chain polyP providing opportunities for applying multifunctional polymers to biomedicine” to “This research presents a biopolymer polyP-based fluorescent nanosheet with GS-like activity that acts as a promising alternative to natural GS.”

In the Introduction, we have modified the statement "This study enriches the application of polyP-based nanomaterials in the realm of biomimetic catalysis, and the integration of AIEgens and polyP offers opportunities for the application of multifunctional biopolymers in the field of biomedicine" to " Our study shows the first artificial GS, which effectively works in both chemical systems and living cells and has the potential to substitute natural GS in nerve cells to execute corresponding physiological functions."

Revisions Made

(Please refer to the manuscript on page 2, line 7)

This research presents a biopolymer polyP-based fluorescent nanosheet with GS-like activity that acts as a promising alternative to natural GS.

(Please refer to the manuscript on page 4, line 12)

Our study shows the first artificial GS, which effectively works in both chemical systems and living cells and has the potential to substitute natural GS in nerve cells to execute corresponding physiological functions.

Query 2

The AIE element does not appear to be essential for nanosheet assembly or glutamine synthetase-like activity; hence, the addition of STPE to nanosheets is puzzling. The rationale for including STPE needs to be discussed, particularly in reference to increased material complexity and potential alterations to material properties. The role of AIEgen as an aggregation sensor might need to be more clearly outlined and linked to its uses within the STPE-PMNS structure.

Response

We thank the reviewer for this comment.

As mentioned by the reviewer, the inclusion of the AIE group is not necessary for the assembly and catalytic activity of the nanosheets. However, the majority of these nanomaterials are inorganic and lack fluorescence emission.^{2, 3} Consequently, the examination of their subcellular localization has been hindered. To overcome this

limitation, we incorporated AIE gens into our catalytic system. This modification endowed the nanosheets with fluorescent properties, enabling us to effectively monitor their intracellular distribution after cellular uptake.

It has to be considered that the introduction of STPE groups into materials results in increased complexity and changes in material properties. Firstly, we investigated the potential biosafety concerns associated with the increased complexity of the nanosheets. The introduction of STPE had minimal effect on cell viability and the nanosheets remain excellent biosafety (Supplementary Fig. 53). Secondly, we specifically examined the variation in catalytic performance between STPE-PMNSs and PMNSs. Our findings revealed that there was a negligible difference in catalytic activity between STPE-PMNSs and PMNSs when compared at the same concentration (Supplementary Fig. 18). In conclusion, although the incorporation of STPE groups increases material complexity, it does not compromise the biosafety or catalytic performance of the nanosheets.

During the extracellular experiment, we made an unexpected discovery that the overall fluorescence intensity of the system increased as the reaction progressed (Supplementary Fig. 30). In the meantime, the analysis of the catalytic mechanism (refer to Query 3 for specific details) indicated Glu activated by the terminal phosphate group of the polyP within STPE-PMNS. This activation is analogous to the role of ATP in the catalysis of natural GS, resulting in the detachment of the terminal phosphate group from the polyP. The consumption of the phosphate group might increase the surface energy of the nanosheets, causing them to aggregate to stabilize the structure. The aggregation of the STPE-PMNSs along the catalytic progress further triggered aggregation-induced emission enhancement. These findings give us an opportunity to use fluorescence intensity changes as a means of assessing the progress of the catalytic reaction. Further investigations revealed a direct correlation between alterations in fluorescence intensity and variations in intracellular Gln levels (Figs. 6a-d). Consequently, the integration of an AIE group can serve as an indicator for monitoring the Glu content in the system.

Revisions Made

(Please refer to the manuscript on page 3, lines 11-18)

Moreover, when nanostructures are applied to biological systems, rapid, synchronous, and nondestructive visualization of biomimetic catalytic processes is key to evaluating subcellular localization and monitoring the progress of catalytic reactions⁴⁴⁻⁴⁷. However, the majority of these nanomaterials are inorganic and lack fluorescence properties^{48,49}. Aggregation-induced emission (AIE), a revolutionary concept, was first established by Tang et al. in 2001⁵⁰. AIE represents a powerful tool in nanoscience, in which the long-term fluorescence tracking of dynamic biological processes with high photostability could be realized using AIEgens⁵¹⁻⁵⁷.

(Please refer to the manuscript on page 16, lines 3-7)

First, cytotoxicity experiments were performed to assess the biocompatibility of STPE-PMNSs and PMNSs. The SH-SY5Y cell viability was not affected by the treatment of nanosheets for up to 24 h, which indicates that the introduction of STPE has no significant effect on cell viability (Supplementary Figs. 53-54).

Supplementary Fig. 53 The Cell viability of SH-SY5Y cells treated with STPE-PMNSs and PMNSs for 24 h.

(Please refer to the manuscript on page 8, lines 6-7)

Additionally, the results show the introduction of STPE does not affect the catalytic activity of the nanosheets (Fig. 3b and Supplementary Fig. 18).

Supplementary Fig. 18 The relative activity of PMNSs and STPE-PMNSs under different concentrations.

(Please refer to the manuscript on page 8, lines 17-19)

Significantly, an increase in fluorescence intensity was observed throughout the catalytic process (Supplementary Fig. 30).

Supplementary Fig. 30 (a) The fluorescence intensity changes in the catalytic system. (b) The fluorescence emission spectrum of STPE-PMNSs before and after the catalytic process.

(Please refer to the manuscript on page 18 line 14 to page 19 line 4)

We utilized confocal laser scanning microscopy (CLSM) to monitor the real-time distribution of nanosheets within cells as well as changes in fluorescence intensity. As the reaction progressed, the fluorescence intensity continuously increased. This phenomenon is consistent with what was observed extracellular and may be attributed to the further aggregation of STPE-PMNSs. In contrast, the 4-AP untreated group does not exhibit significant changes in fluorescence intensity (Fig.6a). The obtained results

have prompted our investigation into the feasibility of evaluating the progress of the catalytic reaction by measuring changes in intracellular fluorescence intensity. Subsequently, we conducted a relative quantification of fluorescence intensity in Fig. 5a and assessed intracellular Gln levels at the above time intervals (Figs. 6b-c). Our findings indicate a linear correlation between alterations in fluorescence intensity and changes in intracellular Gln content (Fig. 6d).

Fig. 6 Intracellular verification of the catalytic activity of STPE-PMNSs. Intracellular fluorescence signals were observed at different time points (a), and relative quantification of the corresponding fluorescent signal was conducted (b). The change in intracellular Gln content corresponds to different time points (c) and the linear relationship between Gln content and relative fluorescence intensity (d).

Query 3

The description of the computational work and the proposed mechanism of catalytic activity lacks details. The calculation results and interpretation need to be presented, and the logic used to determine steps and sequence of events needs to be outlined. It is unclear what drives the endothermic reaction (and whether there is any experimental evidence for that), where the energy for the reaction comes from, and why the system would not remain in the lowest energy state. Further, the model does not explain nanosheet aggregation during catalysis (observed experimentally) or the role of the

unique combination of polyphosphate with Mn^{2+} in nanosheets. Thus, the computational component needs a more thorough analysis and a better integration into the overall flow of the manuscript.

Response

We thank the reviewer for this comment.

We re-proposed a potential catalytic mechanism of STPE-PMNS in combination with DFT calculations. The catalytic reaction proceeds in two steps.

The first step is the activation of the reactants. At the beginning, Glu is adsorbed on the surface of the nanosheet, which binds with the metal atom Mn and then attacks the terminal phosphate group of polyP to originate γ -glutamyl phosphate (Fig. 5a). This step is the activation process of Glu, which is an exergonic reaction with a reaction-free energy of -0.89 eV. The active site of NH_4^+ is the terminal phosphate of polyP within STPE-PMNS, a negatively charged region that rapidly binds NH_4^+ , where NH_4^+ loses its proton and becomes a basic residue ready for the nucleophilic attack on the C_γ of γ -glutamyl phosphate intermediate (Fig. 5a). Activation of NH_4^+ can happen either before or after the formation of γ -glutamyl phosphate intermediate.

The second step corresponds to Gln formation, via a nucleophilic attack of ammonia to the C_γ of the γ -glutamyl phosphate intermediate. At this phase, ammonia starts the nucleophilic attack on the C_γ of γ -glutamyl phosphate intermediate, with the concomitant bond breaking of the C_γ -phosphate and bond formation of the C_γ -ammonia bond (Fig. 5a). Subsequently, upon the formation of protonated glutamine, a rapid proton transfer occurs to the released phosphate group, resulting in the production of Gln. The energy profile is shown in Fig. 5b, and the total catalytic reaction energy is -1.53 eV, which is lower than 1.5 eV. The results indicated that it is thermodynamically feasible. The catalytic mechanism is consistent with a mechanism catalyzed by natural enzymes⁶³⁻⁶⁷.

We performed catalytic experiments both with and without ATP. TEM images revealed that in the absence of ATP, the nanosheets exhibited significant aggregation after catalysis, whereas in the presence of ATP, the nanosheets did not aggregate (Supplementary Figs. 46 and 48). Consequently, we thought that the aggregate of

nanosheets might be attributed to the detachment of phosphate groups within the material, which results in the increased surface energy of the material. We are diligently investigating the underlying factors contributing to the aggregation of nanosheets and more detailed reasons are still under study.

Revisions Made

(Please refer to the manuscript on page 14 line 6 to page 15 line 14)

To confirm this, we proposed a potential catalytic mechanism of STPE-PMNS in combination with DFT calculations. The configuration in our computation is an idealized representation constructed based on experiment results. The AIMD results indicate that the PMNSs catalyst can maintain its structural stability under 800 K conditions (Supplementary Fig. 52). The catalytic reaction proceeds in two steps. The first step is the activation of the reactants. At the beginning, Glu is adsorbed on the surface of the nanosheet, which binds with the metal atom Mn and then attacks the terminal phosphate group of polyP to originate γ -glutamyl phosphate (Fig.5a). This step is the activation process of Glu, which is an exergonic reaction with a reaction-free energy of -0.89 eV. The active site of NH_4^+ is the terminal phosphate of polyP within STPE-PMNS, a negatively charged region that rapidly binds NH_4^+ , where NH_4^+ loses its proton and becomes a basic residue ready for the nucleophilic attack on the $\text{C}\gamma$ of γ -glutamyl phosphate intermediate (Fig. 5a). Activation of NH_4^+ can happen either before or after the formation of γ -glutamyl phosphate intermediate.

The second step corresponds to Gln formation, via a nucleophilic attack of ammonia to the $\text{C}\gamma$ of the γ -glutamyl phosphate intermediate. At this step, ammonia starts the nucleophilic attack on the $\text{C}\gamma$ of γ -glutamyl phosphate intermediate, with the concomitant bond breaking of the $\text{C}\gamma$ -phosphate and bond formation of the $\text{C}\gamma$ -ammonia bond (Fig.5a). Subsequently, upon the formation of protonated glutamine, a rapid proton transfer occurs to the released phosphate group, resulting in the production of Gln. The energy profile is shown in Fig. 5b, and the total catalytic reaction energy is -1.53 eV. The catalytic mechanism is consistent with a mechanism catalyzed by natural enzymes⁶³⁻⁶⁷.

Supplementary Fig. 52 The AIMD diagram of PMNSs catalyst.

Fig. 5 Reaction mechanism and energies of STPE-PMNSs catalyzing the conversion of Glu to Gln. (a) Glu (left) and NH_4^+ (right) activation and Gln formation step (down). (b) Energy profile for STPE-PMNSs catalyzed conversion of Glu to Gln.

Supplementary Fig. 46 The TEM image of STPE-PMNPs in the presence (a) or absence (b) of ATP.

Supplementary Fig. 48 XPS analysis of STPE-PMNSs in the presence (a) or absence (b) of ATP.

Query 4

Aggregation of nanosheets during catalytic activity, accumulation in cells, and long-term toxicity need to be further assessed, particularly since live cell application is presented as the main objective of the manuscript.

Response

We thank the reviewer for this comment.

We conducted a further evaluation of nanosheet aggregation during catalysis. For the

extracellular catalytic system, we found that the fluorescence intensity gradually increased as the catalytic time was prolonged (Supplementary Fig. 30a). Besides, we collected the nanosheets for further analysis. We compared the fluorescence properties of the nanosheets before and after catalysis. Our results revealed that the emission peak positions remained relatively unchanged, but the emission intensity significantly increased after catalysis (Supplementary Figure 30b). Additionally, we investigated the morphological and size changes of the nanosheets in various reaction durations. TEM images and DLS analysis revealed that the aggregation of the nanosheets became more pronounced with the extension of time (Supplementary Fig. 31). Furthermore, we investigated the intracellular aggregation of nanosheets. Fig. 6a illustrates that the fluorescence intensity increased with the extension of catalytic time.

Furthermore, we conducted an assessment of the intracellular accumulation and long-term cytotoxicity of nanosheets. We examined the toxicity of STPE-PMNSs on three types of nerve cells, namely SH-SY5Y, PC-12, and U87 over 7 days. Additionally, we measured the level of Mn in the cells at corresponding time points to reflect intracellular nanosheet accumulation. The findings revealed that even after 7 days of exposure to STPE-PMNSs ($200 \mu\text{g mL}^{-1}$), cell viability remained at approximately 80% (Supplementary Fig. 55a). Moreover, intracellular Mn accumulation reached its maximum level at approximately $15.5 \text{ ng}/10^4$ cells on the third day of treatment and gradually stabilize at this level (Supplementary Fig. 55b).

Revisions Made

(Please refer to the manuscript on page 8, lines 17-22)

Significantly, an increase in fluorescence intensity was observed throughout the catalytic process (Supplementary Fig. 30). To investigate the underlying cause of this phenomenon, the catalyzed nanosheets were collected and further analyzed. The results of TEM and DLS analysis reveal that the nanosheets underwent aggregation after catalyzing, resulting in the further restriction of intramolecular motion of the fluorophore, which further enhanced the fluorescence of the nanosheets (Supplementary Fig. 31).

Supplementary Fig. 30 (a) The fluorescence intensity changes in the catalytic system. (b) The fluorescence emission spectrum of STPE-PMNSs before and after the catalytic process.

Supplementary Fig. 31 The TEM image (a) and DLS analysis (b) of STPE-PMNSs after catalysis at different time points.

(Please refer to the manuscript on page 18, lines 14-20)

We utilized confocal laser scanning microscopy (CLSM) to monitor the real-time distribution of nanosheets within cells as well as changes in fluorescence intensity. As the reaction progressed, the fluorescence intensity continuously increased. This phenomenon is consistent with what was observed extracellularly and may be attributed to the further aggregation of STPE-PMNSs. In contrast, the 4-AP untreated group does not exhibit significant changes in fluorescence intensity (Fig.6a).

Fig. 6 Intracellular verification of the catalytic activity of STPE-PMNSs. (a) Intracellular fluorescence signals were observed at different time points.

(Please refer to the manuscript on page 16, lines 7-11)

Furthermore, We examined the toxicity of nanosheets on three types of nerve cells, namely SH-SY5Y, PC-12, and U87, over 7 days. The results reveal that cell viability remained at approximately 80% even after 7 days of exposure to STPE-PMNSs ($200 \mu\text{g mL}^{-1}$) (Supplementary Fig. 55a), indicating excellent biosafety of STPE-PMNSs.

(Please refer to the manuscript on page 16, lines 17-19)

Intracellular Mn accumulation reached its peak at approximately $15.5 \text{ ng}/10^4 \text{ cells}$ on the third day of treatment (Supplementary Fig.55b).

Supplementary Fig. 55 (a) Cell viability of SH-SY5Y, PC-12, U87 cells treated with 25 mg mL^{-1} STPE-PMNSs for 7 d. (b) ICP results of SH-SY5Y, PC-12, U87 cells treated with 25 mg mL^{-1} STPE-PMNSs for 7 d.

Minor concerns and comments:

Query 1

The term “glutamine synthetase” (not “synthase”) should be consistently used throughout the manuscript.

Response

We thank the reviewer for this comment.

The terminology "synthase" has been amended to "synthetase" in the revised manuscript.

Revisions Made

(Please refer to the manuscript on page 1, line 18)

Glutamine synthetase (GS) plays a crucial role in maintaining ammonia and glutamate (Glu) homeostasis in living organisms, which is extensively explored for chemical engineering and biomedical.

(Please refer to the manuscript on page 2, line 11)

Glutamine synthetase (GS) is a crucial enzyme that is ubiquitous in all living organisms.

Query 2

The abbreviation “AIE” needs to be spelled out in the Abstract

Response

We thank the reviewer for this comment.

We have revised the abstract's content, wherein the term AIE is not currently included in the abstract.

Revisions Made

(Please refer to the manuscript on page 2, lines 6-7)

Furthermore, the intracellular level of Gln can be assessed by monitoring alterations in the intracellular fluorescent signal.

Query 3

The relevance of emphasizing ref. 14 and 15 in the Introduction (lines 44-48) are unclear – this discussion should be directly linked to the topic of the manuscript (or removed).

Response

We thank the reviewer for this comment.

We have removed ref. 14-15 in the Introduction.

Revisions Made

(Please refer to the manuscript on page 2, lines 20-22)

Recently, biopolymer-based functional materials have shown great potential for biomimetic catalysis due to their excellent biocompatibility and abundance of editable sites¹⁴⁻¹⁶. These advances have inspired us to develop polyphosphate (polyP) nanotechnology and explore its applications in biomimetic catalysis.

Query 4

The abbreviation “RIM” (line 68) should stand for “restriction of intramolecular motion”

Response

We thank the reviewer for this comment.

We have corrected "intramolecular motion" in line 68 to "restriction of intramolecular motion" in the revised manuscript.

Revisions Made

(Please refer to the manuscript on page 3, lines 18-21)

From an AIE standpoint, nonemissive “free” AIEgens can be induced to undergo fluorescent emission through the restriction of intramolecular motion (RIM) effect associated with aggregation inherent to self-assembly processes of nanostructures^{58, 59}.

Query 5

Discussion in lines 59-71 of the Introduction should describe how AIE would achieve “visualization of biomimetic catalysis processes” (a stated objective for including AIEgens), building on the current understanding that stable fluorescent self-assembled nanomaterials can be prepared using AIEgen components.

Response

We thank the reviewer for this comment.

We have refined the description of how AIEgens enable "visualization of biomimetic

catalysis processes" in the introduction. By employing AIE-active nanostructures as catalysts, it becomes possible to monitor the biochemical reaction progress through the detection of subsequent alterations in emission intensity or wavelength.

Revisions Made

(Please refer to the manuscript on page 3 line 18 to page 4 line 2)

From an AIE standpoint, nonemissive "free" AIEgens can be induced to undergo fluorescent emission through the restriction of intramolecular motion (RIM) effect associated with aggregation inherent to self-assembly processes of nanostructures^{58, 59}. By employing AIE-active nanostructures as catalysts, it becomes possible to monitor the biochemical reaction progress through the detection of subsequent alterations in emission intensity or wavelength.

Query 6

The glutamine synthetase reuse experiment (Fig. 3e) needs a confirmation of the enzyme retention after ultrafiltration, as enzyme loss would also manifest in lower activity.

Response

We thank the reviewer for this comment.

To ascertain the presence of GS residues during the ultrafiltration process, we analyzed the protein concentration (C1) in the reaction system using the BCA method before the initial cycle. Subsequently, GS concentrations (Cn) were measured before each subsequent cycle and compared to C1. The results are depicted in Supplementary Fig. 41. Throughout three cycles, GS concentration was decreased by approximately 7.17%. Hence, the decline in activity attributed to the GS residue was insignificant.

Revisions Made

(Please refer to the manuscript on page 11, lines 1-2)

The GS concentration was assessed using the BCA method before each cycle to confirm that there was no loss of GS after ultrafiltration (Supplementary Fig. 41).

Supplementary Fig. 41 Natural GS concentration before each cycle.

Query 7

High-resolution microscopy images with cell cross-sections are needed to prove the intracellular localization of STPE-PMNSs. Images in Fig. 5a show localization patterns that could result from extracellular binding of nanosheets to cell membranes.

Response

We thank the reviewer for this comment.

We employed high-resolution confocal microscopy to visualize the intracellular distribution of STPE-PMNSs. As depicted in Fig. 6a, the nanosheets were found to be dispersed evenly inside cells.

Revisions Made

(Please refer to the manuscript on page 17, Fig. 6a)

Fig. 6 (a) Intracellular fluorescence signals were observed at different time points.

Query 8

Further, it would be helpful to discuss the mechanism of PMNS intracellular uptake and

distribution (e.g., lack of endosomal localization).

Response

We thank the reviewer for this comment.

There are numerous studies have indicated that nanomaterials can interact with cell membranes and enter cells through a process known as "endocytosis"⁴⁻⁷. To confirm whether STPE-PMNSs enter cells via the process of endocytosis, we employed bio-transmission electron microscopy (Bio-TEM) to observe cells treated with STPE-PMNSs. The results revealed that the nanosheets were enclosed within vesicles and transported intracellularly following treatment with STPE-PMNSs (Supplementary Fig. 57). Furthermore, we investigated the uptake of STPE-PMNSs by cells under both starvation and normal conditions. Our findings indicate that after starvation for 12 h, STPE-PMNSs treated cells lack uptake of these nanosheets (Supplementary Fig. 58a and b). In contrast, following 12 h of standard cell culture, the cells treat with STPE-PMNSs in a serum-free medium exhibited normal uptake of the material, but the intracellular ATP levels gradually decreased (Supplementary Fig. 58c-d and Supplementary Fig. 59). These findings indicate that the uptake of the STPE-PMNSs requires energy expenditure. As a result, we have deduced that STPE-PMNSs are internalized by cells through the process of endocytosis.

Subsequently, we investigated the ultimate subcellular localization of STPE-PMNS. We labeled various organelles (lysosomes, endoplasmic reticulum, mitochondria, and Golgi apparatus) and then utilized confocal microscopy to visualize the subcellular distribution of STPE-PMNS. These results revealed a significant overlap between the green fluorescence emitted by STPE-PMNSs and the red fluorescence emitted by lysosomes, suggesting that the primary localization of STPE-PMNSs within cells is within the lysosomes (Supplementary Fig. 60).

Revisions Made

(Please refer to the manuscript on page 16, lines 19-22)

Analysis of the mechanism of STPE-PMNS intracellular uptake and distribution reveals that STPE-PMNSs are internalized by cells through the process of endocytosis and the primary localization is the lysosomes (Supplementary Figs. 57-60).

Supplementary Fig. 57 Bio-TEM images of SH-SY5Y cells treated (a) and untreated (b) with STPE-PMNSs.

Supplementary Fig. 58 After starvation for 12 h, fluorescence intensity (a) and Mn content (b) of the cells treated with STPE-PMNSs at different time points. After normal culture 12 h, fluorescence intensity (c) and Mn content (d) of the cells treated with STPE-PMNS at different time points in a serum-free medium.

Supplementary Fig. 59 Intracellular ATP levels of cells treated with STPE-PMNSs at different time points in a serum-free medium.

Supplementary Fig. 60 The subcellular distribution of STPE-PMNS.

Reference

1. LISTROM, D.C. *et al.* Expression, purification, and characterization of recombinant human glutamine synthetase. *Biochem. J.* **328**, 159-163 (1997).
2. Mo, J., Xu, Y., Zhu, L., Wei, W. & Zhao, J. A Cysteine-Mediated Synthesis of Red Phosphorus Nanosheets. *Angew. Chem. Int. Ed.* **60**, 12524-12531 (2021).
3. Gao, M. *et al.* Two-Dimensional Tin Selenide (SnSe) Nanosheets Capable of Mimicking Key Dehydrogenases in Cellular Metabolism. *Angew. Chem. Int. Ed.* **59**, 3618-3623 (2020).
4. Nazemidashtarjandi, S., Sharma, V.M., Puri, V., Farnoud, A.M. & Burdick, M.M. Lipid Composition of the Cell Membrane Outer Leaflet Regulates Endocytosis of Nanomaterials through Alterations in Scavenger Receptor Activity. *ACS Nano* **16**, 2233-2248 (2022).
5. Mao, J., Chen, P., Liang, J., Guo, R. & Yan, L.-T. Receptor-Mediated Endocytosis of Two-Dimensional Nanomaterials Undergoes Flat Vesiculation and Occurs by Revolution and Self-Rotation. *ACS Nano* **10**, 1493-1502 (2016).
6. Iturrioz-Rodríguez, N. *et al.* A Biomimetic Escape Strategy for Cytoplasm Invasion by Synthetic Particles. *Angew. Chem. Int. Ed.* **56**, 13736-13740 (2017).
7. Makvandi, P. *et al.* Endocytosis of abiotic nanomaterials and nanobiovectors: Inhibition of membrane trafficking. *Nano Today* **40**, 101279 (2021).

Reviewers' Comments:

Reviewer #1:

Remarks to the Author:

The authors had made a point-to-point reply to the reviewer's comments, and further supplemented the experimental data and proposed a new catalytic mechanism to improve the quality of manuscripts. However, the authors' response and the supplementary experimental data give rise to the following confusion and contradictions.

1. In general, in the absence of much clearer and adequate crystal characterization of the structure, it is not easy to infer the specific atomic structure configuration of the materials we synthesize based on the oxidation state and proportion of the elements. The theoretical calculation model in this paper cannot adequately represent the synthesized materials.
2. According to the newly proposed reaction mechanism and all the intermediate structures provided by the author, the material STPE-PMNSs in this paper plays the role of providing phosphate group to activate glutamate, which is like to the function of ATP in the natural GS catalytic reaction. From the traditional catalytic chemical point of view, STPE-PMNSs is consumed during the reaction, and the reaction cannot be called a catalytic reaction. STPE-PMNSs cannot be called artificial glutamine synthetase as the authors claimed.
3. For STPE-PMNSs involved in the process of converting glutamate into glutamine, the new supplementary data provided by the authors also confirmed that phosphate ions were leached out in the absence of ATP (Supplementary Fig. 49, 50 and 51), so STPE-PMNSs was consumed when no ATP was involved in the reaction. However, STPE-PMNSs are not consumed until ATP provides the function of phosphate to activate reactants. It was further demonstrated that STPE-PMNSs participated in the reaction in the absence of ATP, rather than being responsible for catalyzing the reaction. The authors should not claim that the material can be an ATP-free artificial glutamine synthetase.
4. Supplementary Fig. 32, after the reaction with the absence of ATP, the material STPE-PMNSs agglomerated. From the view of all the provided data, it is possible that the phosphate bonds in the material further polymerize resulting in material agglomeration to maintain the coordination stability of Mn²⁺ after the leaching of phosphate ions. In this case, the valence state of Mn remains the same, and the P-O and Mn-O ratios do not change much. But this does not support the argument that the physicochemical properties of the material do not change after polymerization.

Reviewer #3:

Remarks to the Author:

The paper by Zhao et al was extensively revised in response to the comments of the reviewers. The new version of the paper includes a substantial amount of new information. I feel that all in all the authors have addressed all key issues in a satisfactory way.

Reviewer #4:

Remarks to the Author:

The revised manuscript provides a more thorough characterization of STPE-PMNSs and a plausible (though incomplete) mechanism of catalytic activity. Authors have made substantial revisions to the discussions and produced additional supporting data in response to reviewers' comments.

Notably, a role of polyP as a source of energy/phosphate for the reaction has been hypothesized, potentially explaining reaction-induced aggregation of PMNSs and addressing several mechanism-related questions. However, this hypothesis now reveals that STPE-PMNSs participate in the reaction as reactants (with phosphates being consumed) and not solely as catalysts. This presents a fundamental difference from natural glutamine synthetase, and, therefore, calls for an in-depth discussion. Related to this point:

1. limitations associated with polyP degradation during reaction need to be explored/discussed (such as expected duration/capacity of catalytic activity; need for re-treatment; fate of aggregated

STPE-PMNSs in cells)

2. term "artificial glutamine synthetase" might be misleading (as it implies pure enzyme-like activity), and it needs to be clarified to include polyP role as a phosphate donor (as well as ammonium activator proposed in the reaction mechanism)
3. similarly, claim that polyP-based nanosheets can be used as alternative to natural GS needs to come with a note of transient activity (due to consumption of polyP and aggregation)
4. it is unclear why intracellular ATP does not prevent reaction with polyP and aggregation inside cells, unlike the behavior shown in Supp. Fig. 46-47 – potential reasons for the lack of ATP activity need to be discussed.

Additional points for the authors to consider:

1. It might be helpful to clearly separate the 2 uses of AIE signal: 1) AIE obtained from nanosheet assembly, which can be used for tracking intracellular uptake and localization vs. 2) further reaction-induced increase in AIE signal, which can be used to monitor reaction progression (though only in cases when reaction triggers further aggregation of nanosheets).
2. A positive charge is claimed to be essential for catalytic activity, but it is not clear where the positive charge in STPE-PMNSs comes from. Does CTAB play a role in this (if retained on the surface)? Do unreacted amine groups on polyP-NH₂ remain after STPE labeling? How does this compare to unlabeled PMNSs (which show similar catalytic activity)?
3. The proposed reaction mechanism heavily relies on terminal phosphates for both glutamate activation and ammonium activation. However, amine modification and STPE labeling would presumably cap majority of terminal phosphates, and yet STPE-PMNSs exhibit catalytic activity similar to that of unmodified PMNSs. This discrepancy needs to be addressed, or further revisions to the mechanism need to be considered.
4. Data presented in Supp. Fig. 57-60 and conclusions need a detailed description in text. Of note, such a description has been provided in response to reviewer's comment, but not included in the revised manuscript.

Reviewer #5:

Remarks to the Author:

[Note from the Editor: Reviewer 5 was asked to review the response given to the original Reviewer 2.]

The authors addressed all issues from reviewer 2.

However, in my view although this is an interesting piece of work, there are obvious problems with the manuscript.

1. The structural characterization is seriously lacking and the actual structure is uncertain.
2. The role of AIE molecules here is not well explained.
3. Lack of research on catalytic mechanism.

In view of this, it is not recommended for publication.

Point-to-point responses to reviewers' comments

Reviewer 1

The authors had made a point-to-point reply to the reviewer's comments, and further supplemented the experimental data and proposed a new catalytic mechanism to improve the quality of manuscripts. However, the authors' response and the supplementary experimental data give rise to the following confusion and contradictions.

Query 1

In general, in the absence of much clearer and adequate crystal characterization of the structure, it is not easy to infer the specific atomic structure configuration of the materials we synthesize based on the oxidation state and proportion of the elements. The theoretical calculation model in this paper cannot adequately represent the synthesized materials.

Response

We thank the reviewer for this comment. To enhance the current understanding of the structural properties of STPE-PMNSs, we have conducted supplementary analyses that elucidate the structural features of these materials and provide further validation for the reliability of our DFT calculations. Specifically, we utilized spherical aberration-corrected electron microscopy (AC-STEM, 80 keV) to characterize the nanosheets. The resulting selection-area intensity surface plot (Fig. 2b, shown below) revealed interatomic distances between two Mn atoms in the transverse and longitudinal orientations to be 3.57 Å and 5.25 Å, respectively. These values are in good agreement with those obtained from DFT calculations (transverse: 3.25 Å; longitudinal: 5.14 Å).

To provide additional validation for our theoretical model, we exposed the synthesized STPE-PMNSs to a 24-hour treatment at 150°C within a Teflon-lined stainless-steel autoclave. The subsequent powder X-ray diffraction (PXRD) pattern displayed diffraction peaks that were in close agreement with the predictions from our theoretical model (Supplementary Fig. 13a), thereby demonstrating the reliability of our calculations in representing the synthesized materials.

Moreover, we conducted FT-IR and Raman spectroscopy analyses to investigate the

characteristic peaks of the phosphate groups in the synthesized STPE-PMNSs (Supplementary Fig. 15). The obtained spectra, in conjunction with XPS analysis of the element valence state and atomic ratio (Fig. 2e-g and Supplementary Fig. 16), further support the consistency between the structure of STPE-PMNSs and our DFT calculations.

In summary, our comprehensive experimental characterization, including AC-STEM, PXRD, FT-IR, Raman spectroscopy, and XPS, provides compelling evidence that the theoretical calculation model presented in this paper can adequately represent the synthesized polymeric manganese phosphate nanosheets.

Fig. 2 (b) An HAADF-STEM image of STPE-PMNSs (left) and intensity surface plot from yellow line rectangle (middle and right).

Supplementary Fig. 13 (a) XRD results of the STPE-PMNSs.

Supplementary Fig. 15 (a) Comparison of Fourier transforms infrared (FT-IR) spectroscopy assay between PMNSs and STPE-PMNSs. (b) Raman analysis of STPE-PMNSs.

Fig. 2 XPS analysis for O 1s spectra (e) Mn 2p spectra (f), and atomic percentage (g) of STPE-PMNSs.

Supplementary Fig. 16 XPS analysis for C 1s (a), S 2p (b), P 2p (c), N 1s (d) of STPE-PMNSs.

Query 2

According to the newly proposed reaction mechanism and all the intermediate structures provided by the author, the material STPE-PMNSs in this paper plays the role of providing a phosphate group to activate glutamate, which is like to the function of ATP in the natural GS catalytic reaction. From the traditional catalytic chemical point of view, STPE-PMNSs are consumed during the reaction, and the reaction cannot be called a catalytic reaction. STPE-PMNSs cannot be called artificial glutamine synthetase as the authors claimed.

Response

We appreciate the reviewer's concerns regarding calling the STPE-PMNSs a catalyst when it was partially consumed under certain conditions. We apologize for any confusion in our previous submission. The role of STPE-PMNSs appears to vary under different reaction conditions.

As astutely noted by the reviewer, the polyP in nanosheets can function as a

phosphate source in the absence of ATP. This role is evident from Fig. 4b, which shows a gradual decrease in phosphorus content within the nanosheets over time, indicating phosphate leaching from polyP. Additionally, significant aggregation of STPE-PMNSs was observed post-reaction, supporting a change of structure (Fig. 4e). These findings suggest that the polyP component of STPE-PMNSs serves as the phosphate source when there are no external phosphates added (i.e. ATP).

Without STPE-PMNSs, ATP and ammonia by themselves were unable to convert Glu to Gln (Fig. 4g), suggesting that STPE-PMNSs not only serve as a phosphate source but also contribute additional catalysis. We propose that the Mn ions within STPE-PMNSs display catalytic functions similar to their counterparts in natural GS. This is achieved by binding to the negatively charged Glu and reducing the free energy barriers associated with phosphate transfer from polyP, as well as the subsequent aminolysis.

The reviewer was right, due to the consumption of the phosphate group in the absence of ATP, STPE-PMNSs do not strictly fit the definition of a catalyst, but rather, function as a reagent with extra catalytic features (Mn^{2+}). We have removed the term "artificial GS" in our manuscript to more accurately reflect the role of STPE-PMNSs in the absence of ATP.

In stark contrast, we observed that STPE-PMNSs exhibited GS-like catalytic activity when the reaction was conducted in the presence of ATP. Interestingly, the phosphorus content remained stable over time in this scenario, as opposed to when ATP was absent (Fig. 4b). The inclusion of ATP also enhanced the recyclability of STPE-PMNSs. TEM images revealed that the STPE-PMNSs maintained their structure post-reaction, signifying their catalytic nature when ATP is present (Fig. 4d). Under conditions of a high ATP concentration, ATP serves as the primary phosphate source, and STPE-PMNSs act as a catalyst for the reaction. Conversely, when ATP levels are low or nonexistent, the polyP within the nanosheets becomes the essential phosphate source for Glu activation (Fig. 4h).

Fig. 4 (b) Phosphorus (P) content of STPE-PMNSs with (20 mM) or without ATP.

Fig. 4 TEM image of STPE-PMNSs after 2 h with (d) or without (e) ATP.

Fig. 4 (g) Reactions were conducted to determine whether STPE-PMNSs act solely as reaction substrates or possess catalytic activity in the absence of ATP. In the absence of STPE-PMNSs, ATP (20 mM) and ammonia alone could not convert Glu to Gln, while the reaction proceeded normally with STPE-PMNSs present.

Fig. 4 (h) Correlation between fluorescence intensity and ATP concentration in the reaction revealed a linear relationship in the low-concentration region (inset). At a low ATP level, polyP in the nanosheets activates Glu, with the fluorescence intensity at its highest level when there is no ATP. As ATP concentration increases, ATP gradually replaces polyP in activating Glu, causing the fluorescence intensity to decrease. Once ATP fully replaces polyP, the fluorescence intensity stabilizes and no longer declines.

Query 3

For STPE-PMNSs involved in the process of converting glutamate into glutamine, the new supplementary data provided by the authors also confirmed that phosphate ions were leached out in the absence of ATP (Supplementary Fig. 49, 50 and 51), so STPE-PMNSs was consumed when no ATP was involved in the reaction. However, STPE-PMNSs are not consumed until ATP provides the function of phosphate to activate reactants. It was further demonstrated that STPE-PMNs participated in the reaction in the absence of ATP, rather than being responsible for catalyzing the reaction. The authors should not claim that the material can be an ATP-free artificial glutamine synthetase.

Response

We appreciate the reviewer's accurate summary of the role of STPE-PMNs in reactions without ATP. Indeed, STPE-PMNs provide the essential phosphate in this scenario and cannot be called an ATP-free artificial GS. We apologize for the confusion caused by the previous submission. We have removed the claim of STPE-PMNSs being

an ATP-free artificial glutamine synthetase in the resubmitted manuscript.

Query 4

Supplementary Fig. 32, after the reaction with the absence of ATP, the material STPE-PMNSs agglomerated. From the view of all the provided data, it is possible that the phosphate bonds in the material further polymerize resulting in material agglomeration to maintain the coordination stability of Mn^{2+} after the leaching of phosphate ions. In this case, the valence state of Mn remains the same, and the P-O and Mn-O ratios do not change much. But this does not support the argument that the physicochemical properties of the material do not change after polymerization.

Response

We fully agree with the reviewer that in the absence of ATP, STPE-PMNSs indeed change due to phosphate leaching. The corresponding discussions have been modified. In the resubmitted manuscript, we discuss the physicochemical properties of STPE-PMNSs before and after the reaction in the following two scenarios:

In the absence of ATP, the consumption of phosphate groups leads to altered physicochemical properties of the STPE-PMNSs pre- and post-reaction (Figs. 4d-e, Supplementary Figs. 56b-57b, and Supplementary Fig. 58c-d). The reviewer's proposal that polymerization of the remaining phosphates to maintain the coordination stability of Mn^{2+} after the leaching of phosphate ions is very reasonable.

Conversely, in the presence of ATP, the nanosheets maintain consistency before and after the reaction, with no alteration in the physicochemical properties of the nanosheets (Fig. 4d, Supplementary Figs. 56a-57a, and Supplementary Fig. 58a-b). This finding supports our argument that the material remains unchanged when ATP is present in the reaction.

We have revised our manuscript to clarify these points and address the reviewer's concerns. We hope that this explanation better demonstrates the role of STPE-PMNSs in the reaction and their physicochemical property changes depending on the presence or absence of ATP.

Fig. 4 TEM image of STPE-PMNSs after 2 h with (d) or without (e) ATP.

Supplementary Fig. 56 XPS analysis of post-reaction STPE-PMNSs after reaction 2 h in the presence (a) or absence (b) of ATP.

Supplementary Fig. 57 The atomic percentage from XPS of STPE-PMNSs in the presence (a) or absence (b) of ATP.

Supplementary Fig. 58 STEM images and elemental mapping for STPE-PMNSs after reaction 2 h in the presence (a-b) or absence (c-d) of ATP. Scale bar: 30 nm.

Reviewer 3

The paper by Zhao et al was extensively revised in response to the comments of the reviewers. The new version of the paper includes a substantial amount of new information. I feel that all in all the authors have addressed all key issues in a satisfactory way.

Response

We thank reviewer 3's kind comments on our previous revision.

Reviewer 4

The revised manuscript provides a more thorough characterization of STPE-PMNSs and a plausible (though incomplete) mechanism of catalytic activity. Authors have made substantial revisions to the discussions and produced additional supporting data in response to reviewers' comments.

Notably, a role of polyP as a source of energy/phosphate for the reaction has been hypothesized, potentially explaining reaction-induced aggregation of PMNSs and addressing several mechanism-related questions. However, this hypothesis now reveals that STPE-PMNSs participate in the reaction as reactants (with phosphates being consumed) and not solely as catalysts. This presents a fundamental difference from natural glutamine synthetase, and, therefore, calls for an in-depth discussion. Related to this point:

Response

We appreciate the reviewer for bringing up this crucial concern, which resonates with inquiries from other reviewers. We have significantly revised the discussion surrounding the role of STPE-PMNSs to more accurately depict their function under varying conditions. Please see the responses below and to other related questions (reviewer 1, Q2, and Q3;) for details.

Query 1

Limitations associated with polyP degradation during reaction need to be explored/discussed (such as expected duration/capacity of catalytic activity; need for re-treatment; fate of aggregated STPE-PMNSs in cells)

Response

In response to the reviewer's question regarding the limitations associated with polyP degradation during the reaction, we have conducted additional experiments and provided a more comprehensive discussion on this topic.

As suggested by the reviewer, we explored the length of time for which STPE-PMNSs can sustain activity in the extracellular environment. Our results demonstrate

that the sustainable activity durations of STPE-PMNSs are 60 hours, 48 hours, 48 hours, and 36 hours at Glu concentrations of 2 mM, 10 mM, 20 mM, and 100 mM, respectively. This indicates that the activity durations of STPE-PMNSs diminish as the Glu concentration rises, necessitating a continuous supply of STPE-PMNSs to maintain extended activity (Fig. 4i). Moreover, we discussed the fate of aggregated STPE-PMNSs during intracellular reaction. We found that a portion of the STPE-PMNSs was consumed as the reaction advanced, while the remaining nanosheets were transported to lysosomes. Within the lysosomes, the STPE-PMNSs undergo digestion and degradation. It is important to note that the consumption of STPE-PMNSs in the extracellular environment is attributable to inadequate ATP concentration, and the nanosheets serve as the phosphate source in this case.

We hope that this revised response addresses the reviewer's concerns and provides a clearer understanding of the limitations and considerations associated with polyP degradation reactions involving STPE-PMNSs.

Fig. 4 (i) Conversion duration of STPE-PMNSs at varying Glu concentrations without ATP.

Query 2

Term “artificial glutamine synthetase” might be misleading (as it implies pure enzyme-like activity), and it needs to be clarified to include polyP role as a phosphate donor (as well as ammonium activator proposed in the reaction mechanism)

Response

We agree with the reviewer on this matter and have revised the description of

"artificial glutamine synthetase" in the manuscript to ensure a more accurate representation. Considering the consumption of the phosphate group in the absence of ATP, STPE-PMNSs are not a catalyst. Instead, they function as a reagent with additional catalytic features (Mn^{2+}). We have explicitly described the role of polyP as a phosphate donor in the absence of ATP in the resubmitted draft.

On the other hand, the GS-like catalytic activity of STPE-PMNSs was observed when the reaction was conducted in the presence of ATP. The phosphorus content remained stable over time in this case, as opposed to when ATP was absent. The inclusion of ATP also enhanced the recyclability of STPE-PMNSs, as the phosphorous content did not decrease over time, unlike when ATP was absent (Fig. 4b). TEM images revealed that the STPE-PMNSs maintained their structure post-reaction, signifying their catalytic nature when ATP is present (Fig. 4e). Under conditions of a high ATP concentration, ATP serves as the primary phosphate source, and STPE-PMNSs act as a catalyst for the reaction. Conversely, when ATP levels are low or nonexistent, the polyP within the nanosheets becomes the essential phosphate source for Glu activation (Fig. 4h).

Fig. 4 (b) Phosphorus (P) content of STPE-PMNSs with (20 mM) or without ATP.

Fig. 4 TEM image of STPE-PMNSs after 2 h with (d) or without (e) ATP.

Fig. 4 (h) Correlation between fluorescence intensity and ATP concentration in the reaction revealed a linear relationship in the low-concentration region (inset). At a low ATP level, polyP in the nanosheets activates Glu, with the fluorescence intensity at its highest level when there is no ATP. As ATP concentration increases, ATP gradually replaces polyP in activating Glu, causing the fluorescence intensity to decrease. Once ATP fully replaces polyP, the fluorescence intensity stabilizes and no longer declines.

Query 3

Similarly, claim that polyP-based nanosheets can be used as alternative to natural GS needs to come with a note of transient activity (due to consumption of polyP and aggregation)

Response

In response to the reviewer's concern, we have discussed the transient nature of the nanosheet's activity due to the consumption of polyP and potential aggregation. Furthermore, the additional experiments we conducted have provided more insight into the duration of the activity of the nanosheets.

As suggested by the reviewer, we explored the length of time for which STPE-PMNSs can sustain activity in the extracellular environment. Our results demonstrate that the sustainable catalytic durations of STPE-PMNSs are 60 hours, 48 hours, 48 hours, and 36 hours at Glu concentrations of 2 mM, 10 mM, 20 mM, and 100 mM, respectively (Fig. 4i). This indicates that the activity durations of STPE-PMNSs diminish as the Glu concentration rises, necessitating a continuous supply of STPE-PMNSs to maintain extended activity. The manuscript has been revised to incorporate these comments.

Fig. 4 (i) Conversion duration of STPE-PMNSs at varying Glu concentrations without ATP.

Query 4

It is unclear why intracellular ATP does not prevent reaction with polyP and aggregation inside cells, unlike the behavior shown in Supp. Fig. 46-47 – potential reasons for the lack of ATP activity need to be discussed.

Response

We appreciate the reviewer's concern about the lack of ATP activity as the phosphate source and not preventing aggregation of STPE-PMNSs inside cells. We hypothesize that this may be due to the reduced concentration of ATP during excitotoxicity. Excitotoxicity is known to be primarily initiated by sustained stimulation of ionotropic Glu receptors, leading to an increase in intracellular Ca^{2+} levels, followed by a cascade of intracellular events, such as mitochondrial depolarization and ATP

depletion^{1,2}.

The ATP level was measured in nerve cells treated with 4-AP (induces SH-SY5Y cells to produce Glu) and a significant decrease in ATP level (6.73 μM) was observed during excitotoxic events (Supplementary Fig. 69). Furthermore, we examined the activation of Glu by polyP under various ATP concentrations (Fig. 4h). In the 0-10 μM range, Glu is predominantly activated by polyP present in the nanosheets. ATP begins to supply phosphate at concentrations above 10 μM . This finding suggests that nanosheets persist in aggregating intracellularly due to phosphate leaching when ATP levels are low.

Supplementary Fig. 69 ATP content after 4-AP (2 mM) was stimulated.

Fig. 4 (h) Correlation between fluorescence intensity and ATP concentration in the reaction revealed a linear relationship in the low-concentration region (inset). At a low ATP level, polyP in the nanosheets activates Glu, with the fluorescence intensity at its highest level when there is no ATP. As ATP concentration increases, ATP gradually replaces polyP in activating Glu, causing the fluorescence intensity to decrease. Once

ATP fully replaces polyP, the fluorescence intensity stabilizes and no longer declines.

Query 5

It might be helpful to clearly separate the 2 uses of AIE signal: 1) AIE obtained from nanosheet assembly, which can be used for tracking intracellular uptake and localization vs. 2) further reaction-induced increase in AIE signal, which can be used to monitor reaction progression (though only in cases when reaction triggers further aggregation of nanosheets).

Response

We appreciate the reviewer's insightful suggestion. In the resubmitted manuscript, we have clearly distinguished the two roles of the AIE signal (**Please refer to page 2, lines 4-8, and page 4, lines 20-22**). First, the AIE signal originating from the assembly of nanosheets allows us to track the intracellular uptake and localization of STPE-PMNSs. Second, the reaction-induced increase in the AIE signal, triggered by the activation of Glu by polyP within the nanosheets, leads to further aggregation of the nanosheets and an amplified AIE signal. This enhanced fluorescence signal serves as a valuable tool for monitoring the catalytic reaction's progression within cells. We have made these distinctions more explicit in the manuscript to avoid confusion.

Query 6

A positive charge is claimed to be essential for catalytic activity, but it is not clear where the positive charge in STPE-PMNSs comes from. Does CTAB play a role in this (if retained on the surface)? Do unreacted amine groups on polyP-NH₂ remain after STPE labeling? How does this compare to unlabeled PMNSs (which show similar catalytic activity)?

Response

We thank the reviewer for this question and apologize for not being clear in the previous submission. The positive charges referred to Mn²⁺ ions in both STPE-PMNSs and PMNSs. The FT-IR results show no obvious characteristic peaks of CTAB (C-H, 3000-2700 cm⁻¹) in PMNSs within the high wavenumber range (Supplementary Fig.

15a), indicating that the residual presence of CTAB in STPE-PMNSs synthesized under the same conditions is minimal.

NMR data reveal that the terminal labeling of organic amines has an impressive modification rate of 99.2%, while the subsequent modification rate of STPE is 47.06% (Supplementary Fig. 2 and Supplementary Fig. 10). This finding implies that nearly half of the polyP-NH₂ remains unlabeled. These unlabeled polyP-NH₂ groups participate in the assembly of nanosheets alongside polyP-STPE. Additionally, we compared the surface properties of STPE-PMNSs and PMNSs and found no significant difference in their zeta potentials (Supplementary Fig. 14).

Supplementary Fig. 15 (a) Comparison of Fourier transforms infrared (FT-IR) spectroscopy assay between PMNSs and STPE-PMNSs.

Supplementary Fig. 2 ^{31}P NMR spectrum of polyP-NH₂.

Supplementary Fig. 10 ^1H NMR spectrum of STPE-polyP.

Supplementary Fig. 14 Comparison of zeta potential between PMNSs and STPE-PMNSs.

Query 7

The proposed reaction mechanism heavily relies on terminal phosphates for both glutamate activation and ammonium activation. However, amine modification and STPE labeling would presumably cap the majority of terminal phosphates, and yet STPE-PMNSs exhibit catalytic activity similar to that of unmodified PMNSs. This discrepancy needs to be addressed, or further revisions to the mechanism need to be considered.

Response

We thank the reviewer for this constructive comment. We have revised the catalytic mechanism of STPE-PMNSs based on the reviewer's feedback and additional experimental results. Our NMR data reveals that the terminal labeling of organic amines has an impressive modification rate of 99.2%, while the subsequent modification rate of STPE is 47.06% (Supplementary Fig. 2 and Supplementary Fig. 10). This finding implies that nearly 50% of the polyP-NH₂ remains unlabeled. Considering this result, we revised the reaction mechanism:

In general, the reaction proceeds in two stages. The initial stage involves the activation of reactants. In the presence of ATP, Glu and ATP interact with the Mn²⁺ on the surface of the nanosheets. Following this interaction, Glu is phosphorylated by ATP, resulting in the formation of activated γ -glutamyl phosphate and ADP with a free energy of -1.01 eV (Fig. 5a). In the absence of ATP, Glu adsorbed on the nanosheets react with

the terminal phosphate group, generating γ -glutamyl phosphate polyoxyethylene bis (amine) with a free energy of -0.12 eV (Fig. 5f). This event also generates new unlabelled terminal phosphates. The phosphorylation of Glu activates the substrate for the subsequent aminolysis. The role of Mn^{2+} within the STPE-PMNSs active site is essential, as it serves both structural and catalytic functions by stabilizing the negative charges of the reactants and activating the substrate. It is important to note that the considerable steric hindrance of the STPE group discourages interactions between Glu and adjacent Mn atoms. Consequently, adsorbed Glu is unable to attack polyP-STPE.

The second step corresponds to Gln formation, via a nucleophilic attack of ammonia to the $C\gamma$ of the γ -glutamyl phosphate intermediate. There are two basic sites on the nanosheets that can potentially activate NH_4^+ for aminolysis (Figs. 5b-c and 5e). One is the unlabeled polyP-NH₂, and the other is the newly exposed terminal phosphate after Glu phosphorylation. The conversion of NH_4^+ to NH_3 is essential as the ammonium cation is not a nucleophile. The formation of the $C\gamma$ -amide bond is believed to rate-limiting (Figs. 5a and 5g). The energy profile of the overall reaction is shown in Figs. 5d and 5h. The total energy of the reaction in the presence or absence of ATP is -4.6 and -0.77 eV respectively.

Overall, we believe that there are two main reasons for the similar activity between STPE-PMNSs and PMNSs:

1. The modification of polyoxyethylene bis (amine) does not affect the activation of Glu. Glu adsorbed on the nanosheet reacts with the terminal phosphate group to generate γ -glutamyl phosphate polyoxyethylene bis(amine).
2. Although the end-capping of STPE leads to a reduction in the active sites of NH_4^+ , phosphorylation of Glu exposes new phosphate groups, which can also activate NH_4^+ . Furthermore, the attack of NH_3 on $C\gamma$ is the rate-determining step of the reaction, and in the presence of sufficient STPE-PMNSs, there is little difference in activity between STPE-PMNSs and PMNSs.

Fig. 5 Proposed reaction pathway and energies of incremental steps by DFT. (a) The proposed catalytic cycle of the GS-like activity of STPE-PMNSs in the presence of ATP. (b) The activation of NH_4^+ in the presence of ATP. (c) The energy profile for the NH_4^+ activation step. (d) STPE-PMNSs catalyzed conversion of Glu to Gln in the presence of ATP. (e) The energy profile for the activation of NH_4^+ after the terminal polyoxyethylene bis(amine) phosphate is detached. The activation of Glu (f) and Gln formation route (g) using polyP as the phosphate source. (h) The energy profile for STPE-PMNSs promoted the conversion of Glu to Gln in the absence of ATP.

Query 8

Data presented in Supp. Fig. 57-60 and conclusions need a detailed description in text.

Of note, such a description has been provided in response to the reviewer's comment, but not included in the revised manuscript.

Response

We thank the reviewer for this comment. We have added a detailed description of Supplementary Fig.57-60 in the text (**Please refer to the manuscript on page 18 line 15 to page 19 line 6**).

Reviewer 5

The authors addressed all issues from reviewer 2.

However, in my view although this is an interesting piece of work, there are obvious problems with the manuscript.

Query 1

The structural characterization is seriously lacking and the actual structure is uncertain.

Response

To enhance the current understanding of the structural properties of STPE-PMNSs, we have conducted supplementary analyses that elucidate the structural features of these materials and provide further validation for the reliability of our DFT calculations. We employed aberration-corrected scanning transmission electron microscopy (AC-STEM, 80 keV) for characterizing the nanosheets. The selection-area intensity surface plot (Fig. 2b) reveals interatomic distances between two Mn atoms in transverse and longitudinal orientations as 3.57 Å and 5.25 Å, respectively. These findings align with our density functional theory (DFT) calculations model (transverse: 3.25 Å; longitudinal: 5.14 Å).

The synthesized STPE-PMNSs were subjected to a 24-hour treatment at 150°C in a Teflon-lined stainless-steel autoclave. The PXRD pattern displayed diffraction peaks that closely matched those predicted by the theoretical model, confirming the accuracy of our calculation model in representing the synthesized materials (Supplementary Fig. 13a). We also employed Raman, Fourier-transform infrared (FT-IR), and X-ray photoelectron spectroscopy (XPS) to elucidate the structure of STPE-PMNSs.

Raman analysis revealed three major bands at approximately 717.0, 616.8, 566.2, and 455 cm⁻¹, corresponding to Mn-O symmetric stretching. The peak at 944.0 cm⁻¹ is attributed to intramolecular symmetric vibrations of PO₄³⁻, whereas the weaker peaks at 891.9, 980.1, and 1023.7 cm⁻¹ are associated with the bending and asymmetric stretching of PO₄³⁻ (Supplementary Fig. 15b). The FT-IR spectrum showed peaks centered at 1088, 867, and 988 cm⁻¹, corresponding to asymmetric stretching of ν_{as} (P-O) (Supplementary Fig. 15a). XPS analysis of Mn 2p indicated that the predominant oxidation state of Mn is Mn (II). Notably, the O 1s XPS spectrum revealed an Mn-O to

P-O bond ratio of approximately 1:1 (Figs. 2e-g and Supplementary Fig. 16), which is consistent with the configuration used in our DFT calculations.

We hope that the comprehensive experimental characterization, including AC-STEM, PXRD, FT-IR, Raman spectroscopy, and XPS, provides compelling evidence that the theoretical calculation model presented in this paper can adequately represent the synthesized polymeric manganese phosphate nanosheets.

Fig.2 (b) An HAADF-STEM image of STPE-PMNSs (left) and intensity surface plot from yellow line rectangle (middle and right).

Supplementary Fig. 13 (a) XRD results of the STPE-PMNSs.

Supplementary Fig. 15 (a) Comparison of Fourier transforms infrared (FT-IR)

spectroscopy assay between PMNSs and STPE-PMNSs. (d) Raman analysis of STPE-PMNSs.

Fig. 2 XPS analysis for O 1s spectra (e) Mn 2p spectra (f), and atomic percentage (g) of STPE-PMNSs.

Supplementary Fig. 16 XPS analysis for C 1s (a), S 2p (b), P 2p (c), N 1s (d) of STPE-PMNSs.

Query 2

The role of AIE molecules here is not well explained.

Response

In the resubmitted manuscript, we have clearly distinguished the two roles of the AIE signal. First, the AIE signal originating from the assembly of nanosheets allows us to track the intracellular uptake and localization of STPE-PMNSs. Second, the reaction-induced increase in the AIE signal, triggered by the activation of Glu by polyP within the nanosheets, leads to further aggregation of the nanosheets and an amplified AIE signal. This enhanced fluorescence signal serves as a valuable tool for monitoring the catalytic reaction's progression within cells. We have made these distinctions more explicit in the manuscript to avoid confusion.

Supplementary Fig. 64 Internalization of STPE-PMNSs by cells at specific time intervals. (a) fluorescence signal changes within the cells over various periods.

Supplementary Fig. 68 The subcellular distribution of STPE-PMNS.

Fig. 6 Assessment of the intracellular GS-like activity of STPE-PMNS. Intracellular fluorescence signals were monitored at different time points (a), followed by relative quantification of the respective fluorescent signals (b). Cells underwent treatment with STPE-PMNSs for 6 h, followed by PBS washing. Glu production was

induced in SH-SY5Y cells using 4-AP for 4 h, and cells were washed with PBS again. Intracellular fluorescence signals were subsequently observed at different time points, with corresponding signal quantification. Changes in intracellular Gln content were recorded at various time points (c), along with the linear relationship between Gln content and fluorescence intensity (d).

Query 3

Lack of research on catalytic mechanism.

Response

We thank the reviewer for this comment.

Based on the reviewers' suggestions and additional experimental results, we modified the reaction mechanism with detailed discussions. The role of STPE-PMNSs appears to vary under different reaction conditions.

The polyP can function as a phosphate source in the absence of ATP. This role is evident from Fig. 4b, which shows a gradual decrease in phosphorus content within the nanosheets over time, indicating phosphate leaching from polyP. Additionally, significant aggregation of STPE-PMNSs was observed post-reaction, supporting a change of structure (Fig. 4e). These findings suggest that the polyP component of STPE-PMNSs serves as the phosphate source when there are no external phosphates added (i.e. ATP).

Without STPE-PMNSs, ATP and ammonia by themselves were unable to convert Glu to Gln (Fig. 4g), suggesting that STPE-PMNSs not only serve as a phosphate source but also contribute additional catalysis. We propose that the Mn ions within STPE-PMNSs display catalytic functions similar to their counterparts in natural GS. This is achieved by binding to the negatively charged Glu and reducing the free energy barriers associated with phosphate transfer from polyP, as well as the subsequent aminolysis.

Due to the consumption of the phosphate group in this case, STPE-PMNSs do not strictly fit the definition of a catalyst, but rather, function as a reagent with extra catalytic features (Mn^{2+}). The consumption of phosphates in STPE-PMNSs led us to investigate the duration of sustained activity. Our results demonstrate that the

sustainable catalytic durations of STPE-PMNSs are 60 hours, 48 hours, 48 hours, and 36 hours at Glu concentrations of 2 mM, 10 mM, 20 mM, and 100 mM, respectively. This indicates that the activity durations of STPE-PMNSs diminish as the Glu concentration rises, necessitating a continuous supply of STPE-PMNSs to maintain extended activity (Fig. 4i).

In sharp contrast, we did observe that STPE-PMNSs displayed GS-like catalytic activity when the reaction was conducted at a high level of ATP, exhibiting a linear relationship concerning the concentration of STPE-PMNS. Notably, ATP improved the recycling of STPE-PMNSs, as the phosphorous content did not decrease over time, unlike when ATP was absent (Fig. 4b). TEM images also supported this conclusion, as the STPE-PMNSs remained unchanged post-reaction, indicating their true catalytic nature with sufficient ATP concentration (Fig. 4d). When there is a high ATP level, ATP will be the primary phosphate source and STPE-PMNSs are a catalyst for this reaction. When the ATP level is low and absent, polyP in the nanosheets will provide the necessary phosphate for the activation of Glu, and STPE-PMNSs become a substrate (Fig. 4h).

Besides the roles of polyP and Mn^{2+} , the terminal NH_2 groups in the nanosheets also play an important role in the activation of NH_4^+ (Fig. 5b) by deprotonation. During the reaction, activated substrates align on the surface of the nanosheets, mimicking their binding interactions with natural GS and consequently resulting in effective rate acceleration. Our DFT studies support the proposed catalytic cycle.

In general, the catalytic reaction proceeds in two steps. The initial stage involves the activation of reactants. In the presence of ATP, Glu and ATP interact with the Mn^{2+} on the surface of the nanosheets. Following this interaction, Glu is phosphorylated by ATP, resulting in the formation of activated γ -glutamyl phosphate and ADP with a free energy of -1.01 eV (Fig. 5a). In the absence of ATP, Glu adsorbed on the nanosheets react with the terminal phosphate group, generating γ -glutamyl phosphate polyoxyethylene bis (amine) with a free energy of -0.12 eV (Fig. 5f). This event also generates new unlabeled terminal phosphates. The phosphorylation of Glu lowers the activation energy required for subsequent NH_3 attack and prepares for the nucleophilic attack of

NH₃. The role of Mn²⁺ within the STPE-PMNSs active site is essential, as it serves both structural and catalytic functions by stabilizing the negative charges of the reactants and activating the substrate. It is important to note that the considerable steric hindrance of the STPE group discourages interactions between Glu and adjacent Mn atoms. Consequently, adsorbed Glu is unable to attack polyP-STPE.

The second step corresponds to Gln formation, via a nucleophilic attack of ammonia to the C_γ of the γ -glutamyl phosphate intermediate. There are two basic sites on the nanosheets that can potentially activate NH₄⁺ for aminolysis (Figs. 5b-c and 5e). One is the unlabeled polyP-NH₂, and the other is the newly exposed terminal phosphate after Glu phosphorylation. These regions are also capable of capturing protons from NH₄⁺, leading to the formation of basic residues. The conversion of NH₄⁺ to NH₃ is essential as the ammonium cation is not a nucleophile. The formation of the C_γ-amide bond is believed to be rate-limiting (Figs. 5a and 5g). The energy profile of the overall reaction is shown in Figs. 5d and 5h. The total energy of the reaction in the presence or absence of ATP is -4.6 and -0.77 eV respectively.

Fig. 4 (b) Phosphorus (P) content of STPE-PMNSs with (20 mM) or without ATP.

Fig. 4 TEM image of STPE-PMNSs after 2 h with (d) or without (e) ATP.

Fig. 4 (g) Reactions were conducted to determine whether STPE-PMNSs act solely as reaction substrates or possess catalytic activity in the absence of ATP. In the absence of STPE-PMNSs, ATP (20 mM) and ammonia alone could not convert Glu to Gln, while the reaction proceeded normally with STPE-PMNSs present.

Fig. 4 Conversion duration of STPE-PMNSs at varying Glu concentrations without ATP.

Fig. 4 (h) Correlation between fluorescence intensity and ATP concentration in the reaction revealed a linear relationship in the low-concentration region (inset). At a low ATP level, polyP in the nanosheets activates Glu, with the fluorescence intensity at its highest level when there is no ATP. As ATP concentration increases, ATP gradually replaces polyP in activating Glu, causing the fluorescence intensity to decrease. Once ATP fully replaces polyP, the fluorescence intensity stabilizes and no longer declines.

Fig. 5 Proposed reaction pathway and energies of incremental steps by DFT. (a) The proposed catalytic cycle of the GS-like activity of STPE-PMNSs in the presence of ATP. (b) The activation of NH_4^+ in the presence of ATP. (c) The energy profile for the NH_4^+ activation step. (d) STPE-PMNSs catalyzed conversion of Glu to Gln in the presence of ATP. (e) The energy profile for the activation of NH_4^+ after the terminal polyoxyethylene bis(amine) phosphate is detached. The activation of Glu (f) and Gln formation route (g) using polyP as the phosphate source. (h) The energy profile for STPE-PMNSs promoted conversion of Glu to Gln in the absence of ATP.

Reference

1. Olney, J.W. Brain Lesions, Obesity, and Other Disturbances in Mice Treated with Monosodium Glutamate. *Science* **164**, 719-721 (1969).
2. Li, H. *et al.* Tanshinone IIA Inhibits Glutamate-Induced Oxidative Toxicity through Prevention of Mitochondrial Dysfunction and Suppression of MAPK Activation in SH-SY5Y Human Neuroblastoma Cells. *Oxid. Med. Cell. Longev.* **2017**, 4517486 (2017).

Reviewers' Comments:

Reviewer #1:

Remarks to the Author:

The authors had addressed all issues from the reviewers.

However, there are still obvious problems with the manuscript.

- 1.The catalytic mechanism is still unclear, especially for the activation of NH_4^+ . In addition, the intermediates along the energy profiles were fuzzy.
- 2.The role of Mn^{2+} in STPE-PMNs should be further verified via DFT calculations.
- 3.Again, the authors fail to distinguish between non-catalytic and catalytic reactions. The conclusions on line 398-401, page 22 in the manuscript is error. A non-catalytic reaction cannot be called enzyme-like catalytic reaction.
- 4.The toxicity of the STPE-PMNs and the agglomeration of the STPE-PMNs should be measured.

Reviewer #4:

Remarks to the Author:

The revised manuscript adequately addresses major deficiencies and questions raised by the reviewers. The need for an ATP-independent GS-like activity in neurotoxicity settings is well articulated, supporting the utility of engineered nanomaterials (specifically, PMNSs) for this application. The role of PolyP as an energy source comparable to ATP, as well as ATP concentration-dependent modes of STPE-PMNS activity are discussed, presenting extensive additional data for ATP-free reaction conditions. The proposed reaction pathway accounts for the main modes of activity – in the presence and in the absence of ATP. The proposed neuroprotective effect appears plausible and supported by the data.

The remaining concerns are:

- 1) Structural characterization of PMNSs, particularly with respect to assumptions made for DFT calculations: despite new data, evidence for a completely crystalline structure consistent with DFT predictions is still lacking. Powder XRD results (Sup. Fig. 13a) appear to be reflective of an overall amorphous sample, rather than a crystal. At the same time, HAADF-STEM results (Fig. 2b) indicate that at least some regions are ordered with expected Mn spacing. As such, it might suffice to revise the discussion, addressing the heterogeneity and partially crystalline nature of the material with appropriate disclaimer for the assumptions/limitations of the proposed reaction pathway (by DFT). Additional experimental characterization efforts would likely add little value and fall outside the scope of an already data-heavy manuscript.
- 2) Fate of PMNSs within live cells: the manuscript states that nanosheets are “transported to the lysosome for digestion and degradation”; however, no experimental evidence is presented to support this claim. In the absence of additional data, the authors might consider revising the discussion in the form of a hypothesis rather than a definitive claim. Though, any evidence of PMNS clearing from cells would be beneficial for the manuscript, as these materials appear to exhibit some cell toxicity over time.

Reviewer #5:

Remarks to the Author:

The authors addressed all issues we concerned.

Point-to-point responses to reviewers' comments

Reviewer 1

The authors had addressed all issues from the reviewers.

However, there are still obvious problems with the manuscript.

Query 1

The catalytic mechanism is still unclear, especially for the activation of NH_4^+ . In addition, the intermediates along the energy profiles were fuzzy.

Response

We apologize for the lack of clarity in our previous submission and appreciate the opportunity to provide a clearer explanation of the catalytic mechanism, particularly regarding the activation of NH_4^+ . The conversion of Glu to Gln occurs in two steps, and the reaction mechanism varies depending on the ATP concentration.

(1) In the presence of high ATP concentration, the first step involves Glu activation through phosphorylation, generating a γ -glutamyl phosphate intermediate. Here, ATP serves as the phosphate source, producing ADP as a byproduct. This process is believed to be facilitated by Mn^{2+} ions in the STPE-PMNSs (step 1 in the revised Fig. 5 shown below, with a free energy of -2.86 eV). NH_4^+ is presumed to be activated by the basic unlabeled terminal NH_2 groups in the nanosheets (Supplementary Fig. 61a and 61b). The second step involves aminolysis of the γ -glutamyl phosphate, yielding Gln.

(2) When ATP levels are low or absent, the reaction mechanism differs slightly. In the first step, terminal phosphate groups in the nanosheet act as the phosphate source, generating a γ -glutamyl phosphate polyoxyethylene bis (amine) species (the step from **a**₁ to **b**₁ in the revised Fig. 5b, with a free energy of -0.50 eV). This step also produces basic terminal phosphates in the nanosheets, which can activate NH_4^+ (Supplementary Fig. 62a and 62b). Similarly, the second step involves aminolysis of the γ -glutamyl phosphate polyoxyethylene bis (amine), yielding Gln.

To enhance clarity, we have significantly revised the illustrative mechanism figure

(the new Fig. 5 replaces the old Figs 5 and 6). The energy profiles of the overall reaction, including detailed structures of the intermediates, are depicted in Supplementary Fig. 61c and 62c, with total reaction energies of -4.6 and -0.67 eV in the presence or absence of ATP, respectively.

Regarding NH_4^+ activation, it is believed to be facilitated by the most basic species in the reaction: (1) unlabeled terminal NH_2 's in the presence of ATP, and (2) newly generated terminal phosphates from internal phosphate transfer in the absence of ATP. As there are multiple basic species in this reaction, for example, unlabeled terminal NH_2 groups are also present in the latter case, the identity of the base for NH_4^+ activation is discussed speculatively.

Fig. 5 Proposed reaction mechanisms by DFT. (a) The proposed catalytic cycle of the GS-like activity of STPE-PMNSs in the presence of ATP. (b) The activation of Glu (up) and the formation of Gln (down) in the absence of ATP.

Query 2

The role of Mn^{2+} in STPE-PMNs should be further verified via DFT calculations.

Response

We thank the reviewer's insightful suggestion regarding the verification of Mn^{2+} in STPE-PMNSs using DFT calculations.

In the presence of ATP, both Glu and ATP bind to Mn^{2+} on the surface of the

nanosheets, resulting in the phosphorylation of Glu with a free energy of -2.86 eV (step 1, Fig. 5a; a zoomed-in image is provided below). In contrast, without Mn^{2+} (i.e., in the absence of STPE-PMNSs), the phosphorylation of Glu by ATP is significantly less thermodynamically favorable, with a free energy change of -0.20 eV. This finding suggests that Mn^{2+} ions play a crucial role in the activation of Glu. The DFT data support our experimental results, which demonstrate that only Mn^{2+} in the nanosheets can promote Gln synthesis. Neither Mn^{2+} -free nor unbound Mn^{2+} conditions can facilitate this reaction.

In the absence of ATP, Glu binds to Mn^{2+} on the surface of the nanosheets, similar to the binding mode in the presence of ATP, with a free energy of -0.38 eV. Subsequently, Glu is phosphorylated by the nearby polyP with a free energy change of -0.12 eV (**b1** in Fig. 5b; a zoomed-in image is provided below). In contrast, unbound Glu molecules are negatively charged and repel negatively charged phosphate ions. Based on these DFT results, it is reasonable to propose that Mn^{2+} also plays a crucial role in the activation of Glu by coordinating to the carboxylate, thereby promoting phosphorylation and subsequent aminolysis.

Fig. 5 (a) The proposed catalytic cycle of the GS-like activity of STPE-PMNSs in the presence of ATP.

Fig. 5 (b) The activation of Glu (up) and the formation of Gln (down) in the absence of ATP.

Query 3

Again, the authors fail to distinguish between non-catalytic and catalytic reactions. The conclusions on line 398-401, page 22 in the manuscript is error. A non-catalytic reaction cannot be called enzyme-like catalytic reaction.

Response

We apologize for any confusion caused by our previous response. Our intention was to clearly differentiate between the two scenarios, i.e., with and without ATP, with the latter being a non-catalytic reaction. To ensure a more accurate representation, we have revised the description of "GS-like activity" in the manuscript. Please refer to page 23, lines 5-8 for the updated version.

Query 4

The toxicity of the STPE-PMNs and the agglomeration of the STPE-PMNs should be measured.

Response

We appreciate the reviewer's concern regarding the toxicity and agglomeration of STPE-PMNSs. In response to this concern, we have thoroughly investigated both aspects of STPE-PMNSs.

To assess the toxicity of STPE-PMNSs, we conducted viability tests on SH-SY5Y cells and observed no significant impact on cell viability after 24 hours of treatment with STPE-PMNSs and PMNSs (Supplementary Figs. 64). Furthermore, long-term toxicity tests on three types of nerve cells (SH-SY5Y, PC-12, and U87) showed that cell viability remained at 80% after exposure to STPE-PMNSs (200 $\mu\text{g}/\text{mL}$) for 7 days,

indicating excellent biocompatibility (Supplementary Fig. 66).

Regarding the agglomeration of STPE-PMNSs, we first observed aggregation outside the cell, and as the reaction progressed, the fluorescence of the system gradually increased when the ATP content was below 10 μM (Fig. 4c). TEM images and DLS analysis revealed that the aggregation of the nanosheets became more pronounced with the extension of time (Supplementary Fig. 59-60).

Furthermore, we investigated the intracellular aggregation of nanosheets. In the neurotoxic SH-SY5Y cells and STPE-PMNSs co-treatment group, the fluorescence intensity continuously increased as the reaction progressed, consistent with extracellular observations and potentially attributable to the aggregation of STPE-PMNSs (Fig. 6a). This enhanced fluorescence signal serves as a valuable tool for monitoring the catalytic reaction's progression within cells (Figs 6b-d).

We hope that these findings address the reviewer's concerns and provide a comprehensive understanding of the toxicity and agglomeration of STPE-PMNSs.

Supplementary Fig. 64 The Cell viability of SH-SY5Y cells treated with STPE-PMNSs and PMNSs for 24 h.

Supplementary Fig. 66 (a) Cell viability of SH-SY5Y, PC-12, U87 cells treated with 25 $\mu\text{g mL}^{-1}$ STPE-PMNSs for 7 d. (b) ICP results of SH-SY5Y, PC-12, U87 cells treated with 25 $\mu\text{g mL}^{-1}$ STPE-PMNSs for 7 d.

Supplementary Fig. 59 The TEM image (a) and DLS analysis (b) of STPE-PMNSs after reaction at different time points. STPE-PMNSs ($100 \mu\text{g mL}^{-1}$) promoted the conversion of Glu to Gln in 5 mL HEPES buffer (100 mM, pH 7.3) in the absence of ATP. After different reaction times, nanosheets were collected by centrifugation at 6000 rpm for 10 minutes and subsequently characterized using TEM and DLS.

Supplementary Fig. 60 The TEM images of STPE-PMNSs before (a) and after (b) 2 h reaction in the absence of ATP. STPE-PMNSs ($100 \mu\text{g mL}^{-1}$) promoted the

conversion of Glu to Gln in 5 mL HEPES buffer (100 mM, pH 7.3). After reaction 2 h, nanosheets were collected by centrifugation at 6000 rpm for 10 minutes and subsequently characterized using TEM.

Fig.4 (c) Changes in fluorescence intensity during the reaction.

Fig. 6 Assessment of the intracellular GS-like activity of STPE-PMNSs. Intracellular fluorescence signals were monitored at different time points (a), followed by quantification of the respective fluorescent signals (b). Cells underwent treatment with STPE-PMNSs for 6 h, followed by PBS washing. Glu production was induced in SH-SY5Y cells using 4-AP for 4 h, and cells were washed with PBS again. Intracellular fluorescence signals were subsequently observed at different time points, with corresponding signal quantification. Changes in intracellular Gln content were recorded at various time points (c), along with the linear relationship between Gln content and fluorescence intensity (d).

Reviewer 4

The revised manuscript adequately addresses major deficiencies and questions raised by the reviewers. The need for an ATP-independent GS-like activity in neurotoxicity settings is well articulated, supporting the utility of engineered nanomaterials (specifically, PMNSs) for this application. The role of PolyP as an energy source comparable to ATP, as well as ATP concentration-dependent modes of STPE-PMNS activity are discussed, presenting extensive additional data for ATP-free reaction conditions. The proposed reaction pathway accounts for the main modes of activity – in the presence and in the absence of ATP. The proposed neuroprotective effect appears plausible and supported by the data.

The remaining concerns are:

Query 1

Structural characterization of PMNSs, particularly with respect to assumptions made for DFT calculations: despite new data, evidence for a completely crystalline structure consistent with DFT predictions is still lacking. Powder XRD results (Sup. Fig. 13a) appear to be reflective of an overall amorphous sample, rather than a crystal. At the same time, HAADF-STEM results (Fig. 2b) indicate that at least some regions are ordered with expected Mn spacing. As such, it might suffice to revise the discussion, addressing the heterogeneity and partially crystalline nature of the material with appropriate disclaimer for the assumptions/limitations of the proposed reaction pathway (by DFT). Additional experimental characterization efforts would likely add little value and fall outside the scope of an already data-heavy manuscript.

Response

We appreciate the reviewer's insightful comment regarding the structural characterization of PMNSs and the assumptions made for DFT calculations. We agree that despite the new data, evidence for a completely crystalline structure consistent with DFT predictions is still lacking. We acknowledge that the Powder XRD results (Sup. Fig. 13a) appear to be reflective of an overall amorphous sample, rather than a crystal. However, the HAADF-STEM results (Fig. 2b) indicate some ordered regions.

Considering these findings, we agree that it would be appropriate to revise the discussion to address the heterogeneity and partially crystalline nature of the material and to include appropriate disclaimers for the assumptions and limitations of the proposed reaction pathway (by DFT). We have made the revisions in both the discussions of the structure of those nanosheets and DFT calculations. Please refer to page 6, lines 5-11, and page 16, lines 15-18.

Query 2

Fate of PMNSs within live cells: the manuscript states that nanosheets are “transported to the lysosome for digestion and degradation”; however, no experimental evidence is presented to support this claim. In the absence of additional data, the authors might consider revising the discussion in the form of a hypothesis rather than a definitive claim. Though, any evidence of PMNS clearing from cells would be beneficial for the manuscript, as these materials appear to exhibit some cell toxicity over time.

Response

We appreciate the reviewer for raising a crucial concern regarding the fate of PMNSs within live cells. We apologize for not being clearer in our previous submission. In response to this concern, we conducted further investigations into the intracellular retention of STPE-PMNSs.

We observed a gradual decrease in intracellular Mn content over time (Supplementary Fig. 68). This suggests that the material can be effectively cleared from the cells. Furthermore, our results indicated that the STPE-PMNSs co-localized with lysosomes (Supplementary Fig. 72). Based on these findings, it appears that STPE-PMNSs are likely eliminated from the cells through the lysosomal pathway, thereby regulating the intracellular dynamic equilibrium of STPE-PMNSs.

We agree with the reviewer's suggestion to revise the discussion in the form of a hypothesis rather than a definitive claim. We have revised the discussion accordingly and have included the findings on the clearing of STPE-PMNSs in a more speculative

tone. We appreciate the reviewer's feedback, and we hope that these findings and revisions address the concerns raised.

Supplementary Fig. 68 ICP-MS analysis of Mn content. SH-SY5Y cells were cultured in a 6-well plate and treated with STPE-PMNSs ($25 \mu\text{g mL}^{-1}$) for 12 h. The cells were then washed with PBS three times to remove the remaining STPE-PMNSs on the cell surface. After washing, the cells were cultured for 0, 3, 6, 9, 12, and 24 h, the cells were collected and the Mn content was measured using ICP-MS.

Supplementary Fig. 72 The subcellular distribution of STPE-PMNS.

Reviewer 5

The authors addressed all issues we concerned.

Response

We thank reviewer 5's kind comments on our previous revision.

Reviewers' Comments:

Reviewer #1:

Remarks to the Author:

The revised manuscript addresses major questions raised by the reviewers. The proposed reaction pathway seems plausible. However, there are still some concerns.

Query 1

The DFT calculated energy was referred to free energy in the manuscript. However, no entropy energy had been calculated.

Query 2

STPE-PMNSs obtains a proton during the activation of NH_4^+ in the presence of ATP. Whether the structure of the material has been restored after completing the catalytic reaction cycle.

Query 3

On line 259-261, page 15, "In contrast,with a free energy change of -0.20 eV ". Supporting information should be provided to support the data.

Query 4

It is best to provide the cif format structure of all intermediates calculated by DFT.

Query 5

From the experimental results, the content of ATP in the cell is not enough to provide enough phosphate to prevent STPE-PMNSs agglomeration. In general, after the material is aggregated in the cell, its toxicity to nerve cells will be increased.

Reviewer #4:

Remarks to the Author:

The revised manuscript has substantially clarified and expanded the proposed reaction mechanism through additional in-depth discussion and the revised figures. Structural characterization of PMNSs has been revised to reflect the presented data, and further evidence has been provided to corroborate material biocompatibility and clearing from the cells. Overall, all of the main concerns and questions raised through prior rounds of review have been adequately addressed.

Point-to-point responses to reviewers' comments.

Reviewer 1

The revised manuscript addresses major questions raised by the reviewers. The proposed reaction pathway seems plausible. However, there are still some concerns.

Query 1

The DFT calculated energy was referred to free energy in the manuscript. However, no entropy energy had been calculated.

Response

We sincerely apologize for any confusion regarding the DFT calculated energy in our previous submission and are grateful for the opportunity to make clarifications.

In our previous submission, the DFT calculated energy was erroneously referred to as “free energy” and we have duly corrected it to “energy” in the revised manuscript.

Given that the number of molecules remains constant before and after the reaction, and there are no phase changes in the resulting substances, we consider the entropy contribution in free energy difference to be relatively small. Therefore, the reaction kinetics and thermodynamics are dominated by energies, which were computed in our work.

Query 2

STPE-PMNSs obtains a proton during the activation of NH_4^+ in the presence of ATP. Whether the structure of the material has been restored after completing the catalytic reaction cycle.

Response

We appreciate the reviewer for raising a crucial concern regarding the structure of the material whether restored after completing the catalytic reaction cycle.

In our experiments, the reaction was conducted at a pH value comparable to that of bodily fluids (pH 7.4, HEPES buffer). Under this condition, the $-\text{NH}_2$ groups within STPE-PMNSs were already in a dynamic equilibrium with $-\text{NH}_3^+$ before initiating the reaction. Even upon activating NH_4^+ , the initial equilibrium is restored.

Query 3

On line 259-261, page 15, “In contrast,with a free energy change of -0.20 eV”. Supporting information should be provided to support the data.

Response

We thank the reviewer’s insightful suggestion.

Supplementary Fig. 61 displays the optimized configurations and energies of ATP, Glu, and ATP-Glu. When Mn^{2+} is not present in the system (i.e., in the absence of STPE-PMNSs), the change in binding energy of ATP and Glu is -0.0075 a.u. (-0.20 eV).

Supplementary Fig. 61 The optimized configurations and energies of ATP, Glu, and ATP-Glu in the absence of STPE-PMNSs.

Query 4

It is best to provide the cif format structure of all intermediates calculated by DFT.

Response

We appreciate the reviewer's insightful comment and have included the CIF files of the intermediates in the supplementary materials.

Query 5

From the experimental results, the content of ATP in the cell is not enough to provide enough phosphate to prevent STPE-PMNSs agglomeration. In general, after the material is aggregated in the cell, its toxicity to nerve cells will be increased.

Response

We appreciate the reviewer's concern regarding the toxicity of aggregated STPE-PMNSs.

To assess whether the aggregation of STPE-PMNSs enhances neurotoxicity, the SH-SY5Y cells were first treated with 4-AP for 4 hours to overexpress Glu, followed by treatment with STPE-PMNSs for 24 hours. During this process, the nanosheets will aggregate due to the reactions. However, even at a concentration of 200 $\mu\text{g/mL}$, the cell survival rate remained at 78.27%, showing no significant deviation from the survival rate observed with the administration of STPE-PMNSs (74.18%) alone. These findings suggest that the aggregation of STPE-PMNSs does not increase cytotoxicity (Supplementary Fig. 68).

Supplementary Fig. 68 The Cell viability of SH-SY5Y cells treated with STPE-PMNSs for 24 h after 4-AP treatment 4 h.

Reviewer 4

The revised manuscript has substantially clarified and expanded the proposed reaction mechanism through additional in-depth discussion and the revised figures. Structural characterization of PMNSs has been revised to reflect the presented data, and further evidence has been provided to corroborate material biocompatibility and clearing from the cells. Overall, all of the main concerns and questions raised through prior rounds of review have been adequately addressed.

Response

We thank reviewer 4's kind comments on our previous revision.

Reviewers' Comments:

Reviewer #1:

Remarks to the Author:

All of the main concerns and questions raised have been adequately addressed. The publication of this article is recommend.